# Riemannian Score-Based Generative Modelling

**Valentin De Bortoli**[*†], **Émile Mathieu**[*‡], **Michael Hutchinson**[*‡],

**James Thornton**[‡], **Yee Whye Teh**[‡], **Arnaud Doucet**[‡]

## Abstract

Score-based generative models (SGMs) are a powerful class of generative models that exhibit remarkable empirical performance. Score-based generative modelling (SGM) consists of a "noising" stage, whereby a diffusion is used to gradually add Gaussian noise to data, and a generative model, which entails a "denoising" process defined by approximating the time-reversal of the diffusion. Existing SGMs assume that data is supported on a Euclidean space, i.e. a manifold with flat geometry. In many domains such as robotics, geoscience or protein modelling, data is often naturally described by distributions living on Riemannian manifolds and current SGM techniques are not appropriate. We introduce here *Riemannian Score-based Generative Models* (RSGMs), a class of generative models extending SGMs to Riemannian manifolds. We demonstrate our approach on a variety of manifolds, and in particular with earth and climate science spherical data.

## 1 Introduction

Score-based Generative Models (SGMs) also called diffusion models (Song and Ermon, 2019; Song et al., 2021; Ho et al., 2020; Dhariwal and Nichol, 2021) formulate generative modelling as a denoising process. Noise is incrementally added to data using a diffusion process until it becomes approximately Gaussian. The generative model is then obtained by simulating an approximation of the corresponding time-reversal process, which progressively denoises a Gaussian sample to obtain a data sample. This process is also a diffusion whose drift depends on the logarithmic gradients of the noised data densities, i.e. the Stein scores, estimated using a neural network via score matching (Hyvärinen, 2005; Vincent, 2011).

SGMs have been primarily applied to data living on Euclidean spaces, i.e. manifolds with flat geometry. However, in a large number of scientific domains the distributions of interest are supported on Riemannian manifolds. These include, to name a few, protein modelling (Shapovalov and Dunbrack Jr, 2011), cell development (Klimovskaia et al., 2020), image recognition (Lui, 2012), geological sciences (Karpatne et al., 2018; Peel et al., 2001), graph-structured and hierarchical data (Roy et al., 2007; Steyvers and Tenenbaum, 2005), robotics (Feiten et al., 2013; Senanayake and Ramos, 2018) and high-energy physics (Brehmer and Cranmer, 2020).

We introduce in this work *Riemannian Score-based Generative Models* (RSGMs), an extension of SGMs to Riemannian manifolds which incorporate the geometry of the data by defining the forward diffusion process directly on the Riemannian manifold, inducing a manifold-valued reverse process. This requires constructing a noising process on the manifold that converges to an easy-to-sample reference distribution. We establish that, as in the Euclidean case, the corresponding time-reversal process is also a diffusion whose drift includes the Stein score which is intractable but can similarly be estimated via score matching. Methodological extensions are required as in most cases the transition kernel of the noising process cannot be sampled exactly. For example on compact manifolds it is

---

[*]equal contribution.

[†]Dept. of Computer Science ENS, CNRS, PSL University Paris, France.

[‡]Dept. of Statistics, University of Oxford, Oxford, UK.

36th Conference on Neural Information Processing Systems (NeurIPS 2022).

typically only available as an infinite sum through the Sturm–Liouville decomposition (Chavel, 1984). To this end, we develop non-standard techniques for score estimation and rely on the use of Geodesic Random Walks for sampling (Jørgensen, 1975). We provide theoretical convergence bounds for RSGMs on compact manifolds and demonstrate our approach on a range of manifolds and tasks, including modelling a number of natural disaster occurrence datasets collected by Mathieu and Nickel (2020). We show that RGSMs achieve better performance than recent baselines (Mathieu and Nickel, 2020; Rozen et al., 2021) and scale better to high-dimensional manifolds.

## 2 Euclidean Score-based Generative Modelling

We recall here briefly the key concepts behind SGMs on the Euclidean space $\mathbb{R}^d$ and refer the readers to Song et al. (2021) for a more detailed introduction. We consider a forward *noising* process $(\mathbf{X}_t)_{t \geq 0}$ defined by the following Stochastic Differential Equation (SDE)

$$\mathrm{d}\mathbf{X}_t = -\mathbf{X}_t \mathrm{d}t + \sqrt{2}\mathrm{d}\mathbf{B}_t, \quad \mathbf{X}_0 \sim p_0, \tag{1}$$

where $(\mathbf{B}_t)_{t \geq 0}$ is a $d$-dimensional Brownian motion and $p_0$ is the data distribution. The available data gives us an empirical approximation of $p_0$. The process $(\mathbf{X}_t)_{t \geq 0}$ is simply an Ornstein–Ulhenbeck (OU) process which converges with geometric rate to $\mathrm{N}(0, \mathrm{Id})$. Under mild conditions on $p_0$, the time-reversed process $(\mathbf{Y}_t)_{t \geq 0} = (\mathbf{X}_{T-t})_{t \in [0,T]}$ also satisfies an SDE (Cattiaux et al., 2021; Haussmann and Pardoux, 1986) given by

$$\mathrm{d}\mathbf{Y}_t = \{\mathbf{Y}_t + 2\nabla \log p_{T-t}(\mathbf{Y}_t)\}\mathrm{d}t + \sqrt{2}\mathrm{d}\mathbf{B}_t, \quad \mathbf{Y}_0 \sim p_T, \tag{2}$$

where $p_t$ denotes the density of $\mathbf{X}_t$. By construction, the law of $\mathbf{Y}_{T-t}$ is equal to the law of $\mathbf{X}_t$ for $t \in [0, T]$ and in particular $\mathbf{Y}_T \sim p_0$. Hence, if one could sample from $(\mathbf{Y}_t)_{t \in [0,T]}$ then its final distribution would be the data distribution $p_0$. Unfortunately we cannot sample exactly from (2) as $p_T$ and the scores $(\nabla \log p_t(x))_{t \in [0,T]}$ are intractable. Hence SGMs rely on a few approximations. First, $p_T$ is replaced by the reference distribution $\mathrm{N}(0, \mathrm{Id})$ as we know that $p_T$ converges geometrically towards it. Second, the following denoising score matching identity is exploited to estimate the scores

$$\nabla_{x_t} \log p_t(x_t) = \int_{\mathbb{R}^d} \nabla_{x_t} \log p_{t|0}(x_t|x_0) \, p_{0|t}(x_0|x_t)\mathrm{d}x_0,$$

where $p_{t|0}(x_t|x_0)$ is the transition density of the OU process (1) which is available in closed-form. It follows directly that $\nabla \log p_t$ is the minimizer of $\ell_t(\mathbf{s}) = \mathbb{E}[\|\mathbf{s}(\mathbf{X}_t) - \nabla_{x_t} \log p_{t|0}(\mathbf{X}_t|\mathbf{X}_0)\|^2]$ over functions $\mathbf{s}$ where the expectation is over the joint distribution of $\mathbf{X}_0, \mathbf{X}_t$. This result can be leveraged by considering a neural network $\mathbf{s}_\theta : [0, T] \times \mathbb{R}^d \to \mathbb{R}^d$ trained by minimizing the loss function $\ell(\theta) = \int_0^T \lambda_t \ell_t(\mathbf{s}_\theta(t, \cdot))\mathrm{d}t$ for some weighting function $\lambda_t > 0$. Finally, an Euler–Maruyama discretization of (2) is performed using a discretization step $\gamma$ such that $T = \gamma N$ for $N \in \mathbb{N}$

$$Y_{n+1} = Y_n + \gamma\{Y_n + 2\mathbf{s}_\theta(T - n\gamma, Y_n)\} + \sqrt{2\gamma}Z_{n+1}, \quad Y_0 \sim \mathrm{N}(0, \mathrm{Id}), \quad Z_n \overset{\text{i.i.d.}}{\sim} \mathrm{N}(0, \mathrm{Id}).$$

The above showcases the basics of SGMs but we highlight that many improvements have been proposed; see (e.g. Song and Ermon, 2020; Jolicoeur-Martineau et al., 2021; Dhariwal and Nichol, 2021). In particular, selecting an adaptive stepsize $(\gamma_n)_{n \in \mathbb{N}}$ (Bao et al., 2022; Watson et al., 2021) and using a predictor-corrector scheme (Song et al., 2021) instead of a simple Euler–Maruyama discretization drastically improves performance.

## 3 Riemannian Score-based Generative Modelling

We now move to the Riemannian manifold setting, and more specifically assume that $\mathcal{M}$ is a complete, orientable connected and boundaryless Riemannian manifold, endowed with a Riemannian metric $g$ [4]. Four components are required to extend SGMs to this setting: i) a forward *noising* process on $\mathcal{M}$ which converges to an easy-to-sample reference distribution, ii) a time-reversal formula on $\mathcal{M}$ which defines a backward generative process, iii) a method for approximating samples of SDEs on manifolds, iv) a method to efficiently approximate the drift of the time-reversal process. Notation are gathered in App. B.

---

[4]Metrics $g$ are sections of $\mathrm{T}^*\mathcal{M} \otimes \mathrm{T}^*\mathcal{M}$, the rank 2 tensor bundle of the dual tangent space, i.e. smooth varying bilinear maps on $\mathrm{T}\mathcal{M}$, verifying symmetry and positive semi-definiteness.

## 3.1 Noising processes on manifolds

The first necessary component is a suitable generic noising process on manifolds that will converge to a convenient stationary distribution. A simple choice is to use Langevin dynamics described by

$$d\mathbf{X}_t = -\tfrac{1}{2} \nabla_{\mathbf{X}_t} U(\mathbf{X}_t) dt + d\mathbf{B}_t^{\mathcal{M}}, \tag{3}$$

which admits the invariant density (w.r.t. the volume form) given by $dp_{\mathrm{ref}}/d\mathrm{Vol}_{\mathcal{M}}(x) \propto e^{-U(x)}$ (Durmus, 2016, Section 2.4), where $\nabla$ is the Riemannian gradient[5].

Two simple choices for $U(x)$ present themselves. Firstly, setting $U(x) = d_{\mathcal{M}}(x,\mu)^2/(2\gamma^2)$, where $d_{\mathcal{M}}$ is the geodesic distance and $\mu \in \mathcal{M}$ is an arbitrary mean location, induces the drift $\nabla_{\mathbf{X}_t} U(\mathbf{X}_t) = -\exp_{\mathbf{X}_t}^{-1}(\mu)/\gamma^2$ [6]. This is the potential of the 'Riemannian normal' (Pennec, 2006) distribution, from which it is in general neither trivial to sample nor to compute the normalisation constant (Hauberg, 2018; Mathieu et al., 2019). An alternative is to target the 'exponential wrapped' Gaussian. This is the pushforward of a Gaussian distribution in the tangent space at the mean location along the exponential map. The potential is given by $U(x) = d_{\mathcal{M}}(x,\mu)^2/(2\gamma^2) + \log|\partial \exp_\mu^{-1}(x)|$ [7]. In contrast to the Riemannian normal, sampling and evaluating the density of this distribution is easy.

One recovers the standard Ornstein–Uhlenbeck noising process (Song et al., 2021) for both of these target distributions when $\mathcal{M} = \mathbb{R}^d$ and $\mu = 0$ since then the drift $b(t,\mathbf{X}_t) = \tfrac{1}{2} \exp_{\mathbf{X}_t}^{-1}(0) = -\tfrac{1}{2}\mathbf{X}_t$. On compact manifolds, the invariant measure $\mathrm{Vol}_{\mathcal{M}}$ has finite volume, thus a natural choice is to target the uniform distribution which is given by $\mathrm{Vol}_{\mathcal{M}}/|\mathcal{M}|$. In this case, $\nabla_{\mathbf{X}_t} U(\mathbf{X}_t) = 0$ and the noising process is simply a Brownian motion on $\mathcal{M}$.

## 3.2 Time-reversal on Riemannian manifolds

In order to use these noising processes we prove the time-reversal formula for manifolds, a generalisation of the results in the Euclidean case, e.g. see Cattiaux et al. (2021, Theorem 4.9). Consider an SDE of the form $d\mathbf{X}_t = b(\mathbf{X}_t)dt + d\mathbf{B}_t^{\mathcal{M}}$ where $\mathbf{B}_t^{\mathcal{M}}$ is a Brownian motion on $\mathcal{M}$. We refer to App. C.3 for an introduction to Brownian motions on manifolds. This result shows that if $(\mathbf{X}_t)_{t\in[0,T]}$ is a diffusion process then $(\mathbf{X}_{T-t})_{t\in[0,T]}$ is also a diffusion process w.r.t. the backward filtration whose coefficients can be computed, and are shown in Eq. (4). The proof relies on an extension of Cattiaux et al. (2021, Theorem 4.9) to the Riemannian manifold case and is postponed to App. H.

**Theorem 1** (Time-reversed diffusion). *Let $T \geq 0$ and $(\mathbf{B}_t^{\mathcal{M}})_{t\geq 0}$ be a Brownian motion on $\mathcal{M}$ such that $\mathbf{B}_0^{\mathcal{M}}$ has distribution the volume form $p_{\mathrm{ref}}$[8]. Let $(\mathbf{X}_t)_{t\in[0,T]}$ be associated with the SDE $d\mathbf{X}_t = b(\mathbf{X}_t)dt + d\mathbf{B}_t^{\mathcal{M}}$. Let $(\mathbf{Y}_t)_{t\in[0,T]} = (\mathbf{X}_{T-t})_{t\in[0,T]}$ and assume that $\mathrm{KL}(\mathbb{P}|\mathbb{Q}) < +\infty$, where $\mathbb{Q}$ is the distribution of $(\mathbf{B}_t^{\mathcal{M}})_{t\in[0,T]}$ and $\mathbb{P}$ the distribution of $(\mathbf{X}_t)_{t\in[0,T]}$. In addition, assume that $\mathbb{P}_t = \mathcal{L}(\mathbf{X}_t)$, the distribution of $\mathbf{X}_t$, admits a smooth positive density $p_t$ w.r.t. $p_{\mathrm{ref}}$ for any $t \in [0,T]$. Then, $(\mathbf{Y}_t)_{t\in[0,T]}$ is associated with the SDE*

$$d\mathbf{Y}_t = \{-b(\mathbf{Y}_t) + \nabla \log p_{T-t}(\mathbf{Y}_t)\}dt + d\mathbf{B}_t^{\mathcal{M}}. \tag{4}$$

## 3.3 Approximate sampling of diffusions

Obtaining samples from SDEs on a manifold is non-trivial in general. If $\mathcal{M}$ is isometrically embedded into $\mathbb{R}^p$ (with $p \geq d$) one can define $(\mathbf{B}_t^{\mathcal{M}})_{t\geq 0}$ as a $\mathbb{R}^p$-valued process, see App. C.3. However, this approach is *extrinsic*, as it requires the knowledge of the projection operator to place points back on the manifold at each step which can accumulate errors. Here we consider an *intrisic* approach based on Geodesic Random Walks (GRWs), see Jørgensen (1975) for a review of their properties. GRWs can approximate *any* well-behaved diffusion on $\mathcal{M}$. Hence, we introduce GRWs in a general framework and consider a discrete-time process $(X_n^\gamma)_{n\in\mathbb{N}}$ which approximates the diffusion $(\mathbf{X}_t)_{t\geq 0}$ defined by

$$d\mathbf{X}_t = b(t,\mathbf{X}_t)dt + \sigma(t,\mathbf{X}_t)d\mathbf{B}_t^{\mathcal{M}}. \tag{5}$$

This generalisation is key to sampling the backward diffusion process defined in Theorem 1.

---

[5]The (Riemannian) gradient $\nabla$ is defined s.t. for any $f : \mathcal{M} \to \mathbb{R}$, $x \in \mathcal{M}$, $v \in T_x\mathcal{M}$, $\langle \nabla f, v \rangle_g = df(v)$.

[6]$\exp_x : T_x\mathcal{M} \to \mathcal{M}$ denotes the exponential mapping on the manifold, see e.g. Lee (2013, Chapter 20).

[7]$|\cdot|$ denotes the absolute value of the determinant, and $\partial f$ the Jacobian of $f$.

[8]Note that in the case of a non-compact manifold $p_{\mathrm{ref}}$ is only a measure and not a probability measure.

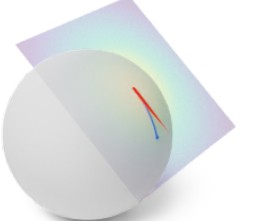 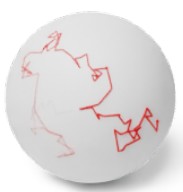 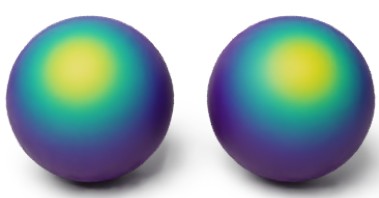

(a) A single step of a Geodesic Random Walk.

(b) Many steps yield an approximate Brownian motion trajectory.

(c) The density of a single step of Gaussian Random Walk [Left] and the Brownian motion density [Right] agree well for small time steps.

Figure 1: Geodesic Random Walks can be used to approximate Brownian motion and more generally SDEs on manifolds. (a) At each step, tangential noise is sampled (red), which is added the drift term (not pictured). This tangent vector is then pushed through the exponential map to produce a geodesics step on the manifold (blue). (b) Iterating this procedure yield approximate sample paths from the process.

---

**Algorithm 1** GRW (Geodesic Random Walk)

---

**Require:** $T, N, X_0^\gamma, b, \sigma, \mathrm{P}$
1: $\gamma = T/N$       ▷ Step-size
2: **for** $k \in \{0, \dots, N-1\}$ **do**
3:     $Z_{k+1} \sim \mathrm{N}(0, \mathrm{Id})$       ▷ Sample a Gaussian in the tangent space of $X_k^\gamma$
4:     $W_{k+1} = \gamma b(k\gamma, X_k^\gamma) + \sqrt{\gamma}\sigma(k\gamma, X_k^\gamma)Z_{k+1}$   ▷ Compute the Euler–Maruyama step on tangent space
5:     $X_{k+1}^\gamma = \exp_{X_k^\gamma}[W_{k+1}]$       ▷ Move along the geodesic defined by $W_{k+1}$ and $X_k^\gamma$ on $\mathcal{M}$
6: **end for**
7: **return** $\{X_k^\gamma\}_{k=0}^N$

---

**Definition 2** (Geodesic Random Walk). *Let $X_0^\gamma$ be a $\mathcal{M}$-valued random variable. For any $\gamma > 0$, we define $(X_n^\gamma)_{n \in \mathbb{N}}$ such that for any $n \in \mathbb{N}$, $X_{n+1}^\gamma = \exp_{X_n^\gamma}[\gamma\{b(X_n^\gamma) + \sqrt{\gamma}V_{n+1}\}]$, where $(V_n)_{n \in \mathbb{N}}$ is a sequence of $\mathrm{T}\mathcal{M}$-valued random variables such that for any $n \in \mathbb{N}$, $\mathbb{E}[V_{n+1}|\mathcal{F}_n] = 0$ and $\mathbb{E}[V_{n+1}V_{n+1}^\top|\mathcal{F}_n] = \sigma\sigma^\top(X_n^\gamma)$, where $\mathcal{F}_n$ is the filtration generated by $\{X_k^\gamma\}_{k=0}^n$. We say that the $\mathcal{M}$-valued process $(X_n^\gamma)_{n \in \mathbb{N}}$ is a Geodesic Random Walk.*

Algorithm 1 approximately simulates the diffusion $(\mathbf{X}_t)_{t \in [0,T]}$ defined in Eq. (5) using GRWs; see Kuwada (2012); Cheng et al. (2022) for quantitative error bounds in the time-homogeneous case and App. I.2 for a novel extenstion for the time-inhomogeneous case. Fig. 1 provides a graphical illustration of this procedure.

### 3.4 Score approximation on Riemannian manifolds

**Score matching and loss functions.** The reverse process from Eq. (4) involves the Stein score $\nabla \log p_t$ which is unfortunately intractable. To derive an approximation, we first remark that for any $s, t \in (0, T]$ with $t > s$ and $x_t \in \mathcal{M}$, $p_t(x_t) = \int_\mathcal{M} p_{t|s}(x_t|x_s)\mathrm{d}\mathbb{P}_s(x_s)$, where $\mathbb{P}_s = \mathcal{L}(\mathbf{X}_s)$, the distribution of $\mathbf{X}_s$. Thus, we have that for any $s, t \in [0, T]$ with $t > s$ and $x_t \in \mathcal{M}$

$$\nabla_{x_t} \log p_t(x_t) = \int_\mathcal{M} \nabla_{x_t} \log p_{t|s}(x_t|x_s)\mathbb{P}_{s|t}(x_t, \mathrm{d}x_s).$$

Hence, for any $s, t \in [0, T]$ with $t > s$ we have that $\nabla \log p_t = \arg\min\{\ell_{t|s}(\mathbf{s}_t) \,:\, \mathbf{s}_t \in \mathrm{L}^2(\mathbb{P}_t)\}$,

where $\ell_{t|s}(\mathbf{s}_t) = \int_{\mathcal{M}^2} \|\nabla_x \log p_{t|s}(x_t|x_s) - \mathbf{s}_t(x_t)\|^2 \mathrm{d}\mathbb{P}_{s,t}(x_s, x_t)$, which is referred as the Denoising Score Matching (DSM) loss. It can also be written in an *implicit* fashion.

**Proposition 3.** *Let $t, s \in (0, T]$ with $t > s$. Then, for any $\mathbf{s}_t \in \mathrm{C}^\infty(\mathcal{M})$, $\ell_{t|s}(\mathbf{s}_t) = 2\ell_t^{\mathrm{im}}(\mathbf{s}_t) + \int_{\mathcal{M}^2} \|\nabla_{x_t} \log p_{t|s}(x_t|x_s)\|^2 \mathrm{d}\mathbb{P}_{s,t}(x_s, x_t)$, where $\ell_t^{\mathrm{im}}(\mathbf{s}_t) = \int_\mathcal{M} \{\frac{1}{2}\|\mathbf{s}_t(x_t)\|^2 + \mathrm{div}(\mathbf{s}_t)(x_t)\}\mathrm{d}\mathbb{P}_t(x_t)$.*

The proof is postponed to App. J. For any $t \in (0, T]$ the minimizers of the loss $\ell_t^{\mathrm{im}}$ on $\mathcal{X}(\mathcal{M})$ (where $\mathcal{X}(\mathcal{M})$ is the set of vector fields on $\mathcal{M}$) are the same as the ones for $\ell_{t|s}$. The loss $\ell_t^{\mathrm{im}}$ is referred to as the *implicit* score matching (ISM) loss (Hyvärinen, 2005). These losses are direct analogous to the versions typically used in Euclidean space.

In the case where we have access to $\{\nabla \log p_{t|s} \,:\, T \le t > s \ge 0\}$, the forward noising process transition kernels, or an approximation of this family, then we can use the DSM loss to learn

**Algorithm 2** RSGM (Riemannian Score-Based Generative Model)

**Require:** $\varepsilon, T, N, \{X_0^m\}_{m=1}^M, \text{loss}, \mathbf{s}, \theta_0, N_{\text{iter}}, p_{\text{ref}}, \mathrm{P}$
  1: /// TRAINING ///
  2: **for** $n \in \{0, \dots, N_{\text{iter}} - 1\}$ **do**
  3:     $X_0 \sim (1/M) \sum_{m=1}^M \delta_{X_0^m}$  ▷ Random mini-batch from dataset
  4:     $t \sim U([\varepsilon, T])$  ▷ Uniform sampling between $\varepsilon$ and $T$
  5:     $\mathbf{X}_t = \text{GRW}(t, N, X_0, b, \text{Id}, \mathrm{P})$  ▷ Approximate forward diffusion with Algorithm 1
  6:     $\ell(\theta_n) = \ell_t(T, N, X_0, \mathbf{X}_t, \text{loss}, \mathbf{s}_{\theta_n})$  ▷ Compute score matching loss from Table 2
  7:     $\theta_{n+1} = \texttt{optimizer\_update}(\theta_n, \ell(\theta_n))$  ▷ ADAM optimizer step
  8: **end for**
  9: $\theta^\star = \theta_{N_{\text{epoch}}}$
 10: /// SAMPLING ///
 11: $Y_0 \sim p_{\text{ref}}$  ▷ Sample from uniform distribution
 12: $b_\theta^\star(t, x) = \mathbf{s}_{\theta^\star}(T - t, x)$ for any $t \in [0, T], x \in \mathcal{M}$  ▷ Reverse process drift
 13: $\{Y_k\}_{k=0}^N = \text{GRW}(T, N, Y_0, b_{\theta^\star}, \text{Id}, \mathrm{P})$  ▷ Approximate reverse diffusion with Algorithm 1
 14: **return** $\theta^\star, \{Y_k\}_{k=0}^N$

Table 1: Differences between SGM on Euclidean spaces and RSGM on Riemannian manifolds.

| Ingredient \ Space | Euclidean | 'Generic' Manifold | Compact Manifold |
|---|---|---|---|
| Forward process $\mathrm{d}\mathbf{X}_t =$ | $-\frac{1}{2}\mathbf{X}_t \mathrm{d}t + \mathrm{d}\mathbf{B}_t^{\mathcal{M}}$ | $-\frac{1}{2}\nabla_{\mathbf{X}_t} U(\mathbf{X}_t)\mathrm{d}t + \mathrm{d}\mathbf{B}_t^{\mathcal{M}}$ | $\mathrm{d}\mathbf{B}_t^{\mathcal{M}}$ |
| Easy-to-sample distribution | Gaussian | Wrapped Gaussian | Uniform |
| Time reversal | Cattiaux et al. (2021) | Theorem 1 | |
| Sampling forward process | Direct | Geodesic Random Walk (Algorithm 1) | |
| Sampling backward process | Euler–Maruyama | Geodesic Random Walk (Algorithm 1) | |

$\{\mathbf{s}_t \in \mathcal{X}(\mathcal{M}) : t \in [0, t]\}$. If this is not the case then we turn to $\ell_t^{\text{im}}$. Note that $\ell_t^{\text{im}}$ requires the computation of a divergence term which requires $d$ Jacobian-vector calls. In high dimension, a stochastic estimator is necessary (Hutchinson, 1989). Following Song and Ermon (2020); Nichol and Dhariwal (2021) the loss can be weighted with a term $\lambda_t > 0$.

**Parametric family of vector fields.** We approximate $(\nabla \log p_t)_{t \in [0, T]}$ by a family of functions $\{\mathbf{s}_\theta\}_{\theta \in \Theta}$ where $\Theta$ is a set of parameters and $\mathbf{s}_\theta : [0, T] \to \mathcal{X}(\mathcal{M})$. In a Euclidean space, vector fields are simply functions $\mathbf{s}_\theta : \mathbb{R}^d \to \mathbb{R}^d$. In manifolds, although for any $x \in \mathcal{M}$, $\mathrm{T}_x\mathcal{M} \cong \mathbb{R}^d$, there does not necessarily exist a set of $d$ smooth vector fields $\{E_i\}_{i=1}^d$ such that $\text{span}\left(\{E_i(x)\}_{i=1}^d\right) = \mathrm{T}_x\mathcal{M}$ (Chapter 8, page 179, Lee, 2006) [9]. Fortunately, one can rely on a larger set of smooth vector fields $\{E_i(x)\}_{i=1}^n$ with $n > d$ that *does* span the tangent bundle. Then it suffices to construct a neural network $\mathbf{s}_\theta : [0, T] \times \mathcal{M} \to \mathbb{R}^n$ to parametrize the score network as $\mathbf{s}_\theta(t, x) = \sum_{i=1}^n \mathbf{s}_\theta^i(t, x) E_i(x)$. See App. E for a discussion on the different choices of generating sets $\{E_i(x)\}_{i=1}^n$.

Combining this parameterization with the score matching losses, the time-reversal formula of Theorem 1 and the sampling of forward and backward processes described in Sec. 3.3, we define our RGSM algorithm in Algorithm 2. This algorithm can also benefit from a predictor-corrector scheme as in (Song et al., 2021), see App. G.

## 4    RSGMs on compact manifolds

Assuming compactness of the manifold $\mathcal{M}$, we can leverage a number of special properties to implement a specific case of our algorithm. In particular we benefit from the fact that on compact manifolds we have a proper *uniform* distribution over the manifold, and have access to a variety of approximations of the heat kernel. As highlighted in Sec. 3.1, in the compact setting we use Brownian motion as the noising SDE, which targets the uniform distribution as the stationary distribution. Table 1 highlights the main differences between RSGMs on compact manifolds, generic manifolds and Euclidean score-based models.

---

[9]Manifolds for which there exists such a *global frame* $\{E_i(x)\}_{i=1}^d$ are referred as *parallelizable*. $\mathbb{S}^2$ is a well-known example of *non-parallelizable* manifold as per the *Hairy ball theorem*.

**Heat kernel on compact Riemannian manifolds.** For any $x_0 \in \mathcal{M}$ and $t \geq s \geq 0$, the heat kernel $p_{t|s}(\cdot|x_s)$ is defined as the density of $\mathbf{B}_t^{\mathcal{M}}$ w.r.t. the uniform measure on the manifold.

Contrary to the Gaussian transition density of the OU process (or the Brownian motion) in the Euclidean setting, it is typically only available as an infinite series. In order to circumvent this issue we consider two techniques: i) a truncation approach, ii) a Taylor expansion around $t = 0$ called a Varadhan asymptotics. First, we recall that in the case of compact manifolds the heat kernel is given by the Sturm–Liouville decomposition (Chavel, 1984) given for any $t > 0$ and $x_0, x_t \in \mathcal{M}$ by

$$p_{t|0}(x_t|x_0) = \sum_{j \in \mathbb{N}} e^{-\lambda_j t} \phi_j(x_0) \phi_j(x_t), \tag{6}$$

where the convergence occurs in $\mathrm{L}^2(p_{\text{ref}} \otimes p_{\text{ref}})$, $(\lambda_j)_{j \in \mathbb{N}}$ and $(\phi_j)_{j \in \mathbb{N}}$ are the eigenvalues, respectively the eigenvectors, of $-\Delta_{\mathcal{M}}$, the Laplace-Beltrami operator in the manifold, in $\mathrm{L}^2(p_{\text{ref}})$ (see Saloff-Coste, 1994, Section 2). When the eigenvalues and eigenvectors are known, we rely on an approximation of the logarithmic gradient of $p_{t|0}$ by truncating the sum in Eq. (S8) with $J \in \mathbb{N}$ terms to obtain for any $t > 0$ and $x_0, x_t \in \mathcal{M}$

$$\nabla_{x_t} \log p_{t|0}(x_t|x_0) \approx S_{J,t}(x_0, x_t) \triangleq \nabla_{x_t} \log \sum_{j=0}^{J} e^{-\lambda_j t} \phi_j(x_0) \phi_j(x_t). \tag{7}$$

Under regularity conditions on $\mathcal{M}$ it can be shown that for any $x, y \in \mathcal{M}$ and $t \geq 0$, $\lim_{J \to +\infty} S_{J,t}(x_0, x_t) = \nabla_{x_t} \log p_{t|0}(x_t|x_0)$ (see Jones et al., 2008, Lemma 1). In the case of the $d$-dimensional torus or sphere the eigenvalues and eigenvectors are computable (see Saloff-Coste, 1994, Section 2) and we can apply this method to approximate $p_{t|0}$ for any $t > 0$, see App. F

When the eigenvalues and eigenvectors are unknown or not tractable, we can still derive an approximation of the heat kernel for small times $t$. Using Varadhan's asymptotics—see Bismut (1984, Theorem 3.8) or Chen et al. (2021, Theorem 2.1)—for any $x, y \in \mathcal{M}$ with $y \notin \mathrm{Cut}(x)$ (where $\mathrm{Cut}(x)$ is the cut-locus of $x$ in $\mathcal{M}$ (see Lee, 2018, Chapter 10)) we have that

$$\lim_{t \to 0} t \nabla_{x_t} \log p_{t|0}(x_t|x_0) = \exp_{x_t}^{-1}(x_0). \tag{8}$$

Using the previously defined score-matching losses and the approximations to the heat kernel above, we highlight three methods to compute $\nabla \log p_t$ in Table 2.

Table 2: Computational complexity of score matching losses w.r.t. score network forward and backward passes. $\varepsilon$ is a random variable on $\mathrm{T}_{\mathbf{X}_t} \mathcal{M}$ such that $\mathbb{E}[\varepsilon] = 0$ and $\mathbb{E}[\varepsilon \varepsilon^\top] = \mathrm{Id}$.

| Loss | Approximation | Loss function | Requirements $p_{t\|0}$ | $\exp_{\mathbf{X}_t}^{-1}$ | Complexity |
|---|---|---|---|---|---|
| $\ell_{t\|0}$ (DSM) | None | $\frac{1}{2}\mathbb{E}\left[\|\mathbf{s}(\mathbf{X}_t) - \nabla \log p_{t\|0}(\mathbf{X}_t\|\mathbf{X}_0)\|^2\right]$ | ✓ | ✗ | $\mathcal{O}(1)$ |
| | Truncation (7) | $\frac{1}{2}\mathbb{E}\left[\|\mathbf{s}(\mathbf{X}_t) - S_{J,t}(\mathbf{X}_0, \mathbf{X}_t)\|^2\right]$ | asymptotic expansion | ✗ | $\mathcal{O}(1)$ |
| | Varhadan (8) | $\frac{1}{2}\mathbb{E}\left[\|\mathbf{s}(\mathbf{X}_t) - \exp_{\mathbf{X}_t}^{-1}(\mathbf{X}_0)/t\|^2\right]$ | ✗ | ✓ | $\mathcal{O}(1)$ |
| $\ell_{t\|s}$ (DSM) | Varhadan (8) | $\frac{1}{2}\mathbb{E}\left[\|\mathbf{s}(\mathbf{X}_t) - \exp_{\mathbf{X}_t}^{-1}(\mathbf{X}_s)/(t-s)\|^2\right]$ | ✗ | ✓ | $\mathcal{O}(1)$ |
| $\ell_t^{\text{im}}$ (ISM) | Deterministic | $\mathbb{E}\left[\frac{1}{2}\|\mathbf{s}(\mathbf{X}_t)\|^2 + \mathrm{div}(\mathbf{s})(\mathbf{X}_t)\right]$ | ✗ | ✗ | $\mathcal{O}(d)$ |
| | Stochastic | $\mathbb{E}\left[\frac{1}{2}\|\mathbf{s}(\mathbf{X}_t)\|^2 + \varepsilon^\top \partial \mathbf{s}(\mathbf{X}_t)\varepsilon\right]$ | ✗ | ✗ | $\mathcal{O}(1)$ |

**Convergence results in the compact setting** We now provide a theoretical analysis of RSGM under the assumption that $\mathcal{M}$ is compact. The following result ensures that RSGM generates samples whose distribution is close to the data distribution $p_0$. Let us denote $\{Y_k\}_{n \in \{0,...,N\}}$ the sequence generated by Algorithm 2. This result relies on the following assumption, which is satisfied for a large class of manifolds $\mathcal{M}$ such as the $d$-dimensional sphere and torus, compact matrix groups and products of these manifolds.

**A1.** *There exist $C, \alpha > 0$ such that for any $t \in (0,1]$ and $x \in \mathcal{M}$, $p_{t|0}(x|x) \leq Ct^{-\alpha/2}$, where $p_{t|0}(\cdot|x_0)$ is the density of the heat kernel, i.e. the density of $\mathbf{B}_t^{\mathcal{M}}$ with initial condition $x_0$* [10].

---

[10] The diagonal upper-bound is implied by Sobolev inequalities which control of the growth of some functions by the growth of their gradient. **A1** is satisfied in our experiments, see Saloff-Coste (1994); Gross (1992).

**Theorem 4.** *Assume* **A**1*, that $p_0$ is smooth and positive and that there exists* $\mathtt{M} \geq 0$ *such that for any* $t \in [0, T]$ *and* $x \in \mathcal{M}$, $\|\mathbf{s}_{\theta^\star}(t, x) - \nabla \log p_t(x)\| \leq \mathtt{M}$, *with* $\mathbf{s}_{\theta^\star} \in \mathrm{C}([0, T], \mathcal{X}(\mathcal{M}))$. *Then if* $T > 1/2$, *there exists* $C \geq 0$ *independent on* $T$ *such that*

$$\mathbf{W}_1(\mathcal{L}(Y_N), p_0) = C(\mathrm{e}^{-\lambda_1 T} + \sqrt{T/2}\mathtt{M} + \mathrm{e}^T \gamma^{1/2}),$$

*where* $\mathbf{W}_1$ *is the Wasserstein distance of order one on the probability measures on* $\mathcal{M}$.

The proof is postponed to App. I. In particular, for any $\varepsilon > 0$, choosing $T > 0$ large enough, $\mathtt{M}$ small enough (which can be achieved using the universal property of neural networks) and $\gamma$ small enough, we get that $\mathbf{W}_1(\mathcal{L}(Y_N), p_0) \leq \varepsilon$. This result might seem weaker than the result obtained for Moser flows in (Rozen et al., 2021, Theorem 3), but we emphasize that our bound takes into account the time-discretization contrary to Rozen et al. (2021) which considers the continuous-time flow. If we consider the time-reversed continuous-time SDE then we recover a bound in total variation distance, see App. I. Note that the upper bound $\mathtt{M}$ encompasses both the bias introduced by the use of a neural network and the bias introduced by the use of an approximation of the score.

# 5 Related work

In this section we discuss previous work on parametrizing family of distributions for manifold-valued data. Here, the manifold structure is considered to be prescribed, in contrast with methods that jointly learn the manifold structure and density (e.g. Brehmer and Cranmer, 2020; Caterini et al., 2021).

**Parametric family of distributions.** The various parametric families of manifold-valued distributions that have been proposed can be categorized into three main approaches (Navarro et al., 2017): wrapping, projecting and conditioning. Wrapped distributions consider a parametric distribution on $\mathbb{R}^n$ that is pushed-forward along a surjective map $\psi : \mathbb{R}^n \to \mathcal{M}$. Projected distributions are defined by marginalizing out some distribution along the normal bundle of $\mathcal{M}$. Conditioning distributions encompass the von Mises-Fisher and Kent distributions (Fisher, 1953; Kent, 1982). Considering mixtures of these distributions is key to increase flexibility (Peel et al., 2001; Mardia et al., 2008).

**Push-forward of Euclidean normalizing flows.** More recently, approaches leveraging the flexibility of normalizing flows (Papamakarios et al., 2019) have been proposed. Following the wrapping method described above, these methods parametrize a normalizing flow in $\mathbb{R}^n$ before being pushed along an invertible map $\psi : \mathbb{R}^n \to \mathcal{M}$. However, to globally represent the manifold, the map $\psi$ needs to be a homeomorphism, which can only happen if $\mathcal{M}$ is topologically equivalent to $\mathbb{R}^n$, hence limiting the scope of that approach. One natural choice for this map is the exponential map $\exp_x : \mathrm{T}_x\mathcal{M} \cong \mathbb{R}^d$. This approach has been taken, for instance, by Falorsi et al. (2019) and Bose et al. (2020), respectively parametrizing distributions on Lie groups and hyperbolic space.

**Neural ODE on manifolds.** To avoid artifacts or numerical instabilities due to the manifold embedding, another line of work uses tools from Riemannian geometry to define flows directly on the manifold of interest (Falorsi and Forré, 2020; Mathieu and Nickel, 2020; Falorsi, 2021). Since these methods do not require a specific embedding mapping, they are referred as *Riemannian*. They extend continuous normalizing flows (CNFs) (Grathwohl et al., 2019) to the manifold setting, by implicity parametrizing flows as solutions of Ordinary Differential Equations (ODEs). As such, the parametric flow is a *continuous* function of time. This approach has recently been extended by Rozen et al. (2021) introducing Moser flows, whose main appeal being that it circumvents the need to solve an ODE in the training process. We refer to App. K for an in-depth discussion on the links between our work and Moser flows.

**Optimal transport on manifolds.** Another line of work has developed flows on manifolds using tools from optimal transport. Sei (2013) introduced a flow that is given by $f_\theta : x \mapsto \exp_x(\nabla \psi_\theta^c)$ with $\psi_\theta^c$ a $c$-convex function and $c = d_\mathcal{M}^2$ the squared geodesic distance. This approach is motivated by the fact that the optimal transport map takes such an expression (Ambrosio, 2003). These methods operate directly on the manifold, similarly to CNFs, yet in contrast they are *discrete* in time. The benefits of this approach depend on the specific choice of parametric family of $c$-convex functions (Rezende and Racanière, 2021; Cohen et al., 2021), trading-off expressivity with scalability.

Table 3: Summary of computational complexity (w.r.t. neural network forward and backward passes) for different methods. $d$ is the manifold dimension, $k$ the number of Monte Carlo batches in Moser flow's regularizer, $N$ is the number of steps in the (adaptive) ODE solver, whereas $N^*$ is the number of steps in the SDE Euler-Maruyama solver–which can usually be lower than $N$. Moser flow and RSGM training complexity varies if the Hutchinson stochastic estimator is used. See Table 2 for score matching losses complexity.

| Method | Training | Likelihood evaluation | Sampling |
|---|---|---|---|
| RCNF | Solving ODE $\mathcal{O}(dN)$ | Solving augmented ODE $\mathcal{O}(dN)$ | Solving ODE $\mathcal{O}(N)$ |
| Moser flow | Computing div $\mathcal{O}(dk)$ or $\mathcal{O}(k)$ | Solving augmented ODE $\mathcal{O}(dN)$ | Solving ODE $\mathcal{O}(N)$ |
| RSGM | Score matching $\mathcal{O}(d)$ or $\mathcal{O}(1)$ | Solving augmented ODE $\mathcal{O}(dN)$ | Solving SDE $\mathcal{O}(N^*)$ |

## 6 Experiments

In this section we benchmark the empirical performance of RSGMs along with other manifold-valued methods introduced in Sec. 5. We also compare to a 'Stereographic' score-based model, introduced in App. N. First, we assess their modelling capacity on earth and climate science spherical data. Then, we test the methods scalability with respect to manifold dimensions with a synthetic experiment on the torus $\mathbb{T}^d$. Eventually, we evaluate the models' regularity and time complexity with a synthetic $SO_3(\mathbb{R})$ target. Experimental details are provided in App. O.

### 6.1 Earth and climate science datasets on the sphere

We start by evaluating RSGMs on a collection of simple datasets, each containing an empirical distribution of occurrences of earth and climate science events on the surface of the earth. These events are: volcanic eruptions (NGDC/WDS), earthquakes (NGDC/WDS), floods (Brakenridge, 2017) and wild fires (EOSDIS, 2020). We compare to previous baseline methods: Riemannian Continuous Normalizing Flows (Mathieu and Nickel, 2020), Moser Flows (Rozen et al., 2021) and a mixture of Kent distributions (Peel et al., 2001). Additionally, we consider a standard SGM on the 2D plane followed by the inverse stereographic projection which induces a density on the sphere (Gemici et al., 2016). We evaluate the log-likelihood of each model, extending to the manifold setting the likelihood computation techniques of SGMs, see App. D. We observe from Table 4, that all benchmarked methods have comparable performance when evaluated on these simple tasks with RSGM performing marginally better on most datasets. However, we empirically notice that Moser flows are slow to train and additionally that both Moser flows and stereographic SGMs are computationally expensive to evaluate.

### 6.2 Synthetic data on tori

We now move to another manifold, that is the torus $\mathbb{T}^d = \mathbb{S}^1 \times \cdots \times \mathbb{S}^1$, so as to assess the scalability of the different methods with respect to the dimension $d$. We consider a wrapped Gaussian target distribution on $\mathbb{T}^d$ with a random mean and unit variance. Moser flows' (Rozen et al., 2021) loss involves a regularization term which involves an integral over the manifold, approximated by a Monte Carlo (MC) estimator with uniform proposal. This term regularizes Moser flows towards probability measures, i.e. with unit volume. We thus expect Moser flows to fail in high-dimension as the number of samples $K$ required for the MC estimator to be accurate will grows as $\mathcal{O}(e^d)$, and the

Table 4: Negative log-likelihood scores for each method on the earth and climate science datasets. Bold indicates best results (up to statistical significance). Means and confidence intervals are computed over 5 different runs. Novel methods are shown with blue shading.

| Method | Volcano | Earthquake | Flood | Fire |
|---|---|---|---|---|
| Mixture of Kent | $-0.80_{\pm 0.47}$ | $0.33_{\pm 0.05}$ | $0.73_{\pm 0.07}$ | $-1.18_{\pm 0.06}$ |
| Riemannian CNF | $\mathbf{-6.05_{\pm 0.61}}$ | $0.14_{\pm 0.23}$ | $1.11_{\pm 0.19}$ | $\mathbf{-0.80_{\pm 0.54}}$ |
| Moser Flow | $-4.21_{\pm 0.17}$ | $\mathbf{-0.16_{\pm 0.06}}$ | $\mathbf{0.57_{\pm 0.10}}$ | $\mathbf{-1.28_{\pm 0.05}}$ |
| Stereographic Score-Based | $-3.80_{\pm 0.27}$ | $\mathbf{-0.19_{\pm 0.05}}$ | $\mathbf{0.59_{\pm 0.07}}$ | $\mathbf{-1.28_{\pm 0.12}}$ |
| Riemannian Score-Based | $-4.92_{\pm 0.25}$ | $\mathbf{-0.19_{\pm 0.07}}$ | $\mathbf{0.45_{\pm 0.17}}$ | $\mathbf{-1.33_{\pm 0.06}}$ |
| Dataset size | 827 | 6120 | 4875 | 12809 |

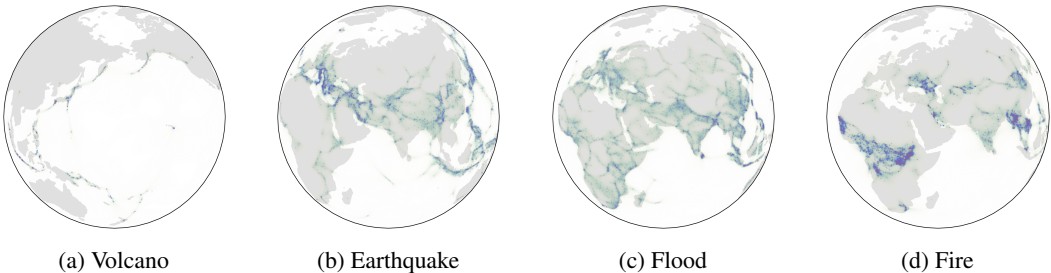

| (a) Volcano | (b) Earthquake | (c) Flood | (d) Fire |

Figure 2: Trained score-based generative models on earth sciences data. The learned density is colored green-blue. Blue and red dots represent training and testing datapoints, respectively.

memory required to compute this estimator grows either in $\mathcal{O}(Kd)$ for exact divergences or $\mathcal{O}(K)$ for approximated divergences (see Table 3).

In Fig. 3, we observe that RSGMs are able to fit well the target distribution even in high dimension, with a linear or constant computational cost—depending on the divergence estimator. In contrast, Moser flows scale poorly with the dimension, to the extent that we are unable to train them for $d \geq 10$. This is due to the combination of the complexity which grows linearly with both the dimension $d$ and the number of MC samples $K$, which itself ought to grow exponentially with $d$—as discussed in the previous paragraph. This is illustrated by the gap between the 'Moser' and 'ODE' likelihoods which increases with the manifold dimension (see left Fig. 3).

## 6.3 Synthetic data on the Special Orthogonal group

In order to demonstrate the broad range of applicability of our model we now turn to the task of density estimation on the special orthogonal group $\mathrm{SO}_d(\mathbb{R}) = \{Q \in \mathrm{M}_d(\mathbb{R}) : QQ^\top = \mathrm{Id}, \det(Q) = 1\}$. We consider the synthetic dataset consisting of samples in $\mathrm{SO}_3(\mathbb{R})$ from a mixture of wrapped normal distributions with $M$ components.

We compare RSGMs against Moser flows and a wrapped-exponential baseline inspired by Falorsi et al. (2019)—where we parametrize a standard Euclidean SGM on $\mathfrak{so}(3)$ that is then pushed-forward on $\mathrm{SO}_3(\mathbb{R})$. RSGMs are trained using the $\ell_{t|0}$ (DSM) loss with the Varadhan approximation (see Table 2). From Table 5 we observe that, RSGMs perform consistently, whether the target distribution has few or many mixture components $M$, as opposed to Exp-wrapped SGMs and Moser flows which only perform well in some range of $M$. Similarly to Sec. 6.2, we find Moser flows to be much slower to train due to the large number of Monte Carlo samples needed in the regularizer ($K = 10^4$). We also note from Table 5 that the number of score network evaluations (NFE) is significantly lower for RSGMs, and is particularly detrimental for Moser flows ($\gg 10^3$).

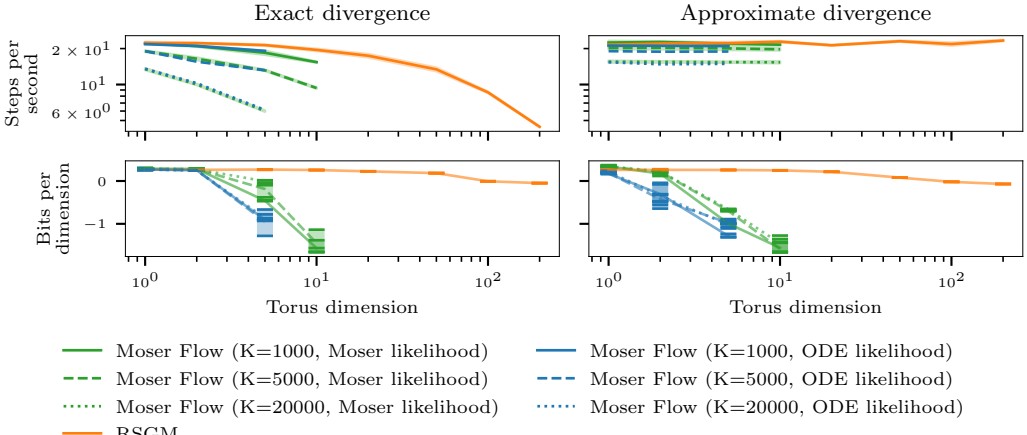

Figure 3: Comparison of Moser flows and RSGMs training speed and performance on the synthetic high-dimension torus task. Moser flows trained with $\lambda_{\min} = 1$. We report two likelihoods, the 'Moser' closed form density—not guaranteed to be normalized—and the 'ODE' likelihood given by solving an augmented ODE (as in CNFs) with the vector field induced by the Moser flow density—which is guaranteed to have unit volume.

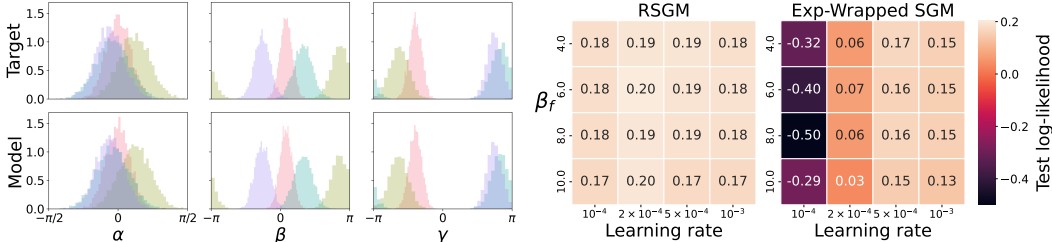

(a) Histograms of $SO_3(\mathbb{R})$ samples from a target mixture distribution with $M = 4$ components, represented via their Euler angles.

(b) RSGMs are much more robust to hyperparameters than Exp-wrapped SGMs. The diffusion coefficient is given by $\sigma(t, \mathbf{X}_t) = \sqrt{\beta(t)}$, $\beta(t) = \beta_0 + (\beta_f - \beta_0)t$.

Figure 4: Trained score-based generative models on synthetic $SO_3(\mathbb{R})$ data.

Table 5: Test log-likelihood and associated number of function evaluations (NFE) in $10^3$ on the synthetic mixture distribution with $M$ components on $SO_3(\mathbb{R})$. Bold indicates best results (up to statistical significance). Means and standard deviations are computed over 5 different runs. Novel methods are shown with blue shading.

| Method | $M = 16$ | | $M = 32$ | | $M = 64$ | |
|---|---|---|---|---|---|---|
| | log-likelihood | NFE | log-likelihood | NFE | log-likelihood | NFE |
| Moser Flow | $0.85_{\pm 0.03}$ | $2.3_{\pm 0.5}$ | $0.17_{\pm 0.03}$ | $2.3_{\pm 0.9}$ | $-0.49_{\pm 0.02}$ | $7.3_{\pm 1.4}$ |
| Exp-wrapped SGM | $\mathbf{0.87_{\pm 0.04}}$ | $0.5_{\pm 0.1}$ | $0.16_{\pm 0.03}$ | $0.5_{\pm 0.0}$ | $-0.58_{\pm 0.04}$ | $0.5_{\pm 0.0}$ |
| RSGM | $\mathbf{0.89_{\pm 0.03}}$ | $\mathbf{0.1_{\pm 0.0}}$ | $\mathbf{0.20_{\pm 0.03}}$ | $\mathbf{0.1_{\pm 0.0}}$ | $\mathbf{-0.49_{\pm 0.02}}$ | $\mathbf{0.1_{\pm 0.0}}$ |

## 6.4 Synthetic data on hyperbolic space

Finally we demonstrate RSGM on a non-compact manifold: the two dimensional hyperbolic space $\mathbb{H}^2$, which is defined as the simply connected space of constant negative curvature. We use Langevin dynamics as the noising process (Eq. (3)) and target a wrapped Gaussian as the invariant distribution. We again consider a synthetic dataset of samples from a mixture of exp-wrapped normal distribution. From Fig. 5, we can qualitatively see that both score-based models are able to fit the target distribution.

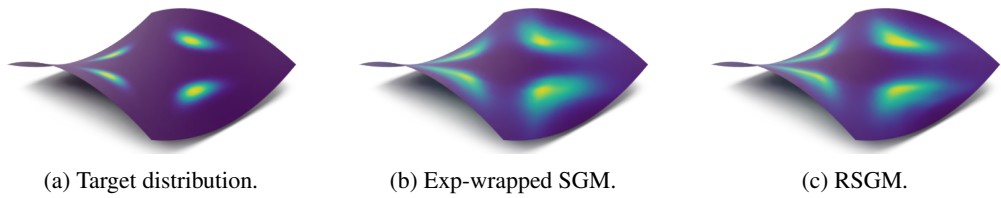

(a) Target distribution.          (b) Exp-wrapped SGM.          (c) RSGM.

Figure 5: Samples from different probability distributions on $\mathbb{H}^2$ coloured w.r.t their density.

## 7 Discussion and limitations

In this paper we introduced Riemannian Score-Based Generative Models (RSGMs), a class of deep generative models that represent target densities supported on manifolds, as the time-reversal of Langevin dynamics. The main benefits of our method stems from its scalability to high dimensions, its applicability to a broad class of manifolds due to the diversity of available loss functions, its robustness and crucially its capacity to model complex datasets. We also provided theoretical guarantees on the convergence of RSGMs. In future work, we would like explore more generic classes of manifolds, such a ones with a boundary, along with alternative noising processes. Another promising extension concerns stochastic control on manifolds and more precisely, deriving efficient algorithms to solve Schrödinger bridges in the same spirit as De Bortoli et al. (2021) on Euclidean state spaces.

## Acknowledgments

We are grateful to the anonymous reviewers for their insightful comments and the for fruitful discussion more generally. We thank the `geomstat` team Miolane et al. (2020a). and Engineering and Physical Research Council (EPSRC) under grant EP/R013616/1. This is part of the collaboration between US DOD, UK MOD and UK EPSRC under the Multidisciplinary University Research Initiative. AD is also partially supported by the EPSRC grant EP/R034710/1 CoSines.

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
