# Supplementary to:
# Riemannian Score-Based Generative Modelling

## A  Organization of the supplementary

In this supplementary we first introduce notation in App. B. We gather the proof of Theorem 1 as well as additional derivations on score-based generative models and Riemannian manifolds. In App. C, we recall basics on stochastic Riemannian geometry following Hsu (2002). In App. D, we introduce an extension to the Riemannian setting of the likelihood computation techniques in diffusion models. Details about parametric vector fields are given in App. E. In App. F, we recall some basic facts about eigenvalues and eigenfunctions of the Laplace–Beltrami operator on the $d$-dimensional sphere and torus. We present an extension of Algorithm 2 using predictor-corrector schemes in App. G. In App. H, we prove the extension of the time-reversal formula to manifold in Theorem 1. We prove the convergence of RSGM, i.e. Theorem 4, in App. I. The proof of Proposition 3 drawing links between the denoising score matching loss and the implicit score matching loss is presented App. J. We provide a thorough comparison between our approach and the one of Rozen et al. (2021) in App. K. We show how our method can be adapted to perform density estimation in App. L. Extensions to conditional SGM and Schrödinger Bridges are discussed in App. M. In Sec. 3.1, we briefly discuss the non compact setting. Details on the stereographic SGM are given in App. N. Experimental details are given in App. O.

## B  Notation

We refer to App. C for more details about the basic concepts of Riemannian geometry and stochastic processes. In this section, we merely introduce the notation used in our work. We postpone an introduction to stochastic processes on manifolds to App. C.2.

In this work we always consider a smooth, connected and complete manifold $\mathcal{M}$. We focus on the case of Riemannian manifolds, namely manifolds equipped with a metric $g$. Metrics $g$ are smooth scalar product on the manifold allowing us to define the notion of *distance* on a manifold. We refer to App. C for a precise definition and a discussion on metrics. Given a smooth map $f \in \mathrm{C}^\infty(\mathcal{M}, \mathbb{R})$, the gradient $\nabla f$ is defined for any $f : \mathcal{M} \to \mathbb{R}$, $x \in \mathcal{M}, v \in \mathrm{T}_x\mathcal{M}$, $\langle \nabla f, v \rangle_g = \mathrm{d}f(v)$. The distane $d_\mathcal{M}(x, y)$ is defined as the infimum of the length of all the curves on $\mathcal{M}$ joining $x$ and $y$. Geodesics are path defined on $\mathcal{M}$ by a second order equation (and a starting point and speed). This second order equation corresponds to the first order minimization of an *energy* functional whose minimizers also minimize the length. In App. C, we introduce the notion of geodesics using parallel transport. The exponential mapping $\exp_x : \mathsf{U}\mathcal{M} \to \mathcal{M}$ with $\mathsf{U} \subset \mathrm{T}_x\mathcal{M}$ is such that $\exp_x(v) = \gamma(1)$ with $\gamma(1)$ the geodesics with initial condition $(x, v)$ at time $t = 1$. Finally the volume form is a differentiable form of same degree as the dimension of $\mathcal{M}$. Since $\mathcal{M}$ is an orientable Riemannian manifold there is a natural volume form defined using the metric $g$, namely $\omega(x) = |g(x)|^{1/2} \, \mathrm{d}x_1 \wedge \ldots \mathrm{d}x_d$. In this paper, we abuse notation and denote by the volume form this natural volume form.

## C  Preliminaries on stochastic Riemannian geometry

In this section, we recall some basic facts on Riemannian geometry and stochastic Riemannian geometry. We follow Hsu (2002); Lee (2018, 2006) and refer to Lee (2010, 2013) for a general introduction to topological and smooth manifolds. Throughout this section $\mathcal{M}$ is a $d$-dimensional smooth manifold, $\mathrm{T}\mathcal{M}$ its tangent bundle and $\mathrm{T}^\star\mathcal{M}$ it cotangent bundle. We denote $\mathrm{C}^\infty(\mathcal{M})$ the set of real-valued smooth functions on $\mathcal{M}$ and $\mathcal{X}(\mathcal{M})$ the set of vector fields on $\mathcal{M}$.

### C.1  Tensor field, metric, connection and transport

**Tensor field and Riemannian metric**  For a vector space $V$ let $\mathrm{T}^{k,\ell}(V) = V^{\otimes k} \otimes (V^\star)^{\otimes \ell}$ with $k, \ell \in \mathbb{N}$. For any $k, \ell \in \mathbb{N}$ we define the space of $(k, \ell)$-tensors as $\mathrm{T}^{k,\ell}\mathcal{M} = \sqcup_{p \in \mathcal{M}} \mathrm{T}^{k,\ell}(\mathrm{T}_p\mathcal{M})$.

Note that $\Gamma(\mathcal{M}, \mathrm{T}^{0,0}\mathcal{M}) = \mathrm{C}^\infty(\mathcal{M})$, $\mathcal{X}(\mathcal{M}) = \Gamma(\mathcal{M}, \mathrm{T}^{1,0}\mathcal{M})$ and that the space of 1-form on $\mathcal{M}$ is given by $\Gamma(\mathcal{M}, \mathrm{T}^{0,1}\mathcal{M})$, where $\Gamma(\mathcal{M}, V(\mathcal{M}))$ is a section of a vector bundle $V(\mathcal{M})$ (see Lee, 2013, Chapter 10). For any $k \in \mathbb{N}$, we denote $\mathrm{T}^{|k|}\mathcal{M} = \sqcup_{j=0}^{k}\mathrm{T}^{j,k-j}\mathcal{M}$. $\mathcal{M}$ is said to be a Riemannian manifold if there exists $g \in \Gamma(\mathcal{M}, \mathrm{T}^{0,2}\mathcal{M})$ such that for any $x \in \mathcal{M}$, $g(x)$ is positive definite. $g$ is called the Riemannian metric of $\mathcal{M}$. Every smooth manifold can be equipped with a Riemannian metric (see Lee, 2018, Proposition 2.4). In local coordinates we define $G = \{g_{i,j}\}_{1 \le i,j \le d} = \{g(X_i, X_j)\}_{1 \le i,j \le d}$, where $\{X_i\}_{i=1}^{d}$ is a basis of the tangent space. In what follows we consider that $\mathcal{M}$ is equipped with a metric $g$ and for any $X, Y \in \mathcal{X}(\mathcal{M})$ we denote $\langle X, Y \rangle_\mathcal{M} = g(X, Y)$.

**Connection**   A connection $\nabla$ is a mapping which allows one to differentiate vector fields w.r.t other vector fields. $\nabla$ is a linear map $\nabla : \mathcal{X}(\mathcal{M}) \times \mathcal{X}(\mathcal{M}) \to \mathcal{X}(\mathcal{M})$. In addition, we assume that i) for any $f \in \mathrm{C}^\infty(\mathcal{M})$, $X, Y \in \mathcal{X}(\mathcal{M})$, $\nabla_{fX}(Y) = f\nabla_X Y$, ii) for any $f \in \mathrm{C}^\infty(\mathcal{M})$, $X, Y \in \mathcal{X}(\mathcal{M})$, $\nabla_X(fY) = f\nabla_X Y + X(f)Y$. Given a system of local coordinates, the Christoffel symbols $\{\Gamma_{i,j}^k\}_{1 \le i,j,k \le d}$ are given for any $i, j \in \{1, \ldots, d\}$ by $\nabla_{X_i} X_j = \sum_{k=1}^{d} \Gamma_{i,j}^k X_k$. We also define the Levi–Civita connection $\nabla$ by considering the additional two conditions: i) $\nabla$ is torsion-free, i.e. for any $X, Y \in \mathcal{X}(\mathcal{M})$ we have $\nabla_X Y - \nabla_Y X = [X, Y]$, where $[X, Y]$ is the Lie bracket between $X$ and $Y$, ii) $\nabla$ is compatible with the metric $g$, i.e. for any $X, Y, Z \in \mathcal{X}(\mathcal{M})$, $X(\langle Y, Z \rangle_\mathcal{M}) = \langle \nabla_X Y, Z \rangle_\mathcal{M} + \langle Y, \nabla_X Z \rangle_\mathcal{M}$. We recall that the Levi–Civita connection is uniquely defined since for any $X, Y, Z \in \mathcal{X}(\mathcal{M})$ we have

$$2\langle \nabla_X Y, Z \rangle_\mathcal{M} = X(\langle Y, Z \rangle_\mathcal{M}) + Y(\langle Z, X \rangle_\mathcal{M}) - Z(\langle X, Y \rangle_\mathcal{M})$$
$$+ \langle [X, Y], Z \rangle_\mathcal{M} - \langle [Z, X], Y \rangle_\mathcal{M} - \langle [Y, Z], X \rangle_\mathcal{M}.$$

In this case, the Christoffel symbols are given for any $i, j, k \in \{1, \ldots, d\}$ by

$$\Gamma_{i,j}^k = \tfrac{1}{2} \sum_{m=1}^{d} g^{km}(\partial_j g_{m,i} + \partial_i g_{m,j} - \partial_m g_{i,j}),$$

where $\{g^{i,j}\}_{1 \le i,j \le d} = G^{-1}$. Note that if $\mathcal{M}$ is Euclidean then for any $i, j, k \in \{1, \ldots, d\}$, $\Gamma_{i,j}^k = 0$. We also extend the connection so that for any $X \in \mathcal{X}(\mathcal{M})$ and $f \in \mathrm{C}^\infty(M)$ we have $\nabla_X f = X(f)$. In particular, we have that $\nabla_X f \in \mathrm{C}^\infty(\mathcal{M})$. In addition, we extend the connection such that for any $\alpha \in \Gamma(\mathcal{M}, \mathrm{T}^{0,1}\mathcal{M})$, $X, Y \in \mathcal{X}(\mathcal{M})$ we have $\nabla_X \alpha(Y) = \alpha(\nabla_X Y) - X(\alpha(Y))$. In particular, we have that $\nabla_X \alpha \in \Gamma(\mathcal{M}, \mathrm{T}^{1,0}\mathcal{M})$. Note that for any $X \in \mathcal{X}(\mathcal{M})$ and $\alpha, \beta \in \mathrm{T}^{|1|}\mathcal{M}$ we have $\nabla_X(\alpha \otimes \beta) = \nabla_X \alpha \otimes \beta + \alpha \otimes \nabla_X \beta$. Similarly, we can define recursively $\nabla_X \alpha$ for any $\alpha \in \Gamma(\mathcal{M}, \mathrm{T}^{k,\ell}\mathcal{M})$ with $k, \ell \in \mathbb{N}$. Such an extension is called a covariant derivative.

**Parallel transport, geodesics and exponential mapping**   Given a connection, we can define the notion of parallel transport, which transports vector fields along a curve. Let $\gamma : [0, 1] \to \mathcal{M}$ be a smooth curve. We define the covariant derivative along the curve $\gamma$ by $D_{\dot\gamma} : \mathcal{X}(\gamma) \to \mathcal{X}(\gamma)$ similarly to the connection, where $\mathcal{X}(\gamma) = \Gamma(\gamma([0, 1]), \mathrm{T}\mathcal{M})$. In particular if $\dot\gamma$ and $X \in \mathcal{X}(\gamma)$ can be extended to $\mathcal{X}(\mathcal{M})$ then we define $D_{\dot\gamma}(X) = \nabla_{\dot\gamma} X \in \mathcal{X}(\mathcal{M})$. In what follows, we denote $D = \nabla$ for simplicity. We say that $X \in \mathcal{X}(\gamma)$ is parallel to $\gamma$ if for any $t \in [0, 1]$, $\nabla_{\dot\gamma} X(t) = 0$. In local coordinates, let $X \in \mathcal{X}(\gamma)$ be given for any $t \in [0, 1]$ by $X = \sum_{i=1}^{d} a_i(t)E_i(t)$ (assuming that $\gamma([0, 1])$ is entirely contained in a local chart), then we have that for any $t \in [0, 1]$ and $k \in \{1, \ldots, d\}$

$$\dot a_k(t) + \sum_{i,j=1}^{d} \Gamma_{i,j}^k(x(t))\dot x_i(t)a_j(t) = 0. \tag{S1}$$

A curve $\gamma$ on $\mathcal{M}$ is said to be a geodesics if $\dot\gamma$ is parallel to $\gamma$. Using Eq. (S1) we get that

$$\ddot x_k(t) + \sum_{i,j=1}^{d} \Gamma_{i,j}^k(x(t))\dot x_i(t)\dot x_j(t) = 0.$$

For more details on geodesics and parallel transport, we refer to Lee (2018, Chapter 4). In addition, we have that parallel transport provides a linear isomorphism between tangent spaces. Indeed, let $v \in \mathrm{T}_x \mathcal{M}$ and $\gamma : [0, 1] \to \mathcal{M}$ with $\gamma(0) = x$ a smooth curve. Then, there exists a unique vector field $X^v \in \mathcal{X}(\gamma)$ such that $X^v(x) = v$ and $X^v$ is parallel to $\gamma$. For any $t \in [0, 1]$, we denote $\Gamma_0^t : \mathrm{T}_x \mathcal{M} \to \mathrm{T}_{\gamma(t)}\mathcal{M}$ the linear isomorphism such that $\Gamma_0^t(v) = X^v(\gamma(t))$.

For any $x \in \mathcal{M}$ and $v \in \mathrm{T}_x \mathcal{M}$ we denote $\gamma^{x,v} : [0, \varepsilon^{x,v}]$ the geodesics (defined on the maximal interval $[0, \varepsilon^{x,v}]$) on $\mathcal{M}$ such that $\gamma(0) = x$ and $\dot\gamma(0) = v$. We denote $\mathsf{U}^x = \{v \in \mathrm{T}_x \mathcal{M} : \varepsilon^{x,v} \ge 1\}$. Note that $0 \in \mathsf{U}^x$. For any $x \in \mathcal{M}$, we define the exponential mapping $\exp_x : \mathsf{U}^x \to \mathcal{M}$ such

that for any $v \in \mathsf{U}^x$, $\exp_x(v) = \gamma^{x,v}(1)$. If for any $x \in \mathcal{M}$, $\mathsf{U}^x = \mathrm{T}_x\mathcal{M}$, the manifold is called *geodesically complete*. As any connected compact manifold is geodesically complete, there exists a geodesic between any two points $x, y \in \mathcal{M}$ (see Lee, 2018, Lemma 6.18). For any $x, y \in \mathcal{M}$, we denote $\mathrm{Geo}_{x,y}$ the sets of geodesics $\gamma$ such that $\gamma(0) = x$ and $\gamma(y) = 1$. For any $x, y \in \mathcal{M}$ we denote $\Gamma_x^y(\gamma) : \mathrm{T}_x\mathcal{M} \to \mathrm{T}_y\mathcal{M}$ the linear isomorphism such that for any $v \in \mathrm{T}_x\mathcal{M}$, $\Gamma_x^y(v) = X^v(\gamma(1))$, where $\gamma \in \mathrm{Geo}_{x,y}$. Note that for any $x \in \mathcal{M}$ there exists $\mathsf{V}^x \subset \mathcal{M}$ such that $x \in \mathsf{V}^x$ and for any $y \in \mathsf{V}^x$ we have that $|\mathrm{Geo}_{x,y}| = 1$. In this case, we denote $\Gamma_x^y = \Gamma_x^y(\gamma)$ with $\gamma \in \mathrm{Geo}_{x,y}$.

**Orthogonal projection**  We will make repeated use of orthonormal projections on manifolds. Recall that since $\mathcal{M}$ is a closed Riemannian manifold we can use the Nash embedding theorem (Gunther, 1991). In the rest of this paragraph, we assume that $\mathcal{M}$ is a Riemannian submanifold of $\mathbb{R}^p$ for some $p \in \mathbb{N}$ such that its metric is induced by the Euclidean metric. In order to define the projection we introduce

$$\mathrm{unpp}(\mathcal{M}) = \{x \in \mathbb{R}^d \ : \ \text{there exists a unique } \xi_x \text{ such that } \|x - \xi_x\| = d(x, \mathcal{M})\}.$$

Let $\mathcal{E}(\mathcal{M}) = \mathrm{int}(\mathrm{unpp}(\mathcal{M}))$. By Leobacher and Steinicke (2021, Theorem 1), we have $\mathcal{M} \subset \mathcal{E}(\mathcal{M})$. We define $\tilde{p} : \mathcal{E}(\mathcal{M}) \to \mathcal{M}$ such that for any $x \in \mathcal{E}(\mathcal{M})$, $\tilde{p}(x) = \xi_x$. Using Leobacher and Steinicke (2021, Theorem 2), we have $\tilde{p} \in \mathrm{C}^\infty(\mathbb{R}^p, \mathcal{M})$ and for any $x \in \mathcal{M}$, $\tilde{P}(x) = \mathrm{d}\tilde{p}(x)$ is the orthogonal projection on $\mathrm{T}_x\mathcal{M}$. Since $\mathbb{R}^p$ is normal and $\mathcal{M}$ and $\mathcal{E}(\mathcal{M})^c$ are closed, there exists $\mathsf{F}$ open such that $\mathcal{M} \subset \mathsf{F} \subset \mathcal{E}(\mathcal{M})$. Let $p \in \mathrm{C}^\infty(\mathbb{R}^p, \mathbb{R}^p)$ such that for any $x \in \mathsf{F}$, $p(x) = \tilde{p}(x)$ (given by Whitney extension theorem for instance). Finally, we define $P : \mathbb{R}^p \to \mathbb{R}^p$ such that for any $x \in \mathbb{R}^p$, $P(x) = \mathrm{d}p(x)$. Note that for any $x \in \mathcal{M}$, $P(x)$ is the orthogonal projection $\mathrm{T}_x\mathcal{M}$ and that $P \in \mathrm{C}^\infty(\mathbb{R}^p, \mathbb{R}^p)$.

### C.2  Stochastic Differential Equations on manifolds

**Stratanovitch integral**  For reasons that will become clear in the next paragraph, it is easier to define Stochastic Differential Equations (SDEs) on manifolds w.r.t the Stratanovitch integral (Kloeden and Platen, 2011, Part II, Chapter 3). We consider a filtered probability space $(\Omega, (\mathcal{F}_t)_{t \geq 0}, \mathbb{P})$. Let $(\mathbf{X}_t)_{t \geq 0}$ and $(\mathbf{Y}_t)_{t \geq 0}$ be two real continuous semimartingales. We define the quadratic covariation $([\mathbf{X}, \mathbf{Y}]_t)_{t \geq 0}$ such that for any $t \geq 0$

$$[\mathbf{X}, \mathbf{Y}]_t = \mathbf{X}_t\mathbf{Y}_t - \mathbf{X}_0\mathbf{Y}_0 - \int_0^t \mathbf{X}_s\mathrm{d}\mathbf{Y}_s - \int_0^t \mathbf{Y}_s\mathrm{d}\mathbf{X}_s.$$

We refer to Revuz and Yor (1999, Chapter IV) for more details on semimartingales and quadratic variations. We denote $[\mathbf{X}] = [\mathbf{X}, \mathbf{X}]$. In particular, we have that $([\mathbf{X}, \mathbf{Y}]_t)_{t \geq 0}$ is an adapted continuous process with finite-variation and therefore $[[\mathbf{X}, \mathbf{Y}]] = 0$. Let $(\mathbf{X}_t)_{t \geq 0}$ and $(\mathbf{Y}_t)_{t \geq 0}$ be two real continuous semimartingales, then we define the Stratanovitch integral as follows for any $t \geq 0$

$$\int_0^t \mathbf{X}_s \circ \mathrm{d}\mathbf{Y}_s = \int_0^t \mathbf{X}_s\mathrm{d}\mathbf{Y}_s + \tfrac{1}{2}[\mathbf{X}, \mathbf{Y}]_t.$$

In particular, denoting $(\mathbf{Z}_t^1)_{t \geq 0}$ and $(\mathbf{Z}_t^2)_{t \geq 0}$ the processes such that for any $t \geq 0$, $\mathbf{Z}_t^1 = \int_0^t \mathbf{X}_s \circ \mathrm{d}\mathbf{Y}_s$ and $\mathbf{Z}_t^2 = \int_0^t \mathbf{X}_s\mathrm{d}\mathbf{Y}_s$, we have that $[\mathbf{Z}^1] = [\mathbf{Z}^2]$. We refer to Kurtz et al. (1995) for more details on Stratanovitch integrals. Note that if for any $t \geq 0$, $\mathbf{X}_t = \int_0^t f(\mathbf{X}_s) \circ \mathrm{d}\mathbf{Y}_s$ with $\mathrm{C}^1(\mathbb{R}, \mathbb{R})$, then $[\mathbf{X}, \mathbf{Y}]_t = \int_0^t f(\mathbf{X}_s)f'(\mathbf{X}_s)\mathrm{d}\mathbf{Y}_s$. Assuming that $f \in \mathrm{C}^3(\mathbb{R}, \mathbb{R})$ we have that (Revuz and Yor, 1999, Chapter IV, Exercise 3.15)

$$f(\mathbf{X}_t) = f(\mathbf{X}_0) + \int_0^t f'(\mathbf{X}_s) \circ \mathrm{d}\mathbf{X}_s.$$

The proof relies on the fact that for any $t \geq 0$, $\mathrm{d}[\mathbf{X}, f'(\mathbf{X})]_t = f''(\mathbf{X}_t)\mathrm{d}[\mathbf{X}]_t$. This result should be compared with Itô's lemma. In particular, Stratanovitch calculus satisfies the ordinary chain rule making it a useful tool in differential geometry which makes a heavy use of diffeomorphism. Finally, we have the following correspondence between Stratanovitch and Itô SDEs. Assume that $(\mathbf{X}_t)_{t \in [0,T]}$ is a strong solution to $\mathrm{d}\mathbf{X}_t = b(t, \mathbf{X}_t)\mathrm{d}t + \sigma(t, \mathbf{X}_t) \circ \mathrm{d}\mathbf{B}_t$, with $b \in \mathrm{C}^\infty(\mathbb{R}^d, \mathbb{R}^d)$ and $\sigma \in \mathrm{C}^\infty(\mathbb{R}^d, \mathbb{R}^{d \times d})$. Then, we have that

$$\mathrm{d}\mathbf{X}_t = \{b(t, \mathbf{X}_t) + \bar{b}(\mathbf{X}_t)\}\mathrm{d}t + \sigma(t, \mathbf{X}_t)\mathrm{d}\mathbf{B}_t, \qquad \bar{b} = (-1/2)[\mathrm{div}(\sigma\sigma^\top) - \sigma\mathrm{div}(\sigma^\top)]. \quad \text{(S2)}$$

where for any $\mathrm{A} \in \mathrm{C}^\infty(\mathbb{R}^d, \mathbb{R}^{d \times d})$ we have that $\mathrm{div}(\mathrm{A}) \in \mathrm{C}^\infty(\mathbb{R}^d, \mathbb{R}^d)$ and for any $i \in \{1, \ldots, d\}$ and $x \in \mathbb{R}^d$, $\mathrm{div}(\mathrm{A})_i(x) = \sum_{j=1}^d \partial_j \mathrm{A}_{i,j}(x)$. In particular, note that if for $x_0 \in \mathbb{R}^d$, $\sigma(x_0)$ is an orthogonal projection, then $\sigma(x_0)\bar{b}(x_0) = 0$.

**SDEs on manifolds** We define semimartingales and SDEs on manifold through the lens of their actions on functions. A continuous $\mathcal{M}$-valued stochastic process $(\mathbf{X}_t)_{t\geq 0}$ is called a $\mathcal{M}$-valued semimartingale if for any $f \in \mathrm{C}^\infty(\mathcal{M})$ we have that $(f(\mathbf{X}_t))_{t\geq 0}$ is a real valued semimartingale. Let $\ell \in \mathbb{N}$, $V^{1:\ell} = \{V_i\}_{i=1}^\ell \in \mathcal{X}(\mathcal{M})^\ell$ and $Z^{1:\ell} = \{Z^i\}_{i=1}^\ell$ a collection of $\ell$ real-valued semimartingales. A $\mathcal{M}$-valued semimartingale $(\mathbf{X}_t)_{t\geq 0}$ is said to be the solution of $\mathrm{SDE}(V^{1:\ell}, Z^{1:\ell}, \mathbf{X}_0)$ up to a stopping $\tau$ with $\mathbf{X}_0$ a $\mathcal{M}$-valued random variable if for all $f \in \mathrm{C}^\infty(\mathcal{M})$ and $t \in [0, \tau]$ we have

$$f(\mathbf{X}_t) = f(\mathbf{X}_0) + \sum_{i=1}^\ell \int_0^t V_i(f)(\mathbf{X}_s) \circ \mathrm{d}\mathbf{Z}_s^i.$$

Since the previous SDE is defined w.r.t the Stratanovitch integral we have that if $(\mathbf{X}_t)_{t\geq 0}$ is a solution of $\mathrm{SDE}(V^{1:\ell}, Z^{1:\ell}, \mathbf{X}_0)$ and $\mathbf{\Phi} : \mathcal{M} \to \mathcal{N}$ is a diffeomorphism then $(\mathbf{\Phi}(\mathbf{X}_t))_{t\geq 0}$ is a solution of $\mathrm{SDE}(\mathbf{\Phi}_\star V^{1:\ell}, Z^{1:\ell}, \mathbf{\Phi}(\mathbf{X}_0))$, where $\mathbf{\Phi}_\star$ is the pushforward operation (see Hsu, 2002, Proposition 1.2.4). Because the vector fields $\{V_i\}_{i=1}^\ell$ are smooth we have that for any $\ell \in \mathbb{N}$, $V^{1:\ell} = \{V_i\}_{i=1}^\ell \in \mathcal{X}(\mathcal{M})^\ell$ and $Z^{1:\ell} = \{Z^i\}_{i=1}^\ell$ a collection of $\ell$ real-valued semimartingales, there exists a unique solution to $\mathrm{SDE}(V^{1:\ell}, Z^{1:\ell}, \mathbf{X}_0)$ (see Hsu, 2002, Theorem 1.2.9).

### C.3 Brownian motion on manifolds

In this section, we introduce the notion of Brownian motion on manifolds. We derive some of its basic convergence properties and provide alternative definitions (stochastic development, isometric embedding, random walk limit). These alternative definitions are the basis for our alternative methodologies to sample from the time-reversal. To simplify our discussion, we assume that $\mathcal{M}$ is a connected compact orientable Riemannian manifold equipped with the Levi–Civita connection $\nabla$. We denote $p_{\mathrm{ref}}^m$ the Haussdorff measure of the manifold (which coincides with the measure associated with the Riemannian volume form (see Federer, 2014, Theorem 2.10.10) and $p_{\mathrm{ref}} = p_{\mathrm{ref}}^m / p_{\mathrm{ref}}^m(\mathcal{M})$ the associated probability measure.

**Gradient, divergence and Laplace operators** Let $f \in \mathrm{C}^\infty(\mathcal{M})$. We define $\nabla f \in \mathcal{X}(\mathcal{M})$ such that for any $X \in \mathcal{X}(\mathcal{M})$ we have $\langle X, \nabla f \rangle_{\mathcal{M}} = X(f)$. Let $\{X_i\}_{i=1}^d \in \mathcal{X}(\mathcal{M})^d$ such that for any $x \in \mathcal{M}$, $\{X_i(x)\}_{i=1}^d$ is an orthonormal basis of $\mathrm{T}_x\mathcal{M}$. Then, we define $\mathrm{div} : \mathcal{X}(\mathcal{M}) \to \mathrm{C}^\infty(\mathcal{M})$ (linear) such that for any $X \in \mathcal{X}(\mathcal{M})$, $\mathrm{div}(X) = \sum_{i=1}^d \langle \nabla_{X_i} X, X_i \rangle_{\mathcal{M}}$. The following Stokes formula (also called divergence theorem, see Lee (2018, p.51)) holds for any $f \in \mathrm{C}^\infty(\mathcal{M})$ and $X \in \mathcal{X}(\mathcal{M})$, $\int_M \mathrm{div}(X)(x) f(x) \mathrm{d}p_{\mathrm{ref}}(x) = -\int_M X(f)(x) \mathrm{d}p_{\mathrm{ref}}(x)$. Let $X = \sum_{i=1}^d a_i X_i$ in local coordinates. Using the Stokes formula and the definition of the gradient we get that in local coordinates

$$\nabla f = \sum_{i,j=1}^d g^{i,j} \partial_i f X_j, \qquad \mathrm{div}(X) = \det(G)^{-1/2} \sum_{i=1}^d \partial_i (\det(G)^{1/2} a_i).$$

The Laplace–Beltrami operator is given by $\Delta_{\mathcal{M}} : \mathrm{C}^\infty(M) \to \mathrm{C}^\infty(M)$ and for any $f \in \mathrm{C}^\infty(M)$ by $\Delta_{\mathcal{M}}(f) = \mathrm{div}(\mathrm{grad}(f))$. In local coordinates we obtain $\Delta_{\mathcal{M}}(f) = \det(G)^{-1/2} \sum_{i=1}^d \partial_i(\det(G)^{1/2} \sum_{j=1}^d g^{i,j} \partial_j f)$. Using the Nash isometric embedding theorem (Gunther, 1991) we will see that $\Delta_{\mathcal{M}}$ can always be written as a sum of squared operators. However, this result requires an *extrinsic* point of view as it relies on the existence of projection operators. In contrast, if we consider the orthonormal bundle $\mathrm{O}\mathcal{M}$, see (Hsu, 2002, Chapter 2), we can define the Laplace–Bochner operator $\Delta_{\mathrm{O}\mathcal{M}} : \mathrm{C}^\infty(\mathrm{O}\mathcal{M}) \to \mathrm{C}^\infty(\mathrm{O}\mathcal{M})$ as $\Delta_{\mathrm{O}\mathcal{M}} = \sum_{i=1}^d H_i^2$, where we recall that for any $i \in \{1, \ldots, d\}$, $H_i$ is the horizontal lift of $e_i$. In this case, $\Delta_{\mathrm{O}\mathcal{M}}$ is a sum of squared operators and we have that for any $f \in \mathrm{C}^\infty(\mathcal{M})$, $\Delta_{\mathrm{O}\mathcal{M}}(f \circ \pi) = \Delta_{\mathcal{M}}(f)$ (see Hsu, 2002, Proposition 3.1.2). Being able to express the various Laplace operators as a sum of squared operators is key to express the associated diffusion process as the solution of an SDE.

**Alternatives definitions of Brownian motion** We are now ready to define a Brownian motion on the manifold $\mathcal{M}$. Using the Laplace–Beltrami operator, we can introduce the Brownian motion through the lens of diffusion processes.

**Definition S5** (Brownian motion). *Let $(\mathbf{B}_t^{\mathcal{M}})_{t\geq 0}$ be a $\mathcal{M}$-valued semimartingale. $(\mathbf{B}_t^{\mathcal{M}})_{t\geq 0}$ is a Brownian motion on $\mathcal{M}$ if for any $f \in \mathrm{C}^\infty(\mathcal{M})$, $(\mathbf{M}_t^f)_{t\geq 0}$ is a local martingale where for any $t \geq 0$*

$$\mathbf{M}_t^f = f(\mathbf{B}_t^{\mathcal{M}}) - f(\mathbf{B}_0^{\mathcal{M}}) - \tfrac{1}{2} \int_0^t \Delta_{\mathcal{M}} f(\mathbf{B}_s^{\mathcal{M}}) \mathrm{d}s.$$

Note that this definition is in accordance with the definition of the Brownian motion as a diffusion process in the Euclidean space $\mathbb{R}^d$, since in this case $\Delta_{\mathcal{M}} = \Delta$. A key property of frame bundles and orthonormal bundles is that any semimartingale on $\mathcal{M}$ can be associated to a process on $F\mathcal{M}$ (or $O\mathcal{M}$) and a process on $\mathbb{R}^d$. The proof of the following result can be found in Hsu (2002, Propositions 3.2.1 and 3.2.2).

**Proposition S6** (Intrinsic view of Brownian motion)**.** *Let* $(\mathbf{B}_t^{\mathcal{M}})_{t\geq 0}$ *be a* $\mathcal{M}$-*valued semimartingales. Then* $(\mathbf{B}_t^{\mathcal{M}})_{t\geq 0}$ *is a Brownian motion on* $\mathcal{M}$ *if and only on the following conditions hold:*

  a) *The horizontal lift* $(\mathbf{U}_t)_{t\geq 0}$ *is a* $\Delta_{O\mathcal{M}}/2$ *diffusion process, i.e. for any* $f \in C^\infty(O\mathcal{M})$*, we have that* $(\mathbf{M}_t^f)_{t\geq 0}$ *is a local martingale where for any* $t \geq 0$

$$\mathbf{M}_t^f = f(\mathbf{U}_t) - f(\mathbf{U}_0) - \tfrac{1}{2}\int_0^t \Delta_{O\mathcal{M}} f(\mathbf{U}_s)\mathrm{d}s.$$

  b) *The stochastic antidevelopment of* $(\mathbf{B}_t^{\mathcal{M}})_{t\geq 0}$ *is a* $\mathbb{R}^d$-*valued Brownian motion* $(\mathbf{B}_t)_{t\geq 0}$*.*

In particular the previous proposition provides us with an *intrisic* way to sample the Brownian motion on $\mathcal{M}$ with initial condition $\mathbf{B}_0^{\mathcal{M}}$. First sample $(\mathbf{U}_t)_{t\geq 0}$ solution of $\mathrm{SDE}(H^{1:d}, \mathbf{B}^{1:d}, \mathbf{U}_0)$ with $H^{1:d} = \{H_i\}_{i=1}^d$ and $\pi(\mathbf{U}_0) = \mathbf{B}_0^{\mathcal{M}}$ and $\mathbf{B}^{1:d}$ the Euclidean $d$-dimensional Brownian motion. Then, we recover the $\mathcal{M}$-valued Brownian motion $(\mathbf{B}_t^{\mathcal{M}})_{t\geq 0}$ upon letting $(\mathbf{B}_t^{\mathcal{M}})_{t\geq 0} = (\pi(\mathbf{U}_t))_{t\geq 0}$.

We now consider an *extrinsic* approach to the sampling of Brownian motions on $\mathcal{M}$. Using the Nash embedding theorem (Gunther, 1991), there exists $p \in \mathbb{N}$ such that without loss of generality we can assume that $\mathcal{M} \subset \mathbb{R}^p$. For any $x \in \mathcal{M}$, we denote $\mathrm{P}(x) : \mathbb{R}^p \to \mathrm{T}_x\mathcal{M}$ the projection operator. In addition for any $x \in \mathcal{M}$, we denote $\{\mathrm{P}_i(x)\}_{i=1}^p = \{\mathrm{P}(x)e_i\}_{i=1}^p$, where $\{e_i\}_{i=1}^p$ is the canonical basis of $\mathbb{R}^p$. For any $i \in \{1, \ldots, p\}$, we smoothly extend $\mathrm{P}_i$ to $\mathbb{R}^p$. In this case, we have the following proposition (Hsu, 2002, Theorem 3.1.4):

**Proposition S7** (Extrinsic view of Brownian motion)**.** *For any* $f \in C^\infty(\mathcal{M})$ *we have that* $\Delta_{\mathcal{M}}(f) = \sum_{i=1}^p \mathrm{P}_i(\mathrm{P}_i(f))$*. Hence, we have that* $(\mathbf{B}_t^{\mathcal{M}})_{t\geq 0}$ *solution of* $\mathrm{SDE}(\{\mathrm{P}_i\}_{i=1}^p, \mathbf{B}^{1:p}, \mathbf{B}_0^{\mathcal{M}})$ *with* $\mathbf{B}_0^{\mathcal{M}}$ *a* $\mathcal{M}$-*valued random variable and* $\mathbf{B}^{1:p}$ *a* $\mathbb{R}^p$-*valued Brownian motion.*

The second part of this proposition, stems from the fact that any solution of $\mathrm{SDE}(\{V_i\}_{i=1}^\ell, \mathbf{B}^{1:\ell}, \mathbf{X}_0)$, where $\mathbf{X}_0$ is a $\mathcal{M}$-valued random variable and $\mathbf{B}^{1:\ell}$ a $\mathbb{R}^\ell$-valued Brownian motion is a diffusion process with generator $\mathcal{A}$ such that for any $f \in C^\infty(\mathcal{M})$, $\mathcal{A}(f) = \sum_{i=1}^\ell V_i(V_i(f))$. The *extrinsic* approach is particularly convenient since the SDE appearing in Proposition S7 can be seen as an SDE on the Euclidean space $\mathbb{R}^p$.

We finish this paragraph, by investigating the behaviour of the Brownian motion in local coordinates. For simplicity, we assume here that we have access to a system of global coordinates. In the case where the coordinates are strictly local then we refer to Ikeda and Watanabe (1989, Chapter 5, Theorem 1) for a construction of a global solution by patching local solutions. We denote $\{X_k, X_{i,j}\}_{1\leq i,j,k\leq d}$ such that for any $u \in F\mathcal{M}$, $\{X_k(u), X_{i,j}(u)\}_{1\leq i,j,k\leq d}$ is a basis of $\mathrm{T}_u F\mathcal{M}$, Using properties of the horizontal lift, see (Hsu, 2002, Chapter 2), we get that $(\mathbf{U}_t)_{t\geq 0} = (\{\mathbf{X}_t^k, \mathbf{E}_t^{i,j}\}_{1\leq i,j,k\leq d})$ obtained in Proposition S6 is given in the global coordinates for any $i, j, k \in \{1, \ldots, d\}$ by

$$\mathrm{d}\mathbf{X}_t^k = \sum_{j=1}^d \mathbf{E}_t^{k,j} \circ \mathrm{d}\mathbf{B}_t^k, \qquad \mathrm{d}\mathbf{E}_t^{i,j} = -\sum_{n=1}^d \{\sum_{\ell,m=1}^d \mathbf{E}_t^{\ell,n}\mathbf{E}_t^{m,j}\Gamma_{\ell,m}^i(\mathbf{X}_t)\} \circ \mathrm{d}\mathbf{B}_t^n.$$

By definition of the Stratanovitch integral we have that for any $k \in \{1, \ldots, d\}$

$$\mathrm{d}\mathbf{X}_t^k = \sum_{j=1}^d \{\mathbf{E}_t^{k,j}\mathrm{d}\mathbf{B}_t^k + \tfrac{1}{2}\mathrm{d}[\mathbf{E}_t^{k,j}, \mathbf{B}_t^j]_t\}.$$

Let $(\mathbf{M}_t)_{t\geq 0} = (\{\mathbf{M}_t^k\}_{k=1}^d)_{t\geq 0}$ such that for any $t \geq 0$ and $k \in \{1, \ldots, d\}$ $\mathbf{M}_t^k = \sum_{j=1}^d \int_0^t \mathbf{E}_t^{k,j}\mathrm{d}\mathbf{B}_t^k$. We obtain that $\mathrm{d}\mathbf{M}_t = G(\mathbf{X}_t)^{-1/2}\mathrm{d}\mathbf{B}_t$ for some $d$-dimensional Brownian motion $(\mathbf{B}_t)_{t\geq 0}$, using Lévy's characterization of Brownian motion. In addition, we have that for any $k, j \in \{1, \ldots, d\}$

$$[\mathbf{E}^{k,j}, \mathbf{B}^j]_t = -\sum_{\ell,m=1}^d \int_0^t \mathbf{E}_t^{\ell,j}\mathbf{E}_t^{m,j}\Gamma_{\ell,m}^k(\mathbf{X}_t)\mathrm{d}t$$

Hence, using this result and the fact that $\sum_{j=1}^d \mathbf{E}_t^{\ell,j}\mathbf{E}_t^{m,j} = g^{\ell,m}(\mathbf{X}_t)$, we get that for any $k \in \{1, \ldots, d\}$

$$\mathrm{d}\mathbf{X}_t^k = -\tfrac{1}{2}\sum_{\ell,m=1}^d g^{\ell,m}(\mathbf{X}_t)\Gamma_{\ell,m}^k(\mathbf{X}_t)\mathrm{d}t + (G(\mathbf{X}_t)^{-1/2}\mathrm{d}\mathbf{B}_t)^k.$$

Note that this result could also have been obtained using the expression of the Laplace–Beltrami in local coordinates.

**Brownian motion and random walks**  In the previous paragraph we consider three SDEs to obtain a Brownian motion on $\mathcal{M}$ (stochastic development, isometric embedding and local coordinates). In this section, we summarize results from Jørgensen (1975) establishing the limiting behaviour of Geodesic Random Walks (GRWs) when the stepsize of the random walk goes to 0. This will be of particular interest when considering the time-reversal process. We start by defining the geodesic random walk on $\mathcal{M}$, following Jørgensen (1975, Section 2).

Let $\{\nu_x\}_{x \in \mathcal{M}}$ such that for any $x \in \mathcal{M}$, $\nu_x : \mathcal{B}(\mathrm{T}_x\mathcal{M}) \to [0, 1]$ with $\nu_x(\mathrm{T}_x\mathcal{M}) = 1$, i.e. for any $x \in \mathcal{M}$, $\nu_x$ is a probability measure on $\mathrm{T}_x\mathcal{M}$. Assume that for any $x \in \mathcal{M}$, $\int_{\mathcal{M}} \|v\|^3 \mathrm{d}\nu_x(v) < +\infty$. In addition assume that there exists $\mu^{(1)} \in \mathcal{X}(\mathcal{M})$ and $\mu^{(2)} \in \mathcal{X}^2(\mathcal{M})$, where $\mathcal{X}^2(\mathcal{M})$ is the section $\Gamma(\mathcal{M}, \sqcup_{x \in \mathcal{M}}\mathcal{L}(\mathrm{T}_x\mathcal{M}))$, such that for any $x \in \mathcal{M}$, $\int_{\mathcal{M}} v\mathrm{d}\nu_x(v) = \mu^{(1)}(x)$ and $\int_{\mathcal{M}} v \otimes v\mathrm{d}\nu_x(v) = \mu^{(2)}(x)$. In addition, we assume that for any $x \in \mathcal{M}$, $\Sigma(x) = \mu^{(2)}(x) - \mu^{(1)}(x) \otimes \mu^{(1)}(x)$ is strictly positive definite and that there exists $\mathrm{L} \geq$ such that for any $x, y \in \mathcal{M}$, $\|\nu_x - \nu_y\|_{\mathrm{TV}} \leq \mathrm{L}d_{\mathcal{M}}(x, y)$. Where we have that for any $\nu_1 \in \mathcal{P}(\mathrm{T}_x\mathcal{M})$ and $\nu_2 \in \mathcal{P}(\mathrm{T}_y\mathcal{M})$,

$$\|\nu_x - \nu_y\|_{\mathrm{TV}} = \sup\{\nu_1[f] - \Gamma_x^y(\gamma)_\#\nu_2[f] \ : \ \gamma \in \mathrm{Geo}_{x,y}, \ f \in \mathrm{C}(\mathrm{T}_x\mathcal{M})\}.$$

Note that if $d_{\mathcal{M}}(x, y) \leq \varepsilon$ then for some $\varepsilon > 0$ we have that $|\mathrm{Geo}_{x,y}| = 1$.

**Definition S8** (Geodesic random walk).  *Let $X_0$ be a $\mathcal{M}$-valued random variable. For any $\gamma > 0$, we define $(\mathbf{X}_t^\gamma)_{t \geq 0}$ such that $\mathbf{X}_0^\gamma = X_0$ and for any $n \in \mathbb{N}$ and $t \in [0, \gamma]$, $\mathbf{X}_{n\gamma+t} = \exp_{\mathbf{X}_{n\gamma}}[t\gamma\{\mu_n + (1/\sqrt{\gamma})(V_n - \mu_n)\}]$, where $(V_n)_{n \in \mathbb{N}}$ is a sequence of random variables in such that for any $n \in \mathbb{N}$, $V_n$ has distribution $\nu_{\mathbf{X}_{n\gamma}}$ conditionally to $\mathbf{X}_{n\gamma}$.*

For any $\gamma > 0$, the process $(X_n^\gamma)_{n \in \mathbb{N}} = (\mathbf{X}_{n\gamma}^\gamma)_{n \in \mathbb{N}}$ is called a geodesic random walk. In particular, for any $\gamma > 0$ we denote $(\mathrm{R}_n^\gamma)_{n \in \mathbb{N}}$ the sequence of Markov kernels such that for any $n \in \mathbb{N}$, $x \in \mathcal{M}$ and $\mathrm{A} \in \mathcal{B}(\mathcal{M})$ we have that $\delta_x\mathrm{R}(\mathrm{A}) = \mathbb{P}(X_n^\gamma \in \mathrm{A})$, with $X_0^\gamma = x$. The following theorem establishes that the limiting dynamics of a geodesic random walk is associated with a diffusion process on $\mathcal{M}$ whose coefficients only depends on the properties of $\nu$ (see Jørgensen, 1975, Theorem 2.1).

**Theorem S9** (Convergence of geodesic random walks).  *For any $t \geq 0$, $f \in \mathrm{C}(\mathcal{M})$ and $x \in \mathcal{M}$ we have that $\lim_{\gamma \to 0} \|\mathrm{R}_\gamma^{[t/\gamma]}[f] - \mathrm{P}_t[f]\|_\infty = 0$, where $(\mathrm{P}_t)_{t \geq 0}$ is the semi-group associated with the infinitesimal generator $\mathcal{A} : \mathrm{C}^\infty(\mathcal{M}) \to \mathrm{C}^\infty(\mathcal{M})$ given for any $f \in \mathrm{C}^\infty(\mathcal{M})$ by $\mathcal{A}(f) = \langle\mu^{(1)}, \nabla f\rangle_{\mathcal{M}} + \frac{1}{2}\langle\Sigma, \nabla^2 f\rangle_{\mathcal{M}}$.*

In particular if $\mu^{(1)} = 0$ and $\mu^{(2)} = \mathrm{Id}$ then the random walk converges towards a Brownian motion on $\mathcal{M}$ in the sense of the convergence of semi-groups. For any $x \in \mathcal{M}$ in local coordinates we have that $\Phi_\#\nu_x$ has zero mean and covariance matrix $G(x)$, where $\Phi$ is a local chart around $x$ and $G(x) = (g_{i,j}(x))_{1 \leq i,j \leq d}$ the coordinates of the metric in that chart.

**Convergence of Brownian motion**  We finish this section with a few considerations regarding the convergence of the Brownian motion on $\mathcal{M}$. Since we have assumed that $\mathcal{M}$ is compact we have that there exist $(\Phi_k)_{k \in \mathbb{N}}$ an orthonormal basis of $-\Delta_{\mathcal{M}}$ in $\mathrm{L}^2(p_{\mathrm{ref}})$, $(\lambda_k)_{k \in \mathbb{N}}$ such that for any $i, j \in \mathbb{N}$, $i \leq j$, $\lambda_i \leq \lambda_j$ and $\lambda_0 = 0$, $\Phi_0 = 1$ and for any $k \in \mathbb{N}$, $\Delta_{\mathcal{M}}\Phi_k = -\lambda_k\Phi_k$. For any $t \geq 0$ and $x, y \in \mathcal{M}$, $p_{t|0}(y|x) = \sum_{k \in \mathbb{N}} \mathrm{e}^{-\lambda_k t}\Phi_k(x)\Phi_k(y)$ where for any $f \in \mathrm{C}^\infty$ we have

$$\mathbb{E}[f(\mathbf{B}_t^{\mathcal{M},x})] = \int_{\mathcal{M}} p_{t|0}(x, y)f(y)\mathrm{d}p_{\mathrm{ref}}(y),$$

where $(\mathbf{B}_t^{\mathcal{M},x})_{t \geq 0}$ is the Brownian motion on $\mathcal{M}$ with $\mathbf{B}_0^{\mathcal{M},x} = x$ and $p_{\mathrm{ref}}$ is the probability measure associated with the Haussdorff measure on $\mathcal{M}$. We also have the following result (see Urakawa, 2006, Proposition 2.6).

**Proposition S10** (Convergence of Brownian motion).  *For any $t > 0$, $\mathrm{P}_t$ admits a density $p_{t|0}$ w.r.t $p_{\mathrm{ref}}$ and $p_{\mathrm{ref}}\mathrm{P}_t = p_{\mathrm{ref}}$, i.e. $p_{\mathrm{ref}}$ is an invariant measure for $(\mathrm{P}_t)_{t \geq 0}$. In addition, if there exists $C, \alpha \geq 0$ such that for any $t \in (0, 1]$, $p_{t|0}(x|x) \leq Ct^{-\alpha/2}$ then for any $p_0 \in \mathcal{P}(\mathcal{M})$ and for any $t \geq 1/2$ we have*

$$\|p_0\mathrm{P}_t - p_{\mathrm{ref}}\|_{\mathrm{TV}} \leq C^{1/2}\mathrm{e}^{\lambda_1/2}\mathrm{e}^{-\lambda_1 t},$$

*where $\lambda_1$ is the first non-negative eigenvalue of $-\Delta_{\mathcal{M}}$ in $\mathrm{L}^2(p_{\mathrm{ref}})$ and we recall that $(\mathrm{P}_t)_{t \geq 0}$ is the semi-group of the Brownian motion.*

A review on lower bounds on the first positive eigenvalue of the Laplace–Beltrami operator can be found in (He, 2013). These lower bounds usually depend on the Ricci curvature of the manifold or its diameter. We conclude this section by noting that in the non-compact case (Li, 1986) establishes similar estimates in the case of a manifold with non-negative Ricci curvature and maximal volume growth.

## D  Likelihood computation

### D.1  ODE likelihood computation

Similarly to Song et al. (2021), once the score is learned we can use it in conjunction with an Ordinary Differential Equation (ODE) solver to compute the likelihood of the model. Let $(\Phi_t)_{t\in[0,T]}$ be a family of vector fields. We define $(\mathbf{X}_t)_{t\in[0,T]}$ such that $\mathbf{X}_0$ has distribution $p_0$ (the data distribution) and satisfying $\mathrm{d}\mathbf{X}_t = \Phi_t(\mathbf{X}_t)\mathrm{d}t$. Assuming that $p_0$ admits a density w.r.t. $p_{\text{ref}}$ then for any $t \in [0,T]$, the distribution of $\mathbf{X}_t$ admits a density w.r.t. $p_{\text{ref}}$ and we denote $p_t$ this density. We recall that $\mathrm{d}\log p_t(\mathbf{X}_t) = -\mathrm{div}(\Phi_t)(\mathbf{X}_t)\mathrm{d}t$, see Mathieu and Nickel (2020, Proposition 2) for instance.

Recall that we consider a Brownian motion on the manifold as a forward process $(\mathbf{B}_t^{\mathcal{M}})_{t\in[0,T]}$ with $\{p_t\}_{t\in[0,T]}$ the associated family of densities. Thus we have that for any $t \in [0,T]$ and $x \in \mathcal{M}$

$$\partial_t p_t(x) = \tfrac{1}{2}\Delta_{\mathcal{M}}p_t(x) = \mathrm{div}\left(\tfrac{1}{2}p_t\nabla\log p_t\right)(x).$$

Hence, we can define $(\mathbf{X}_t)_{t\in[0,T]}$ satisfying $\mathrm{d}\mathbf{X}_t = -\tfrac{1}{2}\nabla\log p_t(\mathbf{X}_t)\mathrm{d}t$ such that $\mathbf{X}_0$ has distribution $p_0$. Defining $(\hat{\mathbf{X}}_t)_{t\in[0,T]} = (\mathbf{X}_{T-t})_{t\in[0,T]}$, it follows that $\hat{\mathbf{X}}_0$ has distribution $\mathcal{L}(\mathbf{X}_T)$ and satisfies

$$\mathrm{d}\hat{\mathbf{X}}_t = \tfrac{1}{2}\nabla\log p_{T-t}(\hat{\mathbf{X}}_t)\mathrm{d}t. \tag{S3}$$

Finally, we introduce $(\mathbf{Y}_t)_{t\in[0,T]}$ satisfying (S3) but such that $\mathbf{Y}_0 \sim p_{\text{ref}}$. Note that if $T \geq 0$ is large then the two processes $(\mathbf{Y}_t)_{t\in[0,T]}$ and $(\hat{\mathbf{X}}_t)_{t\in[0,T]}$ are close since $\mathcal{L}(\mathbf{X}_T)$ is close to $p_{\text{ref}}$.

Therefore, using the score network and a manifold ODE solver (as in Mathieu and Nickel, 2020), we are able to approximately solve the following ODE

$$\mathrm{d}\log q_t(\hat{\mathbf{X}}_t^\theta) = -\tfrac{1}{2}\mathrm{div}(\mathbf{s}_\theta(T-t,\cdot))(\hat{\mathbf{X}}_t^\theta)\mathrm{d}t,$$

with $q_t$ the density of $\mathbf{Y}_t^\theta$ w.r.t. $p_{\text{ref}}$ and $\log q_0(\mathbf{Y}_0) = 0$ with $\mathrm{d}\mathbf{Y}_t^\theta = \tfrac{1}{2}\mathrm{div}(\mathbf{s}_\theta(T-t,\mathbf{Y}_t^\theta))\mathrm{d}t$ and $\mathbf{Y}_0^\theta \sim p_{\text{ref}}$. The likelihood approximation of the model is then given by $\mathbb{E}[\log q_T(\hat{\mathbf{X}}_T^\theta)] = \int_{\mathcal{M}}\log q_T(x)\mathrm{d}p_{\text{data}}(x)$, where $(\hat{\mathbf{X}}_t^\theta)_{t\in[0,T]} = (\mathbf{X}_{T-t}^\theta)_{t\in[0,T]}$ with $\mathrm{d}\mathbf{X}_t^\theta = -\tfrac{1}{2}\mathrm{div}(\mathbf{s}_\theta(t,\mathbf{X}_t^\theta))\mathrm{d}t$ and $\mathbf{X}_0^\theta \sim p_{\text{data}}$. In App. D.2, we highlight that this is *not* the likelihood of the SDE model.

### D.2  Difference between ODE and SDE likelihood computations

In this section, we show that the likelihood computation from Song et al. (2021) does not coincide with the likelihood computation obtained with the SDE model. We present our findings in the Riemannian setting but our results can be adapted to the Euclidean setting with arbitrary forward dynamics. Recall that we consider a Brownian motion on the manifold as a forward process $(\mathbf{B}_t^{\mathcal{M}})_{t\in[0,T]}$ with $(p_t)_{t\in[0,T]}$ the associated family of densities. We have that for any $t \in [0,T]$ and $x \in \mathcal{M}$

$$\partial_t p_t(x) = \tfrac{1}{2}\Delta_{\mathcal{M}}p_{t|0}(x) = \mathrm{div}(\tfrac{1}{2}p_t\nabla\log p_t)(x). \tag{S4}$$

**ODE model.**  In the case of the ODE model, we define $(\mathbf{X}_t)_{t\in[0,T]}$ such that $\mathbf{X}_0 \sim p_0$ and satisfies $\mathrm{d}\mathbf{X}_t = -\tfrac{1}{2}\nabla\log p_t(\mathbf{X}_t)\mathrm{d}t$. The family of densities $(q_t)_{t\in[0,T]}$ associated with $(\mathbf{X}_t)_{t\in[0,T]}$ also satisfies (S4). Now consider $(\hat{\mathbf{X}}_t)_{t\in[0,T]} = (\mathbf{X}_{T-t})_{t\in[0,T]}$, this satisfies $\hat{\mathbf{X}}_0 \sim p_T$ with

$$\mathrm{d}\hat{\mathbf{X}}_t = \tfrac{1}{2}\nabla\log p_{T-t}(\hat{\mathbf{X}}_t)\mathrm{d}t. \tag{S5}$$

Finally, we consider $(\mathbf{Y}_t^{\mathrm{ODE}})_{t\in[0,T]}$ which also satisfies Eq. (S5) and such that $\mathbf{Y}_0^{\mathrm{ODE}} \sim p_{\text{ref}}$. Denoting $(q_t^{\mathrm{ODE}})_{t\in[0,T]}$ the densities of $(\mathbf{Y}_t^{\mathrm{ODE}})_{t\in[0,T]}$ w.r.t. $p_{\text{ref}}$ we have for any $t \in [0,T]$ and $x \in \mathcal{M}$

$$\partial_t q_t^{\mathrm{ODE}}(x) = -\mathrm{div}(\tfrac{1}{2}q_t^{\mathrm{ODE}}\nabla\log p_{T-t})(x). \tag{S6}$$

**SDE model.** When sampling we consider a process $(\mathbf{Y}_t^{\mathrm{SDE}})_{t\in[0,T]}$ such that $\mathbf{Y}_0^{\mathrm{SDE}}$ has distribution $p_{\mathrm{ref}}$ and whose family of densities $(q_t^{\mathrm{SDE}})_{t\in[0,T]}$ satisfies for any $t \in [0,T]$ and $x \in \mathcal{M}$

$$\partial_t q_t^{\mathrm{SDE}}(x) = -\mathrm{div}(\nabla \log p_{T-t} q_t^{\mathrm{SDE}}(x)) + \tfrac{1}{2}\Delta_{\mathcal{M}} q_t^{\mathrm{SDE}}(x) = -\mathrm{div}(q_t^{\mathrm{SDE}}\{\nabla \log p_{T-t} - \tfrac{1}{2}\nabla \log q_t^{\mathrm{SDE}}\})(x).$$
(S7)

Hence, Eq. (S6) and Eq. (S7) do not agree, except if $q_t^{\mathrm{SDE}} = q_t^{\mathrm{ODE}} = p_{T-t}$ which is the case if and only if $\mathbf{Y}_0^{\mathrm{SDE}}$ and $\mathbf{Y}_0^{\mathrm{ODE}}$ have the same distribution as $\mathbf{X}_T$. Note that it is possible to evaluate the likelihood of the SDE model using that

$$\partial_t \log q_t^{\mathrm{SDE}}(\mathbf{Y}_t^{\mathrm{SDE}}) = \{\nabla \log p_{T-t}(\mathbf{Y}_t^{\mathrm{SDE}}) - \tfrac{1}{2}\nabla \log q_t^{\mathrm{SDE}}(\mathbf{Y}_t^{\mathrm{SDE}})\}\mathrm{d}t.$$

We can use the score approximation $\mathbf{s}_\theta(t,x)$ to approximate $\nabla \log p_t(x)$ for any $t \in [0,T]$ and $x \in \mathcal{M}$. In order to approximate $\nabla \log q_t^{\mathrm{SDE}}$, one can consider another neural network $\mathbf{t}_\theta(t,x)$ approximating $\nabla \log q_t^{\mathrm{SDE}}(x)$ for any $t \in [0,T]$ and $x \in \mathcal{M}$. This approximation can be obtained using the implicit score loss presented in Sec. 3.4.

# E  Parametric family of vector fields

We approximate $(\nabla \log p_t)_{t\in[0,T]}$ by a family of functions $\{\mathbf{s}_\theta\}_{\theta\in\Theta}$ where $\Theta$ is a set of parameters and for any $\theta \in \Theta$, $\mathbf{s}_\theta : [0,T] \to \mathcal{X}(\mathcal{M})$. In this work, we consider several parameterisations of vector fields:

- **Projected vector field**. We define $\mathbf{s}_\theta(t,x) = \mathrm{proj}_{T_x\mathcal{M}}(\tilde{\mathbf{s}}_\theta(t,x)) = P(x)\tilde{\mathbf{s}}_\theta(t,x)$ for any $t \in [0,T]$ and $x \in \mathcal{M}$, with $\tilde{\mathbf{s}}_\theta : \mathbb{R}^p \times [0,T] \to \mathbb{R}^p$ an ambient vector field and $P(x)$ the orthogonal projection over $T_x\mathcal{M}$ at $x \in M$. According to Rozen et al. (2021, Lemma 2), then $\mathrm{div}(\mathbf{s}_\theta)(x,t) = \mathrm{div}_E(\mathbf{s}_\theta)(x,t)$ for any $x \in \mathcal{M}$, where $\mathrm{div}_E$ denotes the standard Euclidean divergence.

- **Divergence-free vector fields**: For any Lie group $G$, any basis of the Lie algebra $\mathfrak{g} = T_e G$ yields a global frame. Indeed, let $v \in \mathfrak{g}$ and define the flow $\Phi : \mathbb{R} \times \mathcal{M} \to \mathcal{M}$ given for any $t \in \mathbb{R}$ and $x \in M$ by $\Phi_t^v(x) = x\exp_e(tv)$. Then defining $\{E_i\}_{i=1}^d = \{\partial_t\Phi_0^{v_i}\}_{i=1}^d$, where $\{v_i\}_{i=1}^d$ is a basis of $\mathfrak{g}$, we get that $\{E_i\}_{i=1}^d$ is a left-invariant global frame. As a result, we have that for any $i \in \{1,\dots,d\}$, $\mathrm{div}(E_i) = 0$ (for the classical left invariant metric). This result simplifies the computation of $\mathrm{div}(\mathbf{s}_\theta)$ where $\mathbf{s}_\theta(t,x) = \sum_{i=1}^d s_\theta^i(t,x)E_i(x)$ for any $t \in [0,T]$ and $x \in \mathcal{M}$ since we have that $\mathrm{div}(\mathbf{s}_\theta)(t,x) = \sum_{i=1}^d E_i(s_\theta^i)(t,x) + \sum_{i=1}^d s_\theta^i(t,x)\mathrm{div}(E_i)(x) = \sum_{i=1}^d ds_\theta^i(E_i)(t,x)$ (see Falorsi and Forré, 2020). Note that this approach can be extended to any homogeneous space $(G,H)$.

- **Coordinates vector fields**. We define $\mathbf{s}_\theta(t,x) = \sum_{i=1}^d \mathbf{s}_\theta^i(t,x)E_i(x)$ for any $t \in [0,T]$ and $x \in \mathcal{M}$, with $\{E_i\}_{i=1}^d = \{\partial_i\varphi(\varphi^{-1}(x))\}_{i=1}^d$ the vector fields induced by a choice of local coordinates, where $\varphi$ is a local parameterization $\varphi : \mathsf{U} \to \mathcal{M}$ and $z \in \mathsf{U} \subset \mathbb{R}^d$. Then the divergence can be computed in these local coordinates $\mathrm{div}(\mathbf{s}_\theta)(t,\varphi(z)) = |\det G|^{-1/2}\sum_{i=1}^d \partial_i\{|\det G|^{1/2}\mathbf{s}_\theta^i(t,\varphi(\cdot))\}(z)$. In the case of the sphere, one recovers the standard divergence in spherical coordinates using this formula. Note that $\{E_i\}_{i=1}^d$ does not span the tangent bundle except if the manifold is parallelizable. The sphere is a well-known example of non-parallelizable manifold, as per the *hairy ball theorem*.

# F  Eigensystems of the Laplace–Beltrami operator and heat kernels

In this section, we recall the eigenfunctions and eigenvalues of the Laplace–Beltrami operator in two specific cases: the $d$-dimensional torus and the $d$-dimensional sphere. We also highlight that the heat kernel on compact manifold can be written as an infinite series using the Sturm–Liouville decomposition.

**The case of the torus** Let $\{b_i\}_{i=1}^d$ be a basis of $\mathbb{R}^d$. We consider the associated lattice on $\mathbb{R}^d$, i.e. $\Gamma = \{\sum_{i=1}^d \alpha_i b_i : \{\alpha_i\}_{i=1}^d \in \mathbb{Z}^d\}$. Finally, the associated $d$-dimensional torus is defined as $\mathbb{T}_\Gamma = \mathbb{R}^d/\Gamma$. Denote $\mathrm{B} = (b_1,\dots,b_d) \in \mathbb{R}^{d\times d}$. Let $\{\bar{b}_i\}_{i=1}^d \in (\mathbb{R}^d)^d$ such that $(\mathrm{B}^{-1})^\top = (\bar{b}_1,\dots,\bar{b}_d)$. We define $\Gamma^\star = \{\sum_{i=1}^d \alpha_i\bar{b}_i : \{\alpha_i\}_{i=1}^d \in \mathbb{Z}^d\}$, the dual lattice. Note that for any $x \in \Gamma$ and $y \in \Gamma^\star$ we have that $\langle x,y\rangle \in \mathbb{Z}$ and that if $\{b_i\}_{i=1}^d$ is an orthonormal basis then $\Gamma = \Gamma^\star$. The torus $\mathbb{R}^d/\Gamma$ is a (flat) compact Riemannian manifold. The set of eigenvalues of the Laplace–Beltrami operator is

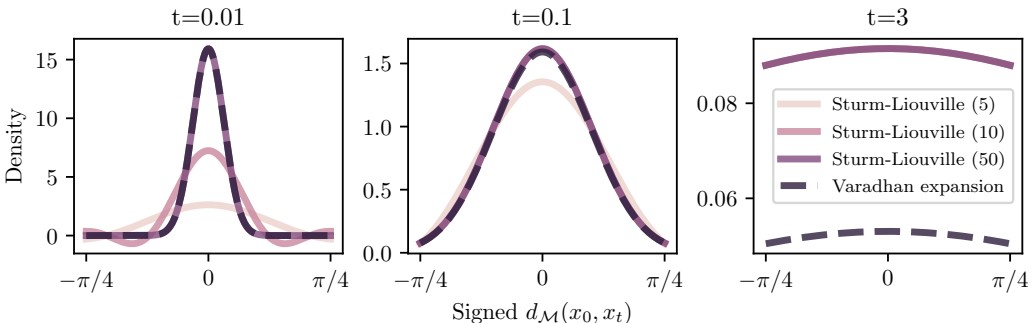

Figure S1: Slice of heat kernel $p_{t|0}(x_t|x_0)$ on $\mathbb{S}^2$ for different approximations.

given by $\{-4\pi^2\|y\|^2 : y \in \Gamma^\star\}$. The eigenfunctions of the Laplace–Beltrami operator are given by $\{x \mapsto \sin(2\pi\langle x, y\rangle) : y \in \Gamma^\star\}$ and $\{x \mapsto \cos(2\pi\langle x, y\rangle) : y \in \Gamma^\star\}$.

**The case of the sphere** Next, we investigate the case of the $d$-dimensional sphere (see Saloff-Coste, 1994). The set of eigenvalues of the Laplace–Beltrami operator is given by $\{-k(k+d-1) : k \in \mathbb{N}\}$. Note that $\lambda_k = k(k + d - 1)$ has multiplicity $d_k = (k + d - 2)!/\{(d - 1)!k\}(2k + d - 1)$. The eigenfunctions of the Laplace–Beltrami operator are known as the spherical harmonics and can be defined in terms of Legendre polynomials. When investigating the heat kernel on the $d$-dimensional sphere, we are interested in the product $(x, y) \mapsto \sum_{\phi \in \Phi_n} \phi(x)\phi(y)$, where $\Phi_n$ is the set of eigenfunctions associated with the eigenvalue $\lambda_n$ for $n \in \mathbb{N}$. This function can be described using the Gegenbauer polynomials (see Atkinson and Han, 2012, Theorem 2.9). More precisely, we have that for any $n \in \mathbb{N}$ and $x, y \in \mathbb{S}^d$

$$G_n(x, y) = \sum_{\phi \in \Phi_n} \phi(x)\phi(y)$$
$$= n!\Gamma((d-1)/2) \sum_{k=0}^{\lfloor n/2 \rfloor} (-1)^k(1 - \langle x, y\rangle^2)\langle x, y\rangle^{n-2k}/(4^k k!(n - 2k)!\Gamma(k + (d - 1)/2)),$$

where here $\Gamma : \mathbb{R}_+ \to \mathbb{R}$ is given for any $v > 0$ by $\Gamma(v) = \int_0^{+\infty} t^{v-1}\mathrm{e}^{-t}\mathrm{d}t$. In the special case where $d = 1$, then the heat kernel coincide with the wrapped Gaussian density and can be easily evaluated.

**Heat kernel on compact Riemannian manifolds.** We recall that in the case of compact manifolds the heat kernel is given by the Sturm–Liouville decomposition Chavel (1984) given for any $t > 0$ and $x, y \in \mathcal{M}$ by

$$p_{t|0}(y|x) = \sum_{j \in \mathbb{N}} \mathrm{e}^{-\lambda_j t}\phi_j(x)\phi_j(y), \tag{S8}$$

where the convergence occurs in $\mathrm{L}^2(p_{\mathrm{ref}} \otimes p_{\mathrm{ref}})$, $(\lambda_j)_{j \in \mathbb{N}}$ and $(\phi_j)_{j \in \mathbb{N}}$ are the eigenvalues, respectively the eigenvectors, of $-\Delta_{\mathcal{M}}$ in $\mathrm{L}^2(p_{\mathrm{ref}})$ (see Saloff-Coste, 1994, Section 2). When the eigenvalues and eigenvectors are known, we approximate the logarithmic gradient of $p_{t|0}$ by truncating the sum in (S8) with $J \in \mathbb{N}$ terms. Another possibility to approximate $\nabla \log p_{t|0}$ is to rely on the so-called Varadhan approximation, see Sec. 3.4, which is valid for small $t > 0$. Fig. S1 illustrates these different approximations of the heat kernel and Table 1 compares the different loss functions.

Table 1: Riemannian score matching losses.

| Loss | Approximation | Loss function | Unbiased | Consistent | Variance |
|------|---------------|---------------|----------|------------|----------|
| $\ell_{t|0}$ (DSM) | Truncation (7) | $\frac{1}{2}\mathbb{E}\left[\|s(\mathbf{X}_t) - S_{J,t}(\mathbf{X}_0, \mathbf{X}_t)\|^2\right]$ | ✗ | ✓$(J \to \infty)$ | 0 |
| | Varhadan (8) | $\frac{1}{2}\mathbb{E}\left[\|s(\mathbf{X}_t) - \log_{\mathbf{X}_t}(\mathbf{X}_0)/t\|^2\right]$ | ✗ | ✓$(t \to 0)$ | 0 |
| $\ell_{t|s}$ (DSM) | Varhadan (8) | $\frac{1}{2}\mathbb{E}\left[\|s(\mathbf{X}_t) - \log_{\mathbf{X}_t}(\mathbf{X}_s)/(t - s)\|^2\right]$ | ✗ | ✓$(t \to s)$ | 0 |
| $\ell_t^{\mathrm{im}}$ (ISM) | Deterministic | $\mathbb{E}\left[\frac{1}{2}\|s(\mathbf{X}_t)\|^2 + \mathrm{div}(s)(\mathbf{X}_t)\right]$ | ✓ | ✓ | 0 |
| | Stochastic | $\mathbb{E}\left[\frac{1}{2}\|s(\mathbf{X}_t)\|^2 + \varepsilon^\top \partial s(\mathbf{X}_t)\varepsilon\right]$ | ✓ | ✓ | $2\|\partial s\|_F$ |

## G  Predictor-corrector schemes

In this section, we present a predictor-corrector scheme, adapting the techniques of Allgower and Georg (2012); Song et al. (2021) to the manifold setting. Changes between Algorithm 1, Algorithm 2 and Algorithm 3, Algorithm 4 are highlighted in red. Let $t \in [0, T]$, $\gamma > 0$ and $k = \lfloor t/\gamma \rfloor$. We remark that Algorithm 3 (Line 11) corresponds to the recursion associated with $(X_j^{t,\gamma})_{j \in \mathbb{N}}$ such that for any $j \in \mathbb{N}$

$$X_{j+1}^{t,\gamma} = \exp_{X_j^{t,\gamma}}[\tfrac{\gamma}{2} \nabla \log p_{T-k\gamma}(X_j^{t,\gamma}) + \sqrt{\gamma} Z_{j+1}],$$

where $\{\bar{Z}_j\}_{j \in \mathbb{N}}$ is a family of i.i.d Gaussian random variables with zero mean and identity covariances matrix in $\mathbb{R}^p$ and for any $j \in \mathbb{N}$, $Z_j = \mathrm{P}(X_j^{t,\gamma})\bar{Z}_j$. Note that here $k \in \{0, N-1\}$ is fixed. Letting $\gamma \to 0$, we obtain that under mild assumptions, see (Kuwada, 2012, Theorem 3.1), $(X_j^{t,\gamma})_{j \in \mathbb{N}}$ converges to $(\mathbf{X}_s^t)_{s \geq 0}$ such that

$$\mathrm{d}\mathbf{X}_s^t = \tfrac{1}{2} \nabla \log p_{T-t}(\mathbf{X}_s^t)\mathrm{d}s + \mathrm{d}\mathbf{B}_t^{\mathcal{M}}.$$

We have that $p_{T-t}$ is the invariant measure of $(\mathbf{X}_s^t)_{s \geq 0}$. Hence, the role of the corrector step is to project the distribution back onto $p_{T-t}$ for all times $t \in [0, T]$, see Fig. S2.

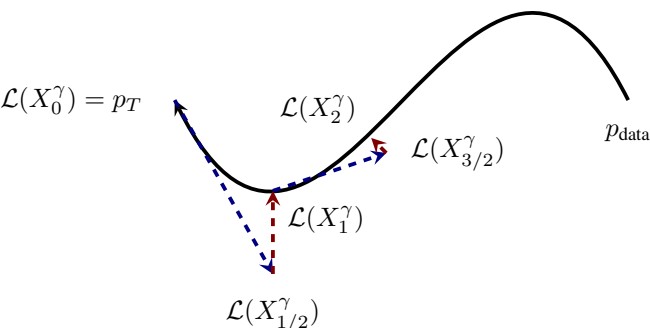

Figure S2: Illustration of the effect of the corrector step on RSGM. The black line corresponds to the dynamics of the noising process $(p_t)_{t \in [0,T]}$. The green dashed lines correspond to the predictor step (going backward in time) and the red dashed lines correspond to the corrector step (projecting back onto the initial dynamics). Note that $\mathcal{L}(X_1^\gamma) \approx p_{T-\gamma}$ and $\mathcal{L}(X_2^\gamma) \approx p_{T-2\gamma}$.

## H  Time-reversal formula: extension to compact Riemannian manifolds

In this section, we provide the proof of Theorem 1. The proof follows the arguments of Cattiaux et al. (2021, Theorem 4.9). We could have also applied the abstract results of Cattiaux et al. (2021, Theorem 5.7) to obtain our results. Note that the time-reversal on manifold could also be obtained by readily extending arguments from Haussmann and Pardoux (1986), however the entropic conditions found by Cattiaux et al. (2021) are more natural when it comes to the study of the Schrödinger Bridge problem. For the interested reader we provide an informal derivation of the time-reversal formula obtained by Haussmann and Pardoux (1986) in App. H.1. The proof of Theorem 1 is given in App. H.2. Finally, we emphasize that García-Zelada and Huguet (2021) have developed a Girsanov theory for stochastic processes defined on compact manifolds with boundary in order to study the Brenier-Schrödinger problem.

---

**Algorithm 3** GRW-c (Geodesic Random Walk with corrector)

---

**Require:** $T, N, X_0^\gamma, b, \sigma, \mathrm{P}$

1: $\gamma = T/N$        ▷ Step-size
2: **for** $k \in \{0, \dots, N-1\}$ **do**
3:      /// PREDICTOR STEP
4:      $\bar{Z}_{k+1/2} \sim \mathrm{N}(0, \mathrm{I}_p)$      ▷ Standard Gaussian in ambient space $\mathbb{R}^p$
5:      $Z_{k+1/2} = \mathrm{P}(X_k^\gamma)\bar{Z}_{k+1/2}$      ▷ Projection in the tangent space $\mathrm{T}_x\mathcal{M}$
6:      $W_{k+1/2} = \gamma b(k\gamma, X_k^\gamma) + \sqrt{\gamma}\sigma(k\gamma, X_k^\gamma)Z_{k+1/2}$      ▷ Euler–Maruyama step on tangent space
7:      $X_{k+1/2}^\gamma = \exp_{X_k^\gamma}[W_{k+1/2}]$      ▷ Geodesic projection onto $\mathcal{M}$
8:      /// CORRECTOR STEP
9:      $\bar{Z}_{k+1} \sim \mathrm{N}(0, \mathrm{I}_p)$      ▷ Standard Gaussian in ambient space $\mathbb{R}^p$
10:      $Z_{k+1} = \mathrm{P}(X_{k+1/2}^\gamma)\bar{Z}_{k+1}$      ▷ Projection in the tangent space $\mathrm{T}_x\mathcal{M}$
11:      $W_{k+1} = \frac{\gamma}{2}b(k\gamma, X_{k+1/2}^\gamma) + \sqrt{\gamma}\sigma(k\gamma, X_{k+1/2}^\gamma)Z_{k+1}$      ▷ Euler–Maruyama step on tangent space
12:      $X_{k+1}^\gamma = \exp_{X_{k+1/2}^\gamma}[W_{k+1}]$      ▷ Geodesic projection onto $\mathcal{M}$
13: **end for**
14: **return** $\{X_k^\gamma\}_{k=0}^N$

---

---

**Algorithm 4** RSGM-c (Riemannian Score-Based Generative Model with corrector)

---

**Require:** $\varepsilon, T, N, \{X_0^m\}_{m=1}^M, \mathrm{loss}, \mathbf{s}, \theta_0, N_{\mathrm{iter}}, p_{\mathrm{ref}}, \mathrm{P}$

1: /// TRAINING ///
2: **for** $n \in \{0, \dots, N_{\mathrm{iter}} - 1\}$ **do**
3:      $X_0 \sim (1/M)\sum_{m=1}^M \delta_{X_0^m}$      ▷ Random mini-batch from dataset
4:      $t \sim U([\varepsilon, T])$      ▷ Uniform sampling between $\varepsilon$ and $T$
5:      $\mathbf{X}_t = \mathrm{GRW}(t, N, X_0, 0, \mathrm{Id}, \mathrm{P})$      ▷ Approximate forward diffusion with Algorithm 1
6:      $\ell(\theta_n) = \ell_t(T, N, X_0, \mathbf{X}_t, \mathrm{loss}, \mathbf{s}_{\theta_n})$      ▷ Compute score matching loss from Table 2
7:      $\theta_{n+1} = \mathtt{optimizer\_update}(\theta_n, \ell(\theta_n))$      ▷ ADAM optimizer step
8: **end for**
9: $\theta^\star = \theta_{N_{\mathrm{epoch}}}$
10: /// SAMPLING ///
11: $Y_0 \sim p_{\mathrm{ref}}$      ▷ Sample from uniform distribution
12: $b_\theta^\star(t, x) = \mathbf{s}_{\theta^\star}(T - t, x)$ for any $t \in [0, T], x \in \mathcal{M}$      ▷ Reverse process drift
13: $\{Y_k\}_{k=0}^N = \mathrm{GRW\text{-}c}(T, N, Y_0, b_{\theta^\star}, \mathrm{Id}, \mathrm{P})$      ▷ Approximate reverse diffusion with Algorithm 3
14: **return** $\theta^\star, \{Y_k\}_{k=0}^N$

---

### H.1 Informal derivation

In this section, we provide a non-rigorous derivation of Theorem 1 following the approach of Haussmann and Pardoux (1986). Let $(\mathbf{X}_t)_{t \in [0,T]}$ be a continuous process such that for any $f \in \mathrm{C}^2(\mathcal{M})$ we have that $(\mathbf{M}_t^{\mathbf{X},f})_{t \in [0,T]}$ is a $\mathbf{X}$-martingale where for any $t \in [0, T]$

$$\mathbf{M}_t^{\mathbf{X},f} = f(\mathbf{X}_t) - \int_0^t \{\langle b(\mathbf{X}_s), \nabla f(\mathbf{X}_s)\rangle + \tfrac{1}{2}\Delta_\mathcal{M} f(\mathbf{X}_s)\}\mathrm{d}s. \tag{S9}$$

Let $(\mathbf{Y}_t)_{t \in [0,T]} = (\mathbf{X}_{T-t})_{t \in [0,T]}$. Our goal is to show that for any $f \in \mathrm{C}^2(\mathcal{M})$, $(\mathbf{M}_t^{\mathbf{Y},f})_{t \in [0,T]}$ is a $\mathbf{Y}$-martingale where for any $t \in [0, T]$

$$\mathbf{M}_t^{\mathbf{Y},f} = f(\mathbf{Y}_t) - \int_0^t \{\langle -b(\mathbf{Y}_s) + \nabla \log p_{T-s}(\mathbf{Y}_s), \nabla f(\mathbf{Y}_s)\rangle + \tfrac{1}{2}\Delta_\mathcal{M} f(\mathbf{Y}_s)\}\mathrm{d}s.$$

Note that here we implicitly assume that for any $t \in [0, T]$, $\mathbf{X}_t$ admits a smooth positive density w.r.t. $p_{\mathrm{ref}}$ denoted $p_t$. In other words, we want to show that for any $g \in \mathrm{C}^2(\mathcal{M})$ and $s, t \in [0, T]$ with $t \geq s$ we have

$$\mathbb{E}[g(\mathbf{Y}_s)(f(\mathbf{Y}_t) - f(\mathbf{Y}_s))] \tag{S10}$$
$$= \mathbb{E}[g(\mathbf{Y}_s)\int_s^t \{\langle -b(\mathbf{Y}_u) + \nabla \log p_{T-u}(\mathbf{Y}_u), \nabla f(\mathbf{Y}_u)\rangle + \tfrac{1}{2}\Delta_\mathcal{M} f(\mathbf{Y}_u)\}\mathrm{d}u].$$

We introduce the infinitesimal generator $\mathcal{A} : \mathrm{C}^2(\mathcal{M}) \to \mathrm{C}(\mathcal{M})$ given for any $f \in \mathrm{C}^2(\mathcal{M})$ and $x \in \mathcal{M}$ by

$$\mathcal{A}(f)(x) = \langle b(x), \nabla f(x)\rangle + \tfrac{1}{2}\Delta_\mathcal{M} f(x).$$

Similarly, we introduce the infinitesimal generator $\tilde{\mathcal{A}} : [0, T] \times \mathrm{C}^2(\mathcal{M}) \to \mathrm{C}(\mathcal{M})$ given for any $f \in \mathrm{C}^2(\mathcal{M}), t \in [0, T]$ and $x \in \mathcal{M}$ by

$$\tilde{\mathcal{A}}(t, f)(x) = \langle -b(x) + \nabla \log p_{T-t}(x), \nabla f(x)\rangle + \tfrac{1}{2}\Delta_\mathcal{M} f(x).$$

With these notations, (S11) can be written as follows: we want to show that for any $g \in \mathrm{C}^2(\mathcal{M})$ and $s, t \in [0, T]$ with $t \geq s$ we have

$$\mathbb{E}[g(\mathbf{Y}_s)(f(\mathbf{Y}_t) - f(\mathbf{Y}_s))] = \mathbb{E}[g(\mathbf{Y}_s) \int_s^t \tilde{\mathcal{A}}(u, \mathbf{Y}_u) \mathrm{d}u]. \tag{S11}$$

The rest of this section follows the first part of the proof of Haussmann and Pardoux (1986, Theorem 2.1). Let $t, s \in [0, T]$ with $t \geq s$. We have

$$\begin{aligned}
\mathbb{E}[g(\mathbf{Y}_s)(f(\mathbf{Y}_t) - f(\mathbf{Y}_s))] &= \mathbb{E}[g(\mathbf{X}_{T-s})(f(\mathbf{X}_{T-t}) - f(\mathbf{X}_{T-t}))] \\
&= \mathbb{E}[\mathbb{E}[g(\mathbf{X}_{T-s})|\mathbf{X}_{T-t}]f(\mathbf{X}_{T-t})] - \mathbb{E}[g(\mathbf{X}_{T-s})f(\mathbf{X}_{T-s})] \\
&= \mathbb{E}[v(T-t, \mathbf{X}_{T-t})f(\mathbf{X}_{T-t})] - \mathbb{E}[v(T-s, \mathbf{X}_{T-s})f(\mathbf{X}_{T-s})],
\end{aligned} \tag{S12}$$

with $v : [0, T-s] \times \mathcal{M} \to \mathbb{R}$ given for any $u \in [0, T-s]$ and $x \in \mathcal{M}$ by $v(u, x) = \mathbb{E}[g(\mathbf{X}_{T-s})|\mathbf{X}_u = x]$. We have that $v$ satisfies the backward Kolmogorov equation, i.e. we have for any $u \in [0, T-s]$ and $x \in \mathcal{M}$

$$\partial_u v(u, x) = -\mathcal{A}v(u, x). \tag{S13}$$

Note that it is not trivial to show that $v$ is regular enough to satisfy the backward Kolmogorov equation. In this informal derivation, we assume that $v$ is regular enough and will provide a different rigorous proof of the time-reversal formula in App. H.2. However, note that it is possible to show that $v$ indeed satisfies the backward Kolmogorov equation by adapting arguments from Haussmann and Pardoux (1986) to the manifold framework.

Let $h : [0, T-s] \times \mathcal{M} \to \mathbb{R}$ given for any $u \in [0, T-s]$ and $x \in \mathcal{M}$ by $h(u, x) = v(u, x)f(x)$. Using (S13), we have for any $u \in [0, T-s]$ and $x \in \mathcal{M}$

$$\begin{aligned}
\partial_u h(u, x) + \mathcal{A}h(u, x) &= f(x)\partial_u v(u, x) + f(x)\mathcal{A}v(u, x) + v(u, x)\mathcal{A}f(x) + \langle \nabla f(x), \nabla v(u, x) \rangle \\
&= v(u, x)\mathcal{A}f(x) + \langle \nabla f(x), \nabla v(u, x) \rangle. \tag{S14}
\end{aligned}$$

In addition, using the divergence theorem (see Lee, 2018, p.51), we have for any $u \in [0, T-s]$

$$\begin{aligned}
\mathbb{E}[\langle \nabla f(\mathbf{X}_u), \nabla v(u, \mathbf{X}_u) \rangle] &= \int_{\mathcal{M}} \langle \nabla f(x_u), \nabla v(u, x_u) p_u(x_u) \rangle \mathrm{d}p_{\mathrm{ref}}(x_u) \\
&= -\int_{\mathcal{M}} v(u, x_u) \mathrm{div}(p_u \nabla f)(x_u) \mathrm{d}p_{\mathrm{ref}}(x_u) \\
&= -\int_{\mathcal{M}} v(u, x_u) \Delta_{\mathcal{M}} f(x_u) p_u(x_u) \mathrm{d}p_{\mathrm{ref}}(x_u) \\
&\quad - \int_{\mathcal{M}} v(u, x_u) \langle \nabla f(x_u), \nabla \log p_u(x_u) \rangle p_u(x_u) \mathrm{d}p_{\mathrm{ref}}(x_u) \\
&= -\mathbb{E}[v(u, \mathbf{X}_u) \Delta_{\mathcal{M}} f(\mathbf{X}_u)] - \mathbb{E}[v(u, \mathbf{X}_u) \langle \nabla f(\mathbf{X}_u), \nabla \log p_u(\mathbf{X}_u) \rangle].
\end{aligned}$$

Therefore, using this result and (S14) we get that for any $u \in [0, T-s]$

$$\begin{aligned}
\mathbb{E}[\partial_u h(u, \mathbf{X}_u) + \mathcal{A}h(u, \mathbf{X}_u)] &= \mathbb{E}[v(u, \mathbf{X}_u)\{\langle b(\mathbf{X}_u) - \nabla \log p_u(\mathbf{X}_u), \nabla f(\mathbf{X}_u) \rangle - \tfrac{1}{2}\Delta_{\mathcal{M}} f(\mathbf{X}_u)\}] \\
&= -\mathbb{E}[v(u, \mathbf{X}_u)\tilde{\mathcal{A}}(T-u, f)(\mathbf{X}_u)].
\end{aligned}$$

Combining this result and (S9) and that for any $u \in [0, T-s]$ and $x \in \mathcal{M}$, $v(u, x) = \mathbb{E}[g(\mathbf{X}_{T-s})|\mathbf{X}_u = x]$ we get

$$\begin{aligned}
\mathbb{E}[v(T-t, \mathbf{X}_{T-t})f(\mathbf{X}_{T-t})] - \mathbb{E}[v(T-s, \mathbf{X}_{T-s})f(\mathbf{X}_{T-s})] &= \mathbb{E}[h(T-t, \mathbf{X}_{T-t}) - h(T-s, \mathbf{X}_{T-s})] \\
&= \int_{T-t}^{T-s} \mathbb{E}[v(u, \mathbf{X}_u)\tilde{\mathcal{A}}(T-u, \mathbf{X}_u)]\mathrm{d}u \\
&= \mathbb{E}[g(\mathbf{X}_{T-s}) \int_{T-t}^{T-s} \tilde{\mathcal{A}}(T-u, \mathbf{X}_u)\mathrm{d}u].
\end{aligned}$$

Using this result, (S12) and the change of variable $u \mapsto T - u$ we obtain

$$\mathbb{E}[g(\mathbf{Y}_s)(f(\mathbf{Y}_t) - f(\mathbf{Y}_s))] = \mathbb{E}[g(\mathbf{X}_{T-s}) \int_{T-t}^{T-s} \tilde{\mathcal{A}}(T-u, \mathbf{X}_u)\mathrm{d}u] = \mathbb{E}[g(\mathbf{Y}_s) \int_s^t \tilde{\mathcal{A}}(u, \mathbf{Y}_u)\mathrm{d}u].$$

Hence, (S11) holds and we have proved Theorem 1. Again, we emphasize that in order to make the proof completely rigourous one needs to derive regularity properties of $v$.

## H.2 Proof of Theorem 1

In this section, we follow another approach to prove the time-reversal formula. We are going to use the integration by part formula of Cattiaux et al. (2021, Theorem 3.17) in a similar spirit as Cattiaux et al. (2021, Theorem 4.9) in the Euclidean setting. In order to adapt arguments from Cattiaux et al. (2021) to our Riemannian setting, we use the Nash embedding theorem in order to embed our processes in a Euclidean space and leverage tools from Girsanov theory. The rest of the section is organized as follows. First in App. H.2.1, we recall basic properties of infinitesimal generators and recall the integration by part formula of Cattiaux et al. (2021, Theorem 3.17). Then in App. H.2.2, we extend some Girsanov theory to compact Riemannian manifolds using the Nash embedding theorem. We conclude the proof in App. H.2.3.

### H.2.1 Diffusion processes and integration by part formula

In this section, we state a simplified version of Cattiaux et al. (2021, Theorem 3.17) for Markov continuous path (probability) measure on Polish spaces. Let $(\mathsf{X}, \mathcal{X})$ be a Polish space. We say that $\mathbb{P}$ is a path measure if $\mathbb{P} \in \mathcal{P}(\mathrm{C}([0,T], \mathsf{X}))$. Let $(\mathbf{X}_t)_{t \in [0,T]}$ with distribution $\mathbb{P}$. We denote $(\mathcal{F}_t)_{t \in [0,T]}$ the filtration such that for any $t \in [0,T]$, $\mathcal{F}_t = \sigma(\mathbf{X}_s, \ s \in [0,t])$. Let $(\mathbf{M}_t)_{t \in [0,T]}$ be a Polish-valued stochastic process. We say that $(\mathbf{M}_t)_{t \in [0,T]}$ is a $\mathbb{P}$-local martingale if it is a local martingale w.r.t. the filtration $(\mathcal{F}_t)_{t \in [0,T]}$. A function $u : [0,T] \times \mathsf{X} \to \mathbb{R}$ is said to be in the domain of the extended generator of $\mathbb{P}$ if there exists a process $(\bar{\mathcal{A}}_\mathbb{P} u(t, \mathbf{X}_{[0,t]}))_{t \in [0,T]}$ such that:

(a) $(\bar{\mathcal{A}}_\mathbb{P} u(t, \mathbf{X}_{[0,t]}))_{t \in [0,T]}$ is adapted w.r.t. $(\mathcal{F}_t)_{t \in [0,T]}$.

(b) $\int_0^T |\bar{\mathcal{A}}_\mathbb{P} u(t, \mathbf{X}_{[0,t]})| \mathrm{d}t < +\infty$, $\mathbb{P}$-a.s.

(c) The process $(\mathbf{M}_t)_{t \in [0,T]}$ is a $\mathbb{P}$-local martingale, where for any $t \in [0,T]$

$$\mathbf{M}_t = u(t, \mathbf{X}_t) - u(0, \mathbf{X}_0) - \int_0^t \bar{\mathcal{A}}_\mathbb{P} u(s, \mathbf{X}_{[0,s]}) \mathrm{d}s.$$

The domain of the extended generator is denoted $\mathrm{dom}(\bar{\mathcal{A}}_\mathbb{P})$. We say that $(u, v)$ with $u, v : [0,T] \times \mathsf{X} \to \mathbb{R}$ is in the domain of the carré du champ if $u, v, uv \in \mathrm{dom}(\bar{\mathcal{A}}_\mathbb{P})$. In this case, we define the carré du champ $\bar{\Upsilon}_\mathbb{P}$ as

$$\bar{\Upsilon}_\mathbb{P}(u, v) = \bar{\mathcal{A}}_\mathbb{P}(uv) - \bar{\mathcal{A}}_\mathbb{P}(u)v - \bar{\mathcal{A}}_\mathbb{P}(v)u.$$

Note that if $\mathsf{X} = \mathcal{M}$ is a Riemannian manifold, $\mathrm{C}^2(\mathcal{M}) \subset \mathrm{dom}(\bar{\mathcal{A}}_\mathbb{P})$ and for any $u \in \mathrm{C}^2(\mathcal{M})$ $\bar{\mathcal{A}}_\mathbb{P}(u) = \langle \nabla u, X \rangle + \frac{1}{2} \Delta_\mathcal{M} u$ with $X \in \Gamma(\mathrm{T}\mathcal{M})$ then we have that $\mathrm{C}^2(\mathcal{M}) \times \mathrm{C}^2(\mathcal{M}) \subset \mathrm{dom}(\bar{\Upsilon}_\mathbb{P})$ and for any $u, v \in \mathrm{C}^2(\mathcal{M})$, $\bar{\Upsilon}_\mathbb{P}(u, v) = \langle \nabla u, \nabla v \rangle$. Assume that there exists $\mathcal{U}_\mathbb{P} \subset \mathrm{dom}(\bar{\mathcal{A}}_\mathbb{P}) \cap \mathrm{C}_b(\mathsf{X})$ such that $\mathcal{U}_\mathbb{P}$ is an algebra. We denote $\mathcal{U}_{\mathbb{P},2}$ such that

$$\mathcal{U}_{\mathbb{P},2} = \{u \in \mathcal{U}_\mathbb{P} \ : \ \bar{\mathcal{A}}_\mathbb{P} u \in \mathrm{L}^2(\mathbb{P}), \ \bar{\Upsilon}_\mathbb{P}(u, u) \in \mathrm{L}^1(\mathbb{P})\}.$$

Finally we denote $R(\mathbb{P})$ the time-reverse path measure, i.e. for any $\mathsf{A} \in \mathcal{B}(\mathrm{C}([0,T], \mathsf{X}))$ we have $R(\mathbb{P})(\mathsf{A}) = \mathbb{P}(R(\mathsf{A}))$, where $R(\mathsf{A}) = \{t \mapsto \omega_{T-t} \ : \ \omega \in \mathsf{A}\}$. In what follows, we assume $\mathbb{P}$ is Markov. It is well-known, see (Léonard et al., 2014, Theorem 1.2) for instance, that in this case $R(\mathbb{P})$ is also Markov. In addition, since $\mathbb{P}$ is Markov, for any $u \in \mathrm{dom}(\bar{\mathcal{A}}_\mathbb{P})$ and $t \in [0,T]$ there exists $\mathcal{A}_\mathbb{P}$ such that $\bar{\mathcal{A}}_\mathbb{P} u(t, \mathbf{X}_{[0,t]}) = \mathcal{A}_\mathbb{P} u(t, \mathbf{X}_t)$ with $\mathcal{A}_\mathbb{P} u : [0,T] \times \mathsf{X} \to \mathbb{R}$. Similarly, we define $\Upsilon_\mathbb{P}(u, v) : [0,T] \times \mathsf{X} \to \mathbb{R}$ from $\bar{\Upsilon}_\mathbb{P}(u, v)$.

We are now ready to state the integration by part formula, (Cattiaux et al., 2021, Theorem 3.17).

**Theorem S11.** *Let* $u, v \in \mathcal{U}_{\mathbb{P},2}$. *The following hold:*

*(a) If* $u \in \mathrm{dom}(\mathcal{A}_{R(\mathbb{P})})$ *and* $\mathcal{A}_{R(\mathbb{P})} u \in \mathrm{L}^1(\mathbb{P})$ *then for almost any* $t \in [0,T]$

$$\mathbb{E}[\{\mathcal{A}_\mathbb{P} u(t, \mathbf{X}_t) + \mathcal{A}_{R(\mathbb{P})} u(T - t, \mathbf{X}_t)\} v(\mathbf{X}_t) + \Upsilon_\mathbb{P}(u, u)(t, \mathbf{X}_t)] = 0.$$

*(b) If the following hold:*

  *i)* $\Upsilon_\mathbb{P}(u, v) \in \mathrm{C}([0,T] \times \mathsf{X}, \mathbb{R})$.

  *ii)* $\mathcal{U}_{2,\mathbb{P}}$ *determines the weak convergence of Borel measures.*

*iii) $\mu$ defines a finite measure on $[0, T] \times \mathsf{X}$ where for any $\omega \in \bar{\mathcal{U}}_{2,\mathbb{P}}$ we have*

$$\mu[\omega] = \mathbb{E}[\int_0^T \Upsilon_{\mathbb{P}}(u, \omega_t)(t, \mathbf{X}_t) \mathrm{d}t],$$

*where $\bar{\mathcal{U}}_{2,\mathbb{P}} = \{\omega \in \mathrm{C}([0, T] \times \mathsf{X}, \mathbb{R}) : \omega(t, \cdot) \in \mathcal{U}_{2,\mathbb{P}} \text{ for any } t \in [0, T]\}$.*

*Then $u \in \mathrm{dom}(\mathcal{A}_{R(\mathbb{P})})$ and $\mathcal{A}_{R(\mathbb{P})} u \in \mathrm{L}^1(\mathbb{P})$.*

Note that this theorem is a simplified version of Cattiaux et al. (2021, Theorem 3.17) where we restrict ourselves to the case of Markov path measures. In what follows, we wish to apply Theorem S11 to diffusion processes on manifolds. To do so, we will verify that under a finite entropy assumption, the conditions $u \in \mathrm{dom}(\mathcal{A}_{R(\mathbb{P})})$ and $\mathcal{A}_{R(\mathbb{P})} u \in \mathrm{L}^1(\mathbb{P})$ are fullfilled for a class of regular functions $u$. These integrability results are obtained using Girsanov theory.

### H.2.2 Girsanov theory on compact Riemannian manifolds

In this section, we will consider two types of martingale problems: one on Euclidean spaces and one on the compact Riemannian manifold $\mathcal{M}$. Let $\mathbb{P} \in \mathcal{P}(\mathrm{C}([0, T], \mathbb{R}^p))$. We say that $\mathbb{P}$ satisfies the (Euclidean) martingale problem with infinitesimal generator $\mathcal{A} : [0, T] \times \mathrm{C}^2(\mathbb{R}^p) \times \mathbb{R}^p \to \mathbb{R}$ if for any $u \in \mathrm{C}_c^2(\mathbb{R}^p)$, $(\mathbf{M}_t)_{t \in [0,T]}$ is a $\mathbb{P}$-martingale where for any $t \in [0, T]$ we have

$$\mathbf{M}_t = \mathbf{M}_0 + \int_0^t \mathcal{A}(t, u)(\mathbf{X}_s) \mathrm{d}s,$$

where $(\mathbf{X}_t)_{t \in [0,T]}$ has distribution $\mathbb{P}$ and $\int_0^T |\mathcal{A}(t, u)(\mathbf{X}_s) \mathrm{d}t| < +\infty$, $\mathbb{P}$-a.s. Let $\mathbb{P} \in \mathcal{P}(\mathrm{C}([0, T], \mathcal{M}))$. We say that $\mathbb{P}$ satisfies the (Riemannian) martingale problem with infinitesimal generator $\tilde{\mathcal{A}} : [0, T] \times \mathrm{C}^2(\mathcal{M}) \times \mathcal{M} \to \mathbb{R}$ if for any $u \in \mathrm{C}^2(\mathcal{M})$, $(\mathbf{M}_t)_{t \in [0,T]}$ is a $\mathbb{P}$-martingale where for any $t \in [0, T]$ we have

$$\mathbf{M}_t = \mathbf{M}_0 + \int_0^t \tilde{\mathcal{A}}(t, u)(\mathbf{X}_s) \mathrm{d}s,$$

where $(\mathbf{X}_t)_{t \in [0,T]}$ has distribution $\mathbb{P}$ and $\int_0^T |\tilde{\mathcal{A}}(t, u)(\mathbf{X}_s) \mathrm{d}t| < +\infty$, $\mathbb{P}$-a.s. We now prove the following theorem.

**Proposition S12.** *Let $\mathbb{Q}$ be the path measure of a Brownian motion on $\mathcal{M}$. Let $\mathbb{P}$ be a Markov path measure on $\mathrm{C}([0, T], \mathcal{M})$ such that $\mathrm{KL}(\mathbb{P}|\mathbb{Q}) < +\infty$. Then there exists $\beta$ such that for any $t \in [0, T]$ and $x \in \mathcal{M}$, $\beta(t, x) \in \mathrm{T}_x\mathcal{M}$ and we have that $\mathbb{P}$ satisfies the martingale problem with infinitesimal generator $\mathcal{A}$ where for any $t \in [0, T]$, $u \in \mathrm{C}^2(\mathcal{M})$ and $x \in \mathcal{M}$ we have*

$$\mathcal{A}(t, u)(x) = \langle \beta(t, x), \nabla u(x) \rangle + \tfrac{1}{2} \Delta_\mathcal{M} u(x).$$

*In addition, we have that*

$$\mathrm{KL}(\mathbb{P}|\mathbb{Q}) = \mathrm{KL}(\mathbb{P}_0|\mathbb{Q}_0) + \tfrac{1}{2} \int_0^T \mathbb{E}[\|\beta(t, \mathbf{X}_t)\|^2] \mathrm{d}t,$$

*where $(\mathbf{X}_t)_{t \in [0,T]}$ has distribution $\mathbb{P}$.*

*Proof.* First, we extend $(\mathbf{B}_t^\mathcal{M})_{t \in [0,T]}$ to $\mathbb{R}^p$ using the Nash embedding theorem (see Gunther, 1991). $(\mathbf{B}_t^\mathcal{M})_{t \in [0,T]}$ can be seen as a process on $\mathbb{R}^p$ (for some $p \in \mathbb{N}$) which satisfies in a weak sense

$$\mathrm{d}\mathbf{B}_t^\mathcal{M} = \sum_{i=1}^p \mathrm{P}_i(\mathbf{B}_t^\mathcal{M}) \circ \mathrm{d}\mathbf{B}_t^i = P(\mathbf{B}_t^\mathcal{M}) \circ \mathrm{d}\mathbf{B}_t,$$

where $(\mathbf{B}_t)_{t \in [0,T]}$ is a $p$-dimensional Brownian motion and $\mathrm{P} \in \mathrm{C}^\infty(\mathbb{R}^p, \mathbb{R}^{p \times p})$ is such that for any $x \in \mathcal{M}$, $\mathrm{P}(x)$ is the projection onto $\mathrm{T}_x\mathcal{M}$ and for any $i \in \{1, \ldots, p\}$, $\mathrm{P}_i \in \mathrm{C}^\infty(\mathbb{R}^p, \mathbb{R}^p)$ with $\mathrm{P}_i = \mathrm{P}e_i$ where $\{e_j\}_{j=1}^p$ is the canonical basis of $\mathbb{R}^p$. We refer to App. C.1 for more details on the projection operator and its extension to $\mathbb{R}^p$. Using the link between Stratanovitch and Itô integral, there exists $\bar{b} \in \mathrm{C}^\infty(\mathbb{R}^p, \mathbb{R}^p)$ such that $(\mathbf{B}_t^\mathcal{M})_{t \in [0,T]}$ can be seen as a process on $\mathbb{R}^p$ which satisfies in a weak sense

$$\mathrm{d}\mathbf{B}_t^\mathcal{M} = \bar{b}(\mathbf{B}_t^\mathcal{M}) \mathrm{d}t + \mathrm{P}(\mathbf{B}_t^\mathcal{M}) \mathrm{d}\mathbf{B}_t,$$

where $\bar{b}$ is given in (S2) and satisfies $\mathrm{P}\bar{b}(x) = 0$ for any $x \in \mathcal{M}$, see the remark following (S2). For any $u, v \in \mathrm{C}_c^2(\mathcal{M})$, we consider $\bar{u}, \bar{v}$ extensions to $\mathrm{C}_c^2(\mathbb{R}^p)$ and we have for any $s, t \in [0, T]$

$$\mathbb{E}[\bar{v}(\mathbf{B}_s^\mathcal{M}) \int_s^t \tfrac{1}{2} \Delta_\mathcal{M} u(\mathbf{B}_u^\mathcal{M}) \mathrm{d}u]$$
$$= \mathbb{E}[\bar{v}(\mathbf{B}_s^\mathcal{M}) \int_s^t \{\langle \nabla \bar{u}(\mathbf{B}_w^\mathcal{M}), \bar{b}(\mathbf{B}_w^\mathcal{M}) \rangle + \tfrac{1}{2} \langle \mathrm{P}(\mathbf{B}_w^\mathcal{M}), \nabla^2 \bar{u}(\mathbf{B}_w^\mathcal{M}) \rangle \} \mathrm{d}w].$$

In particular, we get that for any $x \in \mathcal{M}$, $\Delta_{\mathcal{M}}u(x) = 2\langle \nabla \bar{u}(x), \bar{b}(x)\rangle + \langle \mathrm{P}(x), \nabla^2 \bar{u}(x)\rangle$. Note that $(\mathbf{B}_t^{\mathcal{M}})_{t\in[0,T]}$ (seen as a process on $\mathbb{R}^p$) satisfies the condition (U) in Léonard (2012b), i.e. uniqueness of the trajectories given an initial condition. Therefore applying (Léonard, 2012b, Theorem 2.1), (Cattiaux et al., 2021, Claim 4.5), there exists $\bar{\beta} : [0,T] \times \mathbb{R}^p \to \mathbb{R}^p$ such that

$$\mathrm{KL}\left(\mathbb{P}|\mathbb{Q}\right) = \mathrm{KL}\left(\mathbb{P}_0|\mathbb{Q}_0\right) + \tfrac{1}{2}\int_0^T \mathbb{E}[\|\mathrm{P}(\mathbf{X}_t)\bar{\beta}(t,\mathbf{X}_t)\|^2]\mathrm{d}t. \tag{S15}$$

In addition, $\mathbb{P}$ (seen as a process on $\mathbb{R}^p$) satisfies a martingale problem with infinitesimal generator $\bar{\mathcal{A}} : [0,T] \times \mathrm{C}_c^2(\mathbb{R}^p) \times \mathbb{R}^p \to \mathbb{R}$ such that for any $t \in [0,T]$, $\bar{u} \in \mathrm{C}_c^2(\mathbb{R}^p)$ and $x \in \mathbb{R}^p$

$$\bar{\mathcal{A}}(t,\bar{u})(x) = \langle \bar{b}(x) + \mathrm{P}(x)\bar{\beta}(t,x), \nabla \bar{u}(x)\rangle + \tfrac{1}{2}\langle \mathrm{P}(x), \nabla^2 \bar{u}(x)\rangle.$$

Let $\beta : [0,T] \times \mathcal{M}$ such that for any $t \in [0,T]$ and $x \in \mathcal{M}$ we have $\beta(t,x) = \mathrm{P}(x)\bar{\beta}(t,x)$. In particular, we have that for any $x \in \mathcal{M}$, $\beta(t,x) \in \mathrm{T}_x\mathcal{M}$. Let $u \in \mathrm{C}_c^2(\mathcal{M})$ and consider an extension $\bar{u}$ to $\mathrm{C}_c^2(\mathbb{R}^p)$. For any $t \in [0,T]$ and $x \in \mathcal{M}$ we have

$$\begin{aligned}
\bar{\mathcal{A}}(t,\bar{u})(x) &= \langle \bar{b}(x) + \mathrm{P}(x)\bar{\beta}(t,x), \nabla \bar{u}(x)\rangle + \tfrac{1}{2}\langle \mathrm{P}(x), \nabla^2 \bar{u}(x)\rangle \\
&= \langle \beta(t,x), \nabla \bar{u}(x)\rangle + \tfrac{1}{2}\Delta_{\mathcal{M}}u(x) \\
&= \langle \beta(t,x), \nabla u(x)\rangle + \tfrac{1}{2}\Delta_{\mathcal{M}}u(x).
\end{aligned}$$

In particular, we have that $\mathbb{P}$ (seen as a process on $\mathcal{M}$) satisfies a martingale problem with infinitesimal generator $\mathcal{A} : [0,T] \times \mathrm{C}_c^2(\mathcal{M}) \times \mathcal{M} \to \mathbb{R}$ such that for any $t \in [0,T]$, $u \in \mathrm{C}^2(\mathbb{R}^p)$ and $x \in \mathcal{M}$

$$\mathcal{A}(t,\bar{u})(x) = \langle \beta(t,x), \nabla u(x)\rangle + \tfrac{1}{2}\Delta_{\mathcal{M}}u(x).$$

In addition, rewriting (S16) we have

$$\mathrm{KL}\left(\mathbb{P}|\mathbb{Q}\right) = \mathrm{KL}\left(\mathbb{P}_0|\mathbb{Q}_0\right) + \tfrac{1}{2}\int_0^T \mathbb{E}[\|\beta(t,\mathbf{X}_t)\|^2]\mathrm{d}t, \tag{S16}$$

which concludes the proof. $\qquad\square$

We also derive the following useful lemma, which will be used in the proof of convergence of RSGM.

**Corollary S13.** *Assume* **A**1. *Let* $\mathbb{P}^1, \mathbb{P}^2$ *be a Markov path measure on* $\mathrm{C}([0,T], \mathcal{M})$ *with* $\mathbb{P}_0^1 = \mathbb{P}_0^2$. *In addition, assume that there exist* $b_1, b_2 \in \mathrm{C}^\infty([0,T], \mathcal{X}(\mathcal{M}))$ *such that* $(\mathbf{X}_t^1)_{t\in[0,T]}$ *and* $(\mathbf{X}_t^2)_{t\in[0,T]}$ *are associated to* $\mathbb{P}^1$ *and* $\mathbb{P}^2$ *respectively and satisfy weakly* $\mathrm{d}\mathbf{X}_t^i = b_1(t,\mathbf{X}_t^i)\mathrm{d}t + \mathrm{d}\mathbf{B}_t$ *for* $i \in \{1,2\}$. *Then, we have that*

$$\mathrm{KL}(\mathbb{P}^1|\mathbb{P}^2) = \tfrac{1}{2}\int_0^T \mathbb{E}[\|b_1(t,\mathbf{X}_t^1) - b_2(t,\mathbf{X}_t^1)\|^2]\mathrm{d}t.$$

*Proof.* Upon, using the Nash embedding theorem (see Gunther, 1991), we can assume that $\mathcal{M}$ is a submanifold of $\mathbb{R}^p$ with $p \in \mathbb{N}$ such that the Riemannian metric on $\mathcal{M}$ is induced by the Euclidean metric on $\mathbb{R}^p$. Since $\mathcal{M}$ is compact, there exists $R > 0$ such that $\mathcal{M} \subset \bar{\mathrm{B}}(0,R)$. Let $\varphi \in \mathrm{C}^\infty(\mathbb{R}^p, [0,1])$ such that for any $x \in \bar{\mathrm{B}}(0,R)$, $\varphi(x) = 1$ and for any $x \in \mathbb{R}^p$ with $\|x\| \geq R+1$, $\varphi(x) = 0$. Consider $\bar{b}_1, \bar{b}_2 \in \mathrm{C}_c^2([0,T] \times \mathbb{R}^p, \mathbb{R}^p)$ such that for any $t \in [0,T]$ and $x \in \mathcal{M}$, $\bar{b}_i(x) = b_i(x)$ with $i \in \{1,2\}$. Consider $(\bar{\mathbf{X}}_t^i)_{t\in[0,T]}$ such that for any $i \in \{1,2\}$

$$\mathrm{d}\bar{\mathbf{X}}_t^i = \varphi(\bar{\mathbf{X}}_t^i)\{\mathrm{P}(\bar{\mathbf{X}}_t^i)\bar{b}^i(t,\bar{\mathbf{X}}_t^i) + \bar{b}(\bar{\mathbf{X}}_t^i)\}\mathrm{d}t + \varphi(\bar{\mathbf{X}}_t^i)\mathrm{P}(\bar{\mathbf{X}}_t^i)\mathrm{d}\mathbf{B}_t,$$

where $\bar{b} \in \mathrm{C}^\infty(\mathbb{R}^p, \mathbb{R}^p)$ is defined in the proof of Proposition S12. Let $\bar{\mathbf{X}}_0^i \sim \mathbb{P}_0^1$ for any $i \in \{1,2\}$ then for any $i \in \{1,2\}$, $(\bar{\mathbf{X}}_t^i)_{t\in[0,T]}$ (seen as a process on $\mathcal{M}$) is such that $\mathcal{L}((\bar{\mathbf{X}}_t^i)_{t\in[0,T]}) = \mathbb{P}^i$. Indeed, denote $\{\bar{\mathcal{A}}_t^i\}_{t\in[0,T]}$ the generator of $(\bar{\mathbf{X}}_t^i)_{t\in[0,T]}$ for any $i \in \{1,2\}$. Let $f \in \mathrm{C}^\infty(\mathcal{M}, \mathbb{R})$ and $\bar{f} \in \mathrm{C}^\infty(\mathbb{R}^p, \mathbb{R})$ an extension to $\mathbb{R}^p$. We have that for any $i \in \{1,2\}$, $x \in \mathcal{M}$ and $t \in [0,T]$

$$\begin{aligned}
\bar{\mathcal{A}}_t^i(\bar{f})(x) &= \langle \bar{b}^i(t,x) + \bar{b}(x), \nabla \bar{f}(x)\rangle + \tfrac{1}{2}\langle \mathrm{P}(x), \nabla^2 \bar{f}(x)\rangle \\
&= \langle b^i(t,x), \nabla f(x)\rangle + \tfrac{1}{2}\Delta_{\mathcal{M}}f(x).
\end{aligned}$$

Hence, for any $i \in \{1,2\}$, $(\bar{\mathbf{X}}_t^i)_{t\in[0,T]}$ (seen as a process on $\mathcal{M}$) and $(\mathbf{X}_t^i)_{t\in[0,T]}$ have the same infinitesimal generators. Hence, $\mathcal{L}((\bar{\mathbf{X}}_t^i)_{t\in[0,T]}) = \mathbb{P}^i$ for any $i \in \{1,2\}$. For any $i \in \{1,2\}$, denote $\bar{\mathbb{P}}^i = \mathcal{L}((\bar{\mathbf{X}}_t^i)_{t\in[0,T]})$ (seen as a process on $\mathbb{R}^p$). Note that since for any $x \in \mathbb{R}^p$ with $\|x\| \geq R+1$, $\varphi(x) = 0$ we have that (Liptser and Shiryaev, 2001, Equation (7.137)) is satisfied. In addition, since for any $x \in \mathbb{R}^p$ with $\|x\| \geq R+1$, $\varphi(x) + \|\nabla\varphi(x)\| = 0$, we have that (Liptser and Shiryaev,

2001, Equation (4.110), Equation (4.111)) are satisfied. In addition, letting for any $t \in [0, T]$ and $x \in \mathbb{R}^p$, $\alpha(t, x) = \bar{b}^1(t, x) - \bar{b}^2(t, x) = \mathrm{P}(x)(\bar{b}^1(t, x) - \bar{b}^2(t, x))$, we have that for any $t \in [0, T]$, $\mathrm{P}(x)\alpha(t, x) = \mathrm{P}(x)(\bar{b}^1(t, x) - \bar{b}^2(t, x))$. Therefore, we can apply (Liptser and Shiryaev, 2001, Section 7.6.4) and using that $\mathrm{P}(x)\bar{b}(x) = 0$ for any $x \in \mathcal{M}$ (see the proof of Proposition S12), we have that

$$
\begin{aligned}
(\mathrm{d}\bar{\mathbb{P}}^1/\mathrm{d}\bar{\mathbb{P}}^2)((\bar{\mathbf{X}}_t^1)_{t \in [0,T]}) = \exp \big[ &\textstyle\int_0^T \langle \bar{b}^1(t, \bar{\mathbf{X}}_t^1) - \bar{b}^2(t, \bar{\mathbf{X}}_t^1), \mathrm{P}(\bar{\mathbf{X}}_t^1)\mathrm{d}\bar{\mathbf{X}}_t^1 \rangle \\
&- \tfrac{1}{2} \textstyle\int_0^T \langle \bar{b}^1(t, \bar{\mathbf{X}}_t^1) - \bar{b}^2(t, \bar{\mathbf{X}}_t^1), \mathrm{P}(\bar{\mathbf{X}}_t^1)(\bar{b}^1(t, \bar{\mathbf{X}}_t^1) + \bar{b}^2(t, \bar{\mathbf{X}}_t^1)) \rangle \mathrm{d}t \big] \\
= \exp \big[ &\textstyle\int_0^T \langle \bar{b}^1(t, \bar{\mathbf{X}}_t^1) - \bar{b}^2(t, \bar{\mathbf{X}}_t^1), \mathrm{P}(\bar{\mathbf{X}}_t^1)\{\bar{b}^1(t, \bar{\mathbf{X}}_t^1) + \bar{b}(\bar{\mathbf{X}}_t^1)\} \rangle \mathrm{d}t \\
&+ \textstyle\int_0^T \langle \bar{b}^1(t, \bar{\mathbf{X}}_t^1) - \bar{b}^2(t, \bar{\mathbf{X}}_t^1), \mathrm{P}(\bar{\mathbf{X}}_t^1)\mathrm{d}\mathbf{B}_t \rangle \\
&- \tfrac{1}{2} \textstyle\int_0^T \langle \bar{b}^1(t, \bar{\mathbf{X}}_t^1) - \bar{b}^2(t, \bar{\mathbf{X}}_t^1), \mathrm{P}(\bar{\mathbf{X}}_t^1)(\bar{b}^1(t, \bar{\mathbf{X}}_t^1) + \bar{b}^2(t, \bar{\mathbf{X}}_t^1)) \rangle \mathrm{d}t \big] \\
= \exp[ &\tfrac{1}{2} \textstyle\int_0^T \|\bar{b}^1(t, \bar{\mathbf{X}}_t^1) - \bar{b}^2(t, \bar{\mathbf{X}}_t^1)\|^2 \mathrm{d}t + \textstyle\int_0^T \langle \bar{b}^1(t, \bar{\mathbf{X}}_t^1) - \bar{b}^2(t, \bar{\mathbf{X}}_t^1), \mathrm{P}(\bar{\mathbf{X}}_t^1)\mathrm{d}\mathbf{B}_t \rangle].
\end{aligned}
$$

Therefore, we have that

$$
\mathrm{KL}(\bar{\mathbb{P}}^1 | \bar{\mathbb{P}}^2) = \tfrac{1}{2} \textstyle\int_0^T \mathbb{E}[\|\bar{b}^1(t, \bar{\mathbf{X}}_t^1) - \bar{b}^2(t, \bar{\mathbf{X}}_t^1)\|^2] \mathrm{d}t.
$$

Hence, we get

$$
\mathrm{KL}(\bar{\mathbb{P}}^1 | \bar{\mathbb{P}}^2) = \tfrac{1}{2} \textstyle\int_0^T \mathbb{E}[\|b^1(t, \mathbf{X}_t^1) - b^2(t, \mathbf{X}_t^1)\|^2] \mathrm{d}t.
$$

which concludes the proof. $\qquad\square$

Once Proposition S12 is established, we can obtain the following straightforward extension of Cattiaux et al. (2021, Proposition 4.6).

**Proposition S14.** *Assume* **A**1. *Let* $\mathbb{Q}$ *be a Brownian motion with* $\mathbb{Q}_0 = p_{\mathrm{ref}}$ *and* $\mathbb{P}$ *a path measure on* $\mathrm{C}([0, T], \mathcal{M})$ *such that* $\mathrm{KL}(\mathbb{P}|\mathbb{Q}) < +\infty$. *Then, there exist* $\beta_{\mathbb{P}}, \beta_{R(\mathbb{P})} : [0, T] \times \mathcal{M} \to$ *such that for any* $t \in [0, T]$ *and* $x \in \mathcal{M}$, $\beta_{\mathbb{P}}(t, x), \beta_{R(\mathbb{P})}(t, x) \in \mathrm{T}_x\mathcal{M}$. *In addition, we have that* $\mathbb{P}$ *and* $R(\mathbb{P})$ *satisfy martingale problems with infinitesimal generator* $\mathcal{A}_{\mathbb{P}}$, *respectively* $\mathcal{A}_{R(\mathbb{P})}$ *where for any* $t \in [0, T]$, $u \in \mathrm{C}^2(\mathcal{M})$ *and* $x \in \mathcal{M}$ *we have*

$$
\begin{aligned}
\mathcal{A}_{\mathbb{P}}(t, u)(x) &= \langle \beta_{\mathbb{P}}(t, x), \nabla u(x) \rangle + \tfrac{1}{2}\Delta_{\mathcal{M}} u(x), \\
\mathcal{A}_{R(\mathbb{P})}(t, u)(x) &= \langle \beta_{R(\mathbb{P})}(t, x), \nabla u(x) \rangle + \tfrac{1}{2}\Delta_{\mathcal{M}} u(x).
\end{aligned}
$$

*Finally, we have that*

$$
\textstyle\int_0^T \mathbb{E}[\|\beta_{\mathbb{P}}(t, \mathbf{X}_t)\|^2] \mathrm{d}t + \int_0^T \mathbb{E}[\|\beta_{R(\mathbb{P})}(t, \mathbf{X}_{T-t})\|^2] \mathrm{d}t < +\infty,
$$

*where* $(\mathbf{X}_t)_{t \in [0, T]}$ *has distribution* $\mathbb{P}$.

*Proof.* The proof is straightforward upon combining Proposition S12 and the fact that $\mathrm{KL}(\mathbb{P}|\mathbb{Q}) = \mathrm{KL}(R(\mathbb{P})|R(\mathbb{Q})) = \mathrm{KL}(R(\mathbb{P})|\mathbb{Q}) < +\infty$, using that $\mathbb{Q}$ is stationary. $\qquad\square$

We conclude this section, with the following application of Theorem S11.

**Proposition S15.** *For any* $u, v \in \mathrm{C}_c^\infty(\mathcal{M})$, *we have that for almost any* $t \in [0, T]$

$$
\mathbb{E}[v(\mathbf{X}_t)(\langle \beta_{\mathbb{P}}(t, \mathbf{X}_t) + \beta_{R(\mathbb{P})}(T-t, \mathbf{X}_t), \nabla u(\mathbf{X}_t) \rangle + \Delta_{\mathcal{M}} u(\mathbf{X}_t)) + \langle \nabla u(\mathbf{X}_t), \nabla v(\mathbf{X}_t) \rangle] = 0. \quad \text{(S17)}
$$

*Proof.* Remark that $\mathrm{C}_c^2(\mathcal{M}) \subset \mathrm{dom}(\Upsilon_{\mathbb{P}})$ and $\mathrm{C}_c^2(\mathcal{M}) \subset \mathrm{dom}(\Upsilon_{R(\mathbb{P})})$. In addition, we have that for any $u, v \in \mathrm{C}_c^2(\mathcal{M})$, $\Upsilon_{\mathbb{P}}(u, v) = \Upsilon_{R(\mathbb{P})}(u, v) = \langle u, v \rangle$. Note that by Proposition S14 and Theorem S11 we have that for any $u, v \in \mathrm{C}_c^\infty(\mathcal{M})$, (S17) holds. $\qquad\square$

### H.2.3 Concluding the proof

Using Proposition S15 we can now conclude the proof of Theorem 1. First, remark that we can identify $\beta_{\mathbb{P}} = b$. Let $u, v \in C^\infty(\mathcal{M})$, we have that

$$\mathbb{E}[v(\mathbf{X}_t)\langle b(\mathbf{X}_t) + \beta_{R(\mathbb{P})}(T - t, \mathbf{X}_t), \nabla u(\mathbf{X}_t)\rangle + \Delta_{\mathcal{M}}u(\mathbf{X}_t)v(\mathbf{X}_t) + \langle \nabla u(\mathbf{X}_t), \nabla v(\mathbf{X}_t)\rangle] = 0.$$

Using that for any $t \in [0, T]$, $\mathbb{P}_t$ admits a smooth positive density w.r.t. $p_{\text{ref}}$ denoted $p_t$ and the divergence theorem, see (Lee, 2018, p.51), we have that for any $t \in [0, T]$,

$$\int_{\mathcal{M}}\{\langle\beta_{R(\mathbb{P})}(T - t, x), \nabla u(x)\rangle + \langle b(x), \nabla u(x)\rangle\}v(x)p_t(x)\mathrm{d}p_{\text{ref}}(x)$$
$$= -\int_{\mathcal{M}}\langle\nabla u(x)p_t(x), \nabla v(x)\rangle\mathrm{d}p_{\text{ref}}(x) - \int_{\mathcal{M}}\Delta_{\mathcal{M}}u(x)v(x)p_t(x)\mathrm{d}p_{\text{ref}}(x)$$
$$= \int_{\mathcal{M}}\langle\nabla\log p_t(x), \nabla u(x)v(x)p_t(x)\mathrm{d}p_{\text{ref}}(x).$$

Therefore, we get that for any $t \in [0, T]$ and $x \in \mathcal{M}$, $\langle\beta_{R(\mathbb{P})}(T - t, x), \nabla u(x)\rangle = \langle -b(x) + \nabla\log p_t(x), \nabla u(x)\rangle$, which concludes the proof.

## I Convergence of RSGM

In this section, we study the convergence of RSGM and prove Theorem 4. We state our main results in App. I.1 and give discretization bounds following the recent work of Cheng et al. (2022) in sec:discr-bounds-grw.

### I.1 Main results

In this section, we prove Theorem 4. We start by recalling the sequence considered in RSGM. Let $(Y_k)_{k\in\{0,\ldots,N\}}$ be given by $Y_0 \sim p_{\text{ref}}$ and for any $k \in \{0, \ldots, N - 1\}$

$$Y_{k+1} = \exp_{Y_k}[\gamma\mathbf{s}_{\theta^\star}(T - n\gamma, Y_k) + \sqrt{2}Z_{k+1}],$$

where $\{Z_k\}_{n\in\mathbb{N}}$ is a sequence of independent square integrable random variables with zero mean and identity covariance matrix. For ease of reading, we restate Theorem 4.

**Theorem S16.** *Assume* **A**1, *that $p_0$ is smooth and positive and that there exists* $\mathtt{M} \geq 0$ *such that for any $t \in [0, T]$ and $x \in \mathcal{M}$, $\|\mathbf{s}_{\theta^\star}(t, x) - \nabla\log p_t(x)\| \leq \mathtt{M}$, with $\mathbf{s}_{\theta^\star} \in C([0, T], \mathcal{X}(\mathcal{M}))$. Then if $T > 1/2$, there exists $C \geq 0$ independent on $T$ such that*

$$\mathbf{W}_1(\mathcal{L}(Y_N), p_0) = C(\mathrm{e}^{-\lambda_1 T} + \sqrt{T/2}\mathtt{M} + \mathrm{e}^T\gamma^{1/2}),$$

*where $\mathbf{W}_1$ is the Wasserstein distance of order one on the probability measures on $\mathcal{M}$.*

*Proof.* For any $k \in \{1, \ldots, N\}$, denote $\mathtt{R}_k$ such that for any $x \in \mathbb{R}^d$, $\mathsf{A} \in \mathcal{B}(\mathbb{R}^d)$ and $k \in \{0, \ldots, N - 1\}$ we have

$$\mathbb{E}[\mathtt{R}_{k+1}(Y_k, \mathsf{A})] = \mathbb{E}[\mathbb{1}_{\mathsf{A}}(Y_{k+1})].$$

Define for any $k_0, k_1 \in \{1, \ldots, N\}$ with $k_1 \geq k_0$ $\mathtt{Q}_{k_0,k_1} = \prod_{\ell=k_0}^{k_1}\mathtt{R}_{k_1+k_0-\ell}$. Finally, for ease of notation, we also define for any $k \in \{1, \ldots, N\}$, $\mathtt{Q}_k = \mathtt{Q}_{k+1,N}$. Note that for any $k \in \{1, \ldots, N\}$, $Y_k$ has distribution $\pi_\infty\mathtt{Q}_k$, where $\pi_\infty \in \mathcal{P}(\mathcal{M})$ with density w.r.t. the Hausdorff measure $p_{\text{ref}}$. Let $\mathbb{P} \in \mathcal{P}(\mathcal{C})$ be the probability measure associated with $(\mathbf{B}_t)_{t\in[0,T]}$ with $\mathbf{B}_0 \sim \pi_0$, where $\pi_0 \in \mathcal{P}(\mathcal{M})$ admits a density w.r.t. the Hausdorff measure given by $p_0$. We denote $(\hat{\mathbf{Y}}_t)_{t\in[0,T]}$ the process defined by the diffusion $\mathrm{d}\hat{\mathbf{Y}}_t = \mathbf{s}_{\theta^\star}(T - t, \hat{\mathbf{Y}}_t)\mathrm{d}t + \mathrm{d}\mathbf{B}_t$ and $\hat{\mathbf{Y}}_0 \sim \pi_\infty$. We also denote $\hat{\mathbb{P}}^R \in \mathcal{P}(\mathcal{C})$ the probability measure associated with $(\hat{\mathbf{Y}}_t)_{t\in[0,T]}$. First note that using that $\mathbb{P}_0 = \pi_0$ we have for any $\mathsf{A} \in \mathcal{B}(\mathcal{M})$

$$\pi_0\mathbb{P}_{T|0}(\mathbb{P}^R)_{T|0}(\mathsf{A}) = \mathbb{P}_T(\mathbb{P}^R)_{T|0}(\mathsf{A}) = (\mathbb{P}^R)_0(\mathbb{P}^R)_{T|0}(\mathsf{A}) = (\mathbb{P}^R)_T(\mathsf{A}) = \pi_0(\mathsf{A}).$$

Hence we have that

$$\pi_0 = \pi_0\mathbb{P}_{T|0}(\mathbb{P}^R)_{T|0}. \tag{S18}$$

Let $\varphi \in C(\mathcal{M})$ with is 1-Lipschitz, i.e. for any $x, y \in \mathcal{M}$, $|\varphi(x) - \varphi(y)| \leq d(x, y)$. Since $\mathcal{M}$ is compact, we have that $\varphi$ is bounded. Using this result, (S18), the data processing theorem (Kullback, 1997, Theorem 4.1) and Pinsker's inequality (Bakry et al., 2014, Equation 5.2.2) we have

$$|\mathbb{E}[\varphi(Y_N)] - \int_{\mathcal{M}} \varphi(x) p_0(x) \mathrm{d}\mu(x)|$$

$$\leq |\mathbb{E}[\varphi(\mathbf{B}_0)] - \mathbb{E}[\varphi(\mathbf{Y}_T)]| + |\mathbb{E}[\varphi(\hat{\mathbf{Y}}_T)] - \mathbb{E}[\varphi(\mathbf{Y}_T)]| |\mathbb{E}[\varphi(\hat{\mathbf{Y}}_T)] - \mathbb{E}[\varphi(Y_N)]|$$

$$\leq \|\varphi\|_\infty \|\pi_0 - \pi_\infty (\mathbb{P}^R)_{T|0}\|_{\mathrm{TV}} + |\mathbb{E}[\varphi(\hat{\mathbf{Y}}_T)] - \mathbb{E}[\varphi(\mathbf{Y}_T)]| + |\mathbb{E}[\varphi(\hat{\mathbf{Y}}_T)] - \mathbb{E}[\varphi(Y_N)]|$$

$$\leq \|\varphi\|_\infty \|\pi_0 \mathbb{P}_{T|0} (\mathbb{P}^R)_{T|0} - \pi_\infty (\mathbb{P}^R)_{T|0}\|_{\mathrm{TV}} + |\mathbb{E}[\varphi(\hat{\mathbf{Y}}_T)] - \mathbb{E}[\varphi(\mathbf{Y}_T)]| + |\mathbb{E}[\varphi(\hat{\mathbf{Y}}_T)] - \mathbb{E}[\varphi(Y_N)]|$$

$$\leq \|\varphi\|_\infty \|\pi_0 \mathbb{P}_{T|0} - \pi_\infty\|_{\mathrm{TV}} + |\mathbb{E}[\varphi(\hat{\mathbf{Y}}_T)] - \mathbb{E}[\varphi(\mathbf{Y}_T)]| + |\mathbb{E}[\varphi(\hat{\mathbf{Y}}_T)] - \mathbb{E}[\varphi(Y_N)]|$$

$$\leq \|\varphi\|_\infty \|\pi_0 \mathbb{P}_{T|0} - \pi_\infty\|_{\mathrm{TV}} + \sqrt{2}\|\varphi\|_\infty \mathrm{KL}^{1/2}(\pi_\infty \mathbb{P}^R_{|0} | \pi_\infty \hat{\mathbb{P}}^R_{|0}) + |\mathbb{E}[\varphi(\hat{\mathbf{Y}}_T)] - \mathbb{E}[\varphi(Y_N)]|.$$

We now control each one of these terms. The first term can be easily controlled using the geometric ergodicity of the Brownian motion on compact manifolds. The second term can be controlled using the Girsanov theory on isometrically embedded manifolds. For the last term, we rely on the convergence of the GRW to its associated diffusion as presented in App. I.2. We now control each one of these terms.

(a) Using Proposition S10, we have that $\|\pi_0 \mathbb{P}_{T|0} - \pi_\infty\|_{\mathrm{TV}} \leq C^{1/2} e^{\lambda_1/2} e^{-\lambda_1 T}$ where $\lambda_1$ is the first positive eigenvalue of $-\Delta_{\mathcal{M}}$ in $L^2(\pi_\infty)$. Therefore, we get that

$$\|\varphi\|_\infty \|\pi_0 \mathbb{P}_{T|0} - \pi_\infty\|_{\mathrm{TV}} \leq C^{1/2} e^{\lambda_1/2} \|\varphi\|_\infty e^{-\lambda_1 T}.$$

(b) Recall that we have that $\mathbb{P}^R_{|0}$ is associated with the process $\mathrm{d}\mathbf{Y}_t = \nabla \log p_{T-t}(\mathbf{Y}_t)\mathrm{d}t + \mathrm{d}\mathbf{B}^{\mathcal{M}}_t$ and that $\hat{\mathbb{P}}^R_{|0}$ is associated with the process $\mathrm{d}\hat{\mathbf{Y}}_t = \mathbf{s}_{\theta^\star}(T - t, \hat{\mathbf{Y}}_t)\mathrm{d}t + \mathrm{d}\mathbf{B}^{\mathcal{M}}_t$. Using Corollary S13 we have that

$$\mathrm{KL}(\pi_\infty \mathbb{P}^R_{|0} | \pi_\infty \hat{\mathbb{P}}^R_{|0}) = \tfrac{1}{2} \int_0^T \mathbb{E}[\|\mathbf{s}_{\theta^\star}(T - t, \mathbf{Y}_t) - \nabla \log p_{T-t}(\mathbf{Y}_t)\|^2] \leq M^2 T.$$

(c) Let us define $\{\bar{\mathbf{Y}}^k\}_{k=0}^N$ such that for any $k \in \{0, \ldots, N\}$, $\bar{\mathbf{Y}}_0^k = \hat{\mathbf{Y}}_0 = Y_0$ and for any $t \in [0, k\gamma]$ we have that $\bar{\mathbf{Y}}_t^0 = \hat{\mathbf{Y}}_t$. For any $t \in [k\gamma, T]$, we have that $\bar{\mathbf{Y}}_t^k = Y_{t,k}$, where $Y_{k\gamma,k} = \hat{\mathbf{Y}}_{k\gamma}$ and for any $j \in \{k, \ldots, N-1\}$ and $t \in [0, \gamma]$

$$Y_{j\gamma+t,k} = \exp_{Y_{j\gamma,k}}[t\mathbf{s}_{\theta^\star}(T - j\gamma, Y_{j\gamma,k}) + \sqrt{t}E_j^k Z_j],$$

where $\{Z_j\}_{j=0}^{N-1}$ are independent Gaussian random variables with identity covariance matrix and zero mean and $E_j^k$ is a frame of $\mathrm{T}_{Y_{j\gamma,k}}\mathcal{M}$ such that for any $j \in \{k+1, \ldots, N-1\}$, $E_j^{k+1} = \Gamma_{Y_{j\gamma,k}}^{Y_{j\gamma,k+1}} E_j^k$ and $\{E_j^0\}_{j=0}^{N-1}$ is such that for any $j \in \{0, \ldots, N-1\}$, $E_j^0$ is a frame of $\mathrm{T}_{Y_{j\gamma}}\mathcal{M}$. One $[0, k\gamma]$, we define $(\hat{\mathbf{Y}}_t^k)_{t \in [0,k\gamma]}$ as follows. For any $k \in \{0, \ldots, N-1\}$, we set $(\mathbf{Y}_t^{k+1})_{t \in [0,k\gamma]} = (\mathbf{Y}_t^k)_{t \in [0,k\gamma]}$. For any $k \in \{0, \ldots, N-1\}$, we set $(\mathbf{Y}_t)_{k\gamma,(k+1)\gamma}$ as in Proposition S21 (taking the notations of Proposition S21, $X_1^0 = \hat{\mathbf{Y}}_{(k+1)\gamma}^k$ and $X_\gamma = \hat{\mathbf{Y}}_{k\gamma}^k$). Note that we have $\{\bar{\mathbf{Y}}_{j\gamma,0}^N\}_{j=0}^N = \{Y_j^N\}_{j=0}^N$ and $\{\bar{\mathbf{Y}}_{t,N}\}_{t \in [0,T]} = \{\hat{\mathbf{Y}}_t\}_{t \in [0,T]}$. Therefore, we have that

$$|\varphi(\hat{\mathbf{Y}}_T) - \varphi(Y_N)| = |\varphi(\bar{\mathbf{Y}}_T^0) - \varphi(\bar{\mathbf{Y}}_T^N)|$$

$$\leq \textstyle\sum_{k=0}^{N-1} |\varphi(\bar{\mathbf{Y}}_T^k) - \varphi(\bar{\mathbf{Y}}_T^{k+1})| \leq \|\nabla\varphi\|_\infty \textstyle\sum_{k=0}^{N-1} d(\bar{\mathbf{Y}}_T^k, \bar{\mathbf{Y}}_T^{k+1}).$$

In addition, using Proposition S21 and Proposition S22, we have that there exists $C \geq 0$ such that for any $k \in \{0, \ldots, N-1\}$

$$\mathbb{E}[d(\bar{\mathbf{Y}}_{k,T}, \bar{\mathbf{Y}}_{k+1,T})] \leq C \exp[(N-k)\gamma]\gamma^{3/2}.$$

Therefore, we get that there exists $C \geq 0$ such that

$$|\mathbb{E}[\varphi(\hat{\mathbf{Y}}_T)] - \mathbb{E}[\varphi(Y_N)]| \leq C \exp[T]\gamma^{1/2},$$

Therefore, we get that there exists $C \geq 0$ such that for any $\varphi \in C(\mathcal{M})$ which is 1-Lipschitz, we have

$$\mathbb{E}[\varphi(Y_N)] - \int_{\mathcal{M}} \varphi(x) p_0(x) \mathrm{d}p_{\mathrm{ref}}(x) \leq C(\mathrm{e}^{\lambda_1/2} \|\varphi\|_\infty \mathrm{e}^{-\lambda_1 T} + \sqrt{T/2} \|\varphi\|_\infty \mathtt{M} + \mathrm{e}^T \gamma^{1/2}). \quad \text{(S19)}$$

Let $x_0 \in \mathcal{M}$. Let $\mathrm{Lip}(\mathcal{M})$ the set of Lipschitz functions on $\mathcal{M}$ with Lipschitz constant equal to 1. Let $\mathrm{Lip}(\mathcal{M})_0$ the set of Lipschitz functions on $\mathcal{M}$ with Lipschitz constant equal to 1 and such that for any $\varphi \in \mathrm{Lip}(\mathcal{M})_0$, $\varphi(x_0) = 0$. Note that in this case, we have that $\|\varphi\|_\infty \leq \mathrm{diam}(\mathcal{M})$. Using (S19), we have

$$\begin{aligned}
\mathbf{W}_1(\mathcal{L}(Y_N), p_0) &= \sup\{\mathbb{E}[\varphi(Y_N)] - \int_{\mathcal{M}} \varphi(x) p_0(x) \mathrm{d}p_{\mathrm{ref}}(x) \, : \, \varphi \in \mathrm{Lip}(\mathcal{M})\} \\
&= \sup\{\mathbb{E}[\varphi(Y_N)] - \int_{\mathcal{M}} \varphi(x) p_0(x) \mathrm{d}p_{\mathrm{ref}}(x) \, : \, \varphi \in \mathrm{Lip}(\mathcal{M})_0\} \\
&\leq C(\mathrm{e}^{\lambda_1/2} \mathrm{diam}(\mathcal{M}) \mathrm{e}^{-\lambda_1 T} + \sqrt{T/2} \mathrm{diam}(\mathcal{M}) \mathtt{M} + \mathrm{e}^T \gamma^{1/2}),
\end{aligned}$$

which concludes the proof. $\qquad \square$

We now state a result regarding the continuous-time process (i.e. we now longer consider discretization errors). We recall that we denote $(\hat{\mathbf{Y}}_t)_{t \in [0,T]}$ the process defined by the diffusion $\mathrm{d}\hat{\mathbf{Y}}_t = \mathbf{s}_{\theta^\star}(T - t, \hat{\mathbf{Y}}_t)\mathrm{d}t + \mathrm{d}\mathbf{B}_t$ and $\hat{\mathbf{Y}}_0 \sim \pi_\infty$.

**Theorem S17.** *Assume* **A**1, *that $p_0$ is smooth and positive and that there exists $\mathtt{M} \geq 0$ such that for any $t \in [0,T]$ and $x \in \mathcal{M}$, $\|\mathbf{s}_{\theta^\star}(t,x) - \nabla \log p_t(x)\| \leq \mathtt{M}$, with $\mathbf{s}_{\theta^\star} \in C([0,T], \mathcal{X}(\mathcal{M}))$. Then if $T > 1/2$, there exists $C \geq 0$ independent on $T$ such that*

$$\|\mathcal{L}(\hat{\mathbf{Y}}_T) - p_0\|_{\mathrm{TV}} = C(\mathrm{e}^{-\lambda_1 T} + \sqrt{T/2}\mathtt{M}).$$

*Proof.* The proof is identical to the one of Theorem S16, except that we do not have to deal with the discretization error. We use that for any $\mu, \nu \in \mathcal{P}(\mathcal{M})$

$$\|\mu - \nu\|_{\mathrm{TV}} = \sup\{\mu[f] - \nu[f] \, : \, f \in C(\mathcal{M}), \|f\|_\infty \leq 1\}.$$

$\qquad \square$

The result of Theorem S17 should be compared with the one of (Rozen et al., 2021, Theorem 3). With our result we control a $\mathrm{L}^1$ bound between the density of $\hat{\mathbf{Y}}_T$ and the one of $p_0$. In (Rozen et al., 2021, Theorem 3) a $\mathrm{L}^\infty$ bound between the densities is recovered. It can be shown that $\hat{p}_T = \mathcal{L}(\hat{\mathbf{Y}}_T)$. Let $\kappa$ be the modulus of continuity of $\hat{p}_T - p_0$, i.e. for any $\varepsilon \geq 0$

$$\kappa(\varepsilon) = \sup\{|\hat{p}_T(x) - p_0(x) - \hat{p}_T(y) + p_0(y)| \, : \, x, y \in \mathcal{M}, \, d(x,y) \leq \varepsilon\}.$$

Let $x_0 \in \mathcal{M}$ such that

$$|\hat{p}_T(x_0) - p_0(x_0)| = M = \sup\{|\hat{p}_T(x) - p_0(x)| \, : \, x \in \mathcal{M}\}.$$

For any $x \in \bar{\mathrm{B}}(x_0, \kappa(M/2))$, we have $|\hat{p}_T(x) - p_0(x)| \geq M/2$. Hence, denoting $\mathrm{Vol}_\kappa = \int_{\bar{\mathrm{B}}(x_0, \kappa(M/2))} \mathrm{d}p_{\mathrm{ref}}(x) > 0$, we have

$$(2/\mathrm{Vol}_\kappa) \int_{\mathcal{M}} |\hat{p}_T(x) - p_0(x)| \mathrm{d}p_{\mathrm{ref}}(x) \geq \|\hat{p}_T - p_0\|_\infty \, .$$

Hence, there exists $C \geq 0$ such that for any $T > 1/2$

$$\|\hat{p}_T - p_0\|_\infty \leq C(\mathrm{e}^{-\lambda_1 T} + \sqrt{T/2}\mathtt{M}).$$

Therefore, we recover the same guarantees as Theorem S17 (note that $\mathtt{M}$ is not explicitly controlled using network properties in our work, but we could use universal approximation properties as in Rozen et al. (2021) in order to obtain a similar result).

## I.2 Discretization bounds for GRW

In this section, we establish discretization bounds for GRW. Our results are a straightforward extension of Cheng et al. (2022) to the case where the drift term in the GRW is time-inhomogeneous.

Since $\mathcal{M}$ is compact, we have that for any $x_1, x_2 \in M$, there exists a minimizing geodesic such that $\gamma \in C^\infty([0,1], \mathcal{M})$ and $\gamma(0) = x_1$ and $\gamma(1) = x_2$. When this choice is not unique we fix a

minimizing geodesic. We denote $\Gamma_{x_1}^{x_2} : \mathrm{T}_{x_1}\mathcal{M} \to \mathrm{T}_{x_2}\mathcal{M}$ the associated parallel transport. Let $b \in \mathrm{C}^\infty([0,T], \mathcal{X}(\mathcal{M}))$.

We start by introducing a family of GRWs defined on progressively finer grids. Let $\gamma > 0$, $X_0 \in \mathcal{M}$, $E_0 \in \mathrm{F}_{X_0}\mathcal{M}$ (the vector space of frames at $X_0$) and consider the families $\{E_k^\ell : k \in \{0, \ldots, 2^\ell\}, \ell \in \mathbb{N}\}$, $\{X_k^\ell : k \in \{0, \ldots, 2^\ell\}, \ell \in \mathbb{N}\}$ such that $X_0^0 = X_0$, $X_1^0 = \exp_{X_0^0}[\gamma b(0, X_0^0) + \sqrt{\gamma}(\mathbf{B}_1 - \mathbf{B}_0)E_0^0]$ and $E_1^0 = \Gamma_{X_0^0}^{X_1^0} E_0^0$ (note that $E_{2^\ell}^\ell$ is not used in the proof but defined for completeness). In addition, we have that for any $\ell \in \mathbb{N}$ with $\ell \geq 1$, $X_0^\ell = X_0$, $E_0^\ell = E_0$ and for any $k \in \{0, \ldots, 2^{\ell-1} - 1\}$

$$X_{2k+1}^\ell = \exp_{X_{2k}^\ell}[\gamma_\ell b(2k\gamma_\ell, X_{2k}^\ell) + E_{2k}^\ell(\mathbf{B}_{(2k+1)\gamma_\ell} - \mathbf{B}_{2k\gamma_\ell})],$$

$$E_{2k+1}^\ell = \Gamma_{X_{2k}^\ell}^{X_{2k+1}^\ell} E_{2k}^\ell,$$

$$X_{2k+2}^\ell = \exp_{X_{2k+1}^\ell}[\gamma_\ell b((2k+1)\gamma_\ell, X_{2k+1}^\ell) + E_{2k+1}^\ell(\mathbf{B}_{(2k+2)\gamma_\ell} - \mathbf{B}_{(2k+1)\gamma_\ell})],$$

$$E_{2k+2}^\ell = \Gamma_{X_{k+1}^{\ell-1}}^{X_{2k+2}^\ell} E_{k+1}^{\ell-1}, \tag{S20}$$

where $\gamma_\ell = \gamma/2^\ell$. For any $\ell \in \mathbb{N}$, we also define $(\mathbf{X}_t^\ell)_{t \in [0, \gamma]}$ such that for any $\ell \in \mathbb{N}$, $k \in \{0, \ldots, 2^\ell - 1\}$, we have for any $t \in [k\gamma_\ell, (k+1)\gamma_\ell)$, $\mathbf{X}_t^\ell = \exp_{X_k^\ell}[(t - k\gamma_\ell)b(k\gamma_\ell, X_k^\ell) + E_k^\ell(\mathbf{B}_t - \mathbf{B}_{k\gamma_\ell})]$. Note that for any $\ell \in \mathbb{N}$ and $k \in \{0, \ldots, 2^\ell - 1\}$, $\mathbf{X}_{k\gamma_\ell}^\ell = X_k$.

We are going to use the following useful lemma, see (Cheng et al., 2022, Lemma 62).

**Lemma S18.** *Assume* **A**1. *Then, there exists $C \geq 0$ such that for any $x, y \in \mathcal{M}$, $\gamma : [0,1] \to \mathcal{M}$ minimizing geodesic with $\gamma(0) = x$, $\gamma(1) = y$ and $u \in \mathrm{T}_x\mathcal{M}$, $v \in \mathrm{T}_y\mathcal{M}$ we have*

$$d(\exp_y[v], \exp_x[u])^2 \leq (1 + C\kappa^2 \exp[4\kappa])d(x,y)^2 + C\exp[4\kappa]\|\Gamma_y^x v - u\|^2 + 2\langle \gamma'(0), \Gamma_y^x v - u \rangle,$$

*with $\kappa = \|u\| + \|v\|$.*

We are now ready to state the main result of this section.

**Proposition S19.** *Assume* **A**1. *Then, there exists $C \geq 0$ such that for any $\ell \in \mathbb{N}$*

$$\mathbb{E}[\sup_{t \in [0,\gamma]} d(\mathbf{X}_t^\ell, \mathbf{X}_t^{\ell+1})^2] \leq C\gamma^3 2^{-2\ell}.$$

*Proof.* Let $\ell \in \mathbb{N}$, $k \in \{0, \ldots, 2^\ell - 1\}$ and $t \in [k\gamma_\ell, (k+1)\gamma_\ell]$. We define $U_k^t = d(\mathbf{X}_t^\ell, \mathbf{X}_t^{\ell+1})^2$, $U_k = \sup\{U_k^t : t \in [k\gamma_\ell, (k+1)\gamma_\ell]\}$ and $U_{-1} = 0$. We also introduce for any $j \in \{0, \ldots, 2^\ell - 1\}$ and for $t \in [k\gamma_\ell, (2k+1)\gamma_{\ell+1})$, $\bar{\mathbf{X}}_t^{\ell+1} = \mathbf{X}_t^{\ell+1}$ and for $t \in [(2k+1)\gamma_{\ell+1}, (k+1)\gamma_\ell)$

$$\bar{\mathbf{X}}_t^{\ell+1} = \exp_{X_{2j}^{\ell+1}}[\gamma_{\ell+1}b(2j\gamma_{\ell+1}, X_{2j}^{\ell+1}) + (t-(2k+1)\gamma_{\ell+1})b((2j+1)\gamma_{\ell+1}, X_{2j}^{\ell+1}) + (\mathbf{B}_t - \mathbf{B}_{j\gamma_\ell})E_{2j}^{\ell+1}].$$

Using this result and that for any $a, b \geq 0$, $(a+b)^2 \leq (1 + 2^{-\ell})a^2 + (1 + 2^\ell)b^2$, we have that for any $t \in [k\gamma_\ell, (k+1)\gamma_\ell]$

$$U_{k+1}^t \leq (1 + 2^{-\ell})d(\mathbf{X}_t^\ell, \bar{\mathbf{X}}_t^{\ell+1})^2 + (1 + 2^\ell)d(\bar{\mathbf{X}}_t^{\ell+1}, \mathbf{X}_t^{\ell+1})^2. \tag{S21}$$

Note that for $t \in [k\gamma_\ell, (2k+1)\gamma_{\ell+1}]$, the second term in (S21) is zero. We now bound each one of these terms:

(a) First, we assume that $t \in [(k+1)\gamma_\ell, (2k+1)\gamma_{\ell+1}]$. Recall that

$$\bar{\mathbf{X}}_t^{\ell+1} = \exp_{X_{2k}^{\ell+1}}[\gamma_{\ell+1}b(k\gamma_\ell, X_{2k}^{\ell+1}) + (t - (2k+1)\gamma_{\ell+1})b((2k+1)\gamma_{\ell+1}, X_{2k}^{\ell+1}) + (\mathbf{B}_t - \mathbf{B}_{k\gamma_\ell})E_{2k}^{\ell+1}],$$

$$\mathbf{X}_t^\ell = \exp_{X_k^\ell}[(t - k\gamma_\ell)b(k\gamma_\ell, X_k^\ell) + (\mathbf{B}_t - \mathbf{B}_{k\gamma_\ell})E_k^\ell].$$

Hence, using Lemma S18, we have that

$$d(\bar{\mathbf{X}}_t^{\ell+1}, \mathbf{X}_t^\ell)^2 \leq (1 + C\kappa_k^2 \exp[4\kappa_k])d(X_k^\ell, X_{2k}^{\ell+1})^2 \tag{S22}$$

$$+ C\exp[4\kappa_k]\|\Gamma_{X_{2k}^{\ell+1}}^{X_k^\ell} v_k - u_k\|^2 + 2\langle w'(0), \Gamma_{X_{2k}^{\ell+1}}^{X_k^\ell} v_k - u_k \rangle,$$

with $w : [0,1] \to \mathcal{M}$ a minimizing geodesic between $X_k^\ell$ and $X_{2k}^{\ell+1}$

$$
\begin{aligned}
\kappa_k &= \|u_k\| + \|v_k\|, \\
u_k^1 &= (t - k\gamma_\ell)b(k\gamma_\ell, X_k^\ell), \\
v_k^1 &= \gamma_{\ell+1}b(2k\gamma_{\ell+1}, X_{2k}^{\ell+1}) + (t - (2k+1)\gamma_{\ell+1})b((2k+1)\gamma_{\ell+1}, X_{2k}^{\ell+1}), \\
u_k^2 &= (\mathbf{B}_t - \mathbf{B}_{k\gamma_\ell})E_k^\ell, \qquad\qquad v_k^2 = (\mathbf{B}_t - \mathbf{B}_{k\gamma_\ell})E_{2k}^{\ell+1}, \\
u_k &= u_k^1 + u_k^2, \qquad\qquad\qquad v_k = v_k^1 + v_k^2.
\end{aligned}
$$

In particular, since $E_k^\ell = \Gamma_{X_{2k}^{\ell+1}}^{X_k^\ell} E_{2k}^{\ell+1}$ using (S20), we have that $u_k^2 = \Gamma_{X_{2k}^{\ell+1}}^{X_k^\ell} v_k^2$. Therefore, combining this result and that $t - (2k+1)\gamma_{\ell+1} + \gamma_{\ell+1} = t - k\gamma_\ell$, we get that

$$
\begin{aligned}
\|\Gamma_{X_{2k}^{\ell+1}}^{X_k^\ell} v_k^1 - u_k^1\| &\leq \gamma_{\ell+1}\|b(k\gamma_\ell, X_k^\ell) - \Gamma_{X_{2k}^{\ell+1}}^{X_k^\ell} b(k\gamma_\ell, X_{2k}^{\ell+1})\| \\
&\quad + \gamma_{\ell+1}\|b(k\gamma_\ell, X_k^\ell) - \Gamma_{X_{2k}^{\ell+1}}^{X_k^\ell} b((2k+1)\gamma_{\ell+1}, X_{2k}^{\ell+1})\| \\
&\leq \gamma_\ell\|b(k\gamma_\ell, X_k^\ell) - \Gamma_{X_{2k}^{\ell+1}}^{X_k^\ell} b(k\gamma_\ell, X_{2k}^{\ell+1})\| + \mathsf{L}_2\gamma_\ell^2 \\
&\leq \mathsf{L}_1\gamma_\ell d(X_k^\ell, X_{2k}^{\ell+1}) + \mathsf{L}_2\gamma_\ell^2.
\end{aligned}
$$

Therefore, we get that $\|u_k - v_k\| \leq \mathsf{L}_1\gamma_\ell d(X_k^\ell, X_{2k}^{\ell+1}) + \mathsf{L}_2\gamma_\ell^2$. In addition, we have that $\|w'(0)\| \leq d(X_k^\ell, X_{2k}^{\ell+1})$ since $w$ is a minimizing geodesic. Combining these results and (S22) we get that

$$
\begin{aligned}
d(\bar{\mathbf{X}}_t^{\ell+1}, \mathbf{X}_t^\ell)^2 &\leq (1 + C\kappa_k^2 \exp[4\kappa_k])d(X_k^\ell, X_{2k}^{\ell+1})^2 \\
&\quad + C\exp[4\kappa_k](\mathsf{L}_1\gamma_\ell d(X_k^\ell, X_{2k}^{\ell+1}) + \mathsf{L}_2\gamma_\ell^2)^2 \\
&\quad + 2(\mathsf{L}_1\gamma_\ell d(X_k^\ell, X_{2k}^{\ell+1}) + \mathsf{L}_2\gamma_\ell^2)d(X_k^\ell, X_{2k}^{\ell+1}) \\
&\leq (1 + C\kappa_k^2 \exp[4\kappa_k] + 2C\exp[4\kappa_k]\mathsf{L}_1^2\gamma_\ell^2)d(X_k^\ell, X_{2k}^{\ell+1})^2 \\
&\quad + 2(\mathsf{L}_1\gamma_\ell d(X_k^\ell, X_{2k}^{\ell+1}) + \mathsf{L}_2\gamma_\ell^2)d(X_k^\ell, X_{2k}^{\ell+1}) + 2\mathsf{L}_2^2\gamma_\ell^4 \\
&\leq (1 + C\kappa_k^2 \exp[4\kappa_k] + 2C\exp[4\kappa_k]\mathsf{L}_1^2\gamma_\ell^2 + 2\mathsf{L}_1\gamma_\ell + 4\mathsf{L}_2\gamma_\ell)d(X_k^\ell, X_{2k}^{\ell+1})^2 + 8\mathsf{L}_2\gamma_\ell^3,
\end{aligned}
$$

Hence, there exists $C_1 \geq 0$ (not dependent on $k$ or $\ell$) such that

$$
(1 + 2^{-\ell})d(\bar{\mathbf{X}}_t^{\ell+1}, \mathbf{X}_t^\ell)^2 \leq (1 + C_1\{\kappa_k^2 \exp[4\kappa_k] + \gamma_\ell^2 \exp[4\kappa_k] + 2^{-\ell}\})d(X_k^\ell, X_{2k}^{\ell+1})^2 + C_1\gamma_\ell^3.
$$

Next, we assume that $t \in [k\gamma_\ell, (2k+1)\gamma_{\ell+1}]$. Recall that

$$
\begin{aligned}
\bar{\mathbf{X}}_t^{\ell+1} &= \exp_{X_{2k}^{\ell+1}}[(t - k\gamma_\ell)b(k\gamma_\ell, X_{2k}^{\ell+1}) + (\mathbf{B}_t - \mathbf{B}_{k\gamma_\ell})E_{2k}^{\ell+1}], \\
\mathbf{X}_t^\ell &= \exp_{X_k^\ell}[(t - k\gamma_\ell)b(k\gamma_\ell, X_k^\ell) + (\mathbf{B}_t - \mathbf{B}_{k\gamma_\ell})E_k^\ell].
\end{aligned}
$$

Hence, using Lemma S18, we have that

$$
\begin{aligned}
d(\bar{\mathbf{X}}_t^{\ell+1}, \mathbf{X}_t^\ell)^2 &\leq (1 + C\kappa_k^2 \exp[4\kappa_k])d(X_k^\ell, X_{2k}^{\ell+1})^2 \\
&\quad + C\exp[4\kappa_k]\|\Gamma_{X_{2k}^{\ell+1}}^{X_k^\ell} v_k - u_k\|^2 + 2\langle w'(0), \Gamma_{X_{2k}^{\ell+1}}^{X_k^\ell} v_k - u_k\rangle,
\end{aligned}
\tag{S23}
$$

with $w : [0,1] \to \mathcal{M}$ a minimizing geodesic between $X_k^\ell$ and $X_{2k}^{\ell+1}$

$$
\begin{aligned}
\kappa_k &= \|u_k\| + \|v_k\|, \\
u_k^1 &= (t - k\gamma_\ell)b(k\gamma_\ell, X_k^\ell), \\
v_k^1 &= (t - k\gamma_\ell)b(k\gamma_\ell, X_{2k}^{\ell+1}), \\
u_k^2 &= (\mathbf{B}_t - \mathbf{B}_{k\gamma_\ell})E_k^\ell, \qquad\qquad v_k^2 = (\mathbf{B}_t - \mathbf{B}_{k\gamma_\ell})E_{2k}^{\ell+1}, \\
u_k &= u_k^1 + u_k^2, \qquad\qquad\qquad v_k = v_k^1 + v_k^2.
\end{aligned}
$$

In particular, since $E_k^\ell = \Gamma_{X_{2k}^{\ell+1}}^{X_k^\ell} E_{2k}^{\ell+1}$ using (S20) and $t - (2k+1)\gamma_{\ell+1} + \gamma_{\ell+1} = t - k\gamma_\ell$, we have that $u_k^2 = \Gamma_{X_{2k}^{\ell+1}}^{X_k^\ell} v_k^2$. Therefore, we get that

$$\|\Gamma_{X_{2k}^{\ell+1}}^{X_k^\ell} v_k^1 - u_k^1\| \leq \gamma_{\ell+1}\|b(k\gamma_\ell, X_k^\ell) - \Gamma_{X_{2k}^{\ell+1}}^{X_k^\ell} b(k\gamma_\ell, X_{2k}^{\ell+1})\|$$

$$\leq \gamma_\ell\|b(k\gamma_\ell, X_k^\ell) - \Gamma_{X_{2k}^{\ell+1}}^{X_k^\ell} b(k\gamma_\ell, X_{2k}^{\ell+1})\| + \mathtt{L}_2\gamma_\ell^2$$

$$\leq \mathtt{L}_1\gamma_\ell d(X_k^\ell, X_{2k}^{\ell+1}).$$

Therefore, we get that $\|u_k - v_k\| \leq \mathtt{L}_1\gamma_\ell d(X_k^\ell, X_{2k}^{\ell+1})$. In addition, we have that $\|w'(0)\| \leq d(X_k^\ell, X_{2k}^{\ell+1})$ since $w$ is a minimizing geodesic. Combining these results and (S23) we get that

$$d(\bar{\mathbf{X}}_t^{\ell+1}, \mathbf{X}_t^\ell)^2 \leq (1 + C\kappa_k^2 \exp[4\kappa_k])d(X_k^\ell, X_{2k}^{\ell+1})^2$$

$$+ C\exp[4\kappa_k]\mathtt{L}_1^2\gamma_\ell^2 d(X_k^\ell, X_{2k}^{\ell+1})^2$$

$$+ 2\mathtt{L}_1\gamma_\ell d(X_k^\ell, X_{2k}^{\ell+1})d(X_k^\ell, X_{2k}^{\ell+1})$$

$$\leq (1 + C\kappa_k^2 \exp[4\kappa_k] + 2C\exp[4\kappa_k]\mathtt{L}_1^2\gamma_\ell^2)d(X_k^\ell, X_{2k}^{\ell+1})^2$$

$$+ 2\mathtt{L}_1\gamma_\ell d(X_k^\ell, X_{2k}^{\ell+1})^2 + 2\mathtt{L}_2^2\gamma_\ell^4$$

$$\leq (1 + C\kappa_k^2 \exp[4\kappa_k] + 2C\exp[4\kappa_k]\mathtt{L}_1^2\gamma_\ell^2 + 2\mathtt{L}_1\gamma_\ell)d(X_k^\ell, X_{2k}^{\ell+1})^2.$$

Hence, there exists $C_1 \geq 0$ (not dependent on $k$ or $\ell$) such that for any $t \in [k\gamma_\ell, (k+1)\gamma_\ell]$

$$(1 + 2^{-\ell})d(\bar{\mathbf{X}}_t^{\ell+1}, \mathbf{X}_t^\ell)^2 \leq (1 + C_1\{\kappa_k^2\exp[4\kappa_k] + \gamma_\ell^2\exp[4\kappa_k] + 2^{-\ell}\})d(X_k^\ell, X_{2k}^{\ell+1})^2 + C_1\gamma_\ell^3. \tag{S24}$$

(b) We recall that if $t \in [k\gamma_\ell, (2k+1)\gamma_{\ell+1}]$ the second term in (S21) is zero. Therefore in what follows, we assume $t \in [(2k+1)\gamma_{\ell+1}, (k+1)\gamma_\ell]$. We introduce

$$\hat{\mathbf{X}}_t^{\ell+1} = \exp_{X_{2k+1}^{\ell+1}}[(t-(2k+1)\gamma_{\ell+1})\Gamma_{X_{2k}^{\ell+1}}^{X_{2k+1}^{\ell+1}} b((2k+1)\gamma_{\ell+1}, X_{2k}^{\ell+1}) + (\mathbf{B}_t - \mathbf{B}_{(2k+1)\gamma_{\ell+1}})E_{2k+1}^{\ell+1}]. \tag{S25}$$

In what follows, we provide an upper-bound for $d(\bar{\mathbf{X}}_t^{\ell+1}, \mathbf{X}_t^{\ell+1})$. First, we have that

$$d(\bar{\mathbf{X}}_t^{\ell+1}, \mathbf{X}_t^{\ell+1}) \leq d(\bar{\mathbf{X}}_t^{\ell+1}, \hat{\mathbf{X}}_t^{\ell+1}) + d(\hat{\mathbf{X}}_t^{\ell+1}, \mathbf{X}_t^{\ell+1}).$$

We recall that

$$\bar{\mathbf{X}}_t^{\ell+1} = \exp_{X_{2k}^{\ell+1}}[\gamma_{\ell+1}b(2k\gamma_{\ell+1}, X_{2k}^{\ell+1}) + (t-(2k+1)\gamma_{\ell+1})b((2k+1)\gamma_{\ell+1}, X_{2k}^{\ell+1}) + (\mathbf{B}_t - \mathbf{B}_{k\gamma_\ell})E_{2k}^{\ell+1}]. \tag{S26}$$

Denote $a_k, b_k$ such that

$$a_k = b(2k\gamma_{\ell+1}, X_{2k}^{\ell+1}) + (\mathbf{B}_{(2k+1)\gamma_{\ell+1}} - \mathbf{B}_{k\gamma_\ell})E_{2k}^{\ell+1},$$

$$b_k = (t-(2k+1)\gamma_{\ell+1})b((2k+1)\gamma_{\ell+1}, X_{2k}^{\ell+1}) + (\mathbf{B}_t - \mathbf{B}_{(2k+1)\gamma_{\ell+1}})E_{2k}^{\ell+1}.$$

Using (S20), (S25) and (S26) we have that

$$X_{2k+1}^{\ell+1} = \exp_{X_{2k}^{\ell+1}}[a_k], \qquad \hat{\mathbf{X}}_t^{\ell+1} = \exp_{X_{2k+1}^{\ell+1}}[\Gamma_{X_{2k}^{\ell+1}}^{X_{2k+1}^{\ell+1}} b_k], \qquad \bar{\mathbf{X}}_t^{\ell+1} = \exp_{X_{2k}^{\ell+1}}[a_k + b_k].$$

Using this result and (Sun et al., 2019, Lemma 3), there exists $C_2 \geq 0$ (not dependent on $k$ or $\ell$) such that

$$d(\hat{\mathbf{X}}_t^{\ell+1}, \bar{\mathbf{X}}_t^{\ell+1}) \leq C_2(\|a_k\| + \|b_k\|)^3.$$

Using this result and that for any $t \in [0, \gamma]$ and $x \in \mathcal{M}$, $\|b(t,x)\| \leq \mathtt{K}$ we get that there exists $C_3 \geq 0$ (not dependent on $k$ or $\ell$) such that

$$d(\hat{\mathbf{X}}_t^{\ell+1}, \bar{\mathbf{X}}_t^{\ell+1})^2 \leq C_3(\gamma_{\ell+1}^6 + \|\mathbf{B}_t - \mathbf{B}_{(2k+1)\gamma_{\ell+1}}\|^6 + \|\mathbf{B}_{(2k+1)\gamma_\ell} - \mathbf{B}_{(k+1)\gamma_\ell}\|^6). \tag{S27}$$

Finally, we recall that

$$\hat{\mathbf{X}}_t^{\ell+1} = \exp_{X_{2k+1}^{\ell+1}}[(t-(2k+1)\gamma_{\ell+1})\Gamma_{X_{2k}^{\ell+1}}^{X_{2k+1}^{\ell+1}} b((2k+1)\gamma_{\ell+1}, X_{2k}^{\ell+1}) + (\mathbf{B}_t - \mathbf{B}_{(2k+1)\gamma_{\ell+1}})E_{2k+1}^{\ell+1}],$$

$$\mathbf{X}_t^{\ell+1} = \exp_{X_{2k+1}^{\ell+1}}[(t-(2k+1)\gamma_{\ell+1})b((2k+1)\gamma_{\ell+1}, X_{2k+1}^{\ell+1}) + (\mathbf{B}_t - \mathbf{B}_{(2k+1)\gamma_{\ell+1}})E_{2k+1}^{\ell+1}].$$

Let us define

$$\tau_k = \|c_k\| + \|d_k\|,$$
$$c_k = c_k^1 + c_k^2, \qquad\qquad d_k = d_k^1 + d_k^2,$$
$$c_k^1 = (t - (2k+1)\gamma_{\ell+1})b((2k+1)\gamma_{\ell+1}, X_{2k+1}^{\ell+1}),$$
$$d_k^1 = (t - (2k+1)\gamma_{\ell+1})\Gamma_{X_{2k}^{\ell+1}}^{X_{2k+1}^{\ell+1}}b((2k+1)\gamma_{\ell+1}, X_{2k}^{\ell+1}),$$
$$c_k^2 = d_k^2 = (\mathbf{B}_t - \mathbf{B}_{(2k+1)\gamma_{\ell+1}})E_{2k+1}^{\ell+1}. \tag{S28}$$

Using Lemma S18, we get that

$$d(\mathbf{X}_t^{\ell+1}, \hat{\mathbf{X}}_t^{\ell+1})^2 \leq C\exp[4\tau_k]\|c_k - d_k\|^2 \leq C\mathtt{L}_2^2\gamma_{\ell+1}^2\exp[4\tau_k]d(X_{2k+1}^{\ell+1}, X_{2k}^{\ell+1})^2. \tag{S29}$$

In addition, using Lemma S18, we get that

$$d(X_{2k+1}^{\ell+1}, X_{2k}^{\ell+1})^2 \leq \exp[4\|e_k\|]\|e_k\|,$$

with $e_k = \gamma_{\ell+1}b(k\gamma_\ell, X_{2k}^{\ell+1}) + (\mathbf{B}_{(2k+1)\gamma_{\ell+1}} - \mathbf{B}_{k\gamma_\ell})E_{2k}^{\ell+1}$. Combining this result and (S29), we get that

$$d(\mathbf{X}_t^{\ell+1}, \hat{\mathbf{X}}_t^{\ell+1})^2 \leq C_3\gamma_{\ell+1}^2(\gamma_{\ell+1}^2 + \|\mathbf{B}_{(2k+1)\gamma_{\ell+1}} - \mathbf{B}_{k\gamma_\ell}\|^2)\exp[4\tau_k + \|e_k\|]. \tag{S30}$$

Combining (S27) and (S30), there exists $C_5$ such that

$$d(\bar{\mathbf{X}}_t^{\ell+1}, \mathbf{X}_t^{\ell+1})^2 \leq C_5\gamma_{\ell+1}^2(\gamma_{\ell+1}^2 + \|\mathbf{B}_{(2k+1)\gamma_{\ell+1}} - \mathbf{B}_{k\gamma_\ell}\|^2)\exp[4\tau_k + \|e_k\|]$$
$$+ C_5(\gamma_{\ell+1}^6 + \|\mathbf{B}_t - \mathbf{B}_{(2k+1)\gamma_{\ell+1}}\|^6 + \|\mathbf{B}_{(2k+1)\gamma_\ell} - \mathbf{B}_{(k+1)\gamma_\ell}\|^6). \tag{S31}$$

In what follows, we denote

$$\alpha_k = C_1\{(\kappa_k^+)^2\exp[4\kappa_k] + \gamma_\ell^2\exp[4\kappa_k^+] + 2^{-\ell}\}.$$
$$\beta_k = C_1\gamma_\ell^3 + C_5(1 + 2^\ell)\gamma_{\ell+1}^2(\gamma_{\ell+1}^2 + \|\mathbf{B}_{(2k+1)\gamma_{\ell+1}} - \mathbf{B}_{k\gamma_\ell}\|^2)\exp[4\tau_k^+ + \|e_k\|]$$
$$+ C_5(1 + 2^\ell)\,(\gamma_{\ell+1}^6 + \sup_{t\in[k\gamma_\ell,(k+1)\gamma_\ell]}\{\|\mathbf{B}_t - \mathbf{B}_{(2k+1)\gamma_{\ell+1}}\|^6\} + \|\mathbf{B}_{(2k+1)\gamma_\ell} - \mathbf{B}_{(k+1)\gamma_\ell}\|^6),$$

with $\tau_k^+ = \sup\{\|c_k\| + \|d_k\| \ : \ t \in [k\gamma_\ell, (k+1)\gamma_\ell]\}$, see (S28). Therefore, using (S21), (S24) and (S31), we get that for any $k \in \{0, \ldots, 2^\ell - 1\}$

$$U_{k+1} \leq (1 + \alpha_k)U_k + \beta_k.$$

Let $\{R_k\}_{k=-1}^{2^\ell}$ such that $R_{-1} = 0$ and for any $k \in \{0, \ldots, 2^\ell - 1\}$

$$R_{k+1} = (1 + \alpha_k)R_k + \beta_k.$$

Then, for any $k \in \{0, \ldots, 2^\ell - 1\}$, we have that $R_{2^\ell - 1} \geq R_k \geq U_k$. Therefore

$$\mathbb{E}[R_{2^\ell}] \geq \mathbb{E}[\sup\{U_k \ : \ k \in \{0, \ldots, 2^\ell\}\}] \geq \mathbb{E}[\sup\{d(\mathbf{X}_t^\ell, \mathbf{X}_t^{\ell+1})^2 \ : \ t \in [0, \gamma]\}]. \tag{S32}$$

In addition, using that for any $k \in \{0, \ldots, 2^\ell - 1\}$, $\mathbb{E}[\alpha_k|\mathcal{F}_k] = \bar{\alpha}_k$ and $\mathbb{E}[\beta_k|\mathcal{F}_k] = \bar{\beta}_k$ are constant, where $\mathcal{F}_k = \sigma(\{\mathbf{B}_t \ : \ t \in [0, k\gamma_\ell]\})$. Therefore, we get that for any $k \in \{0, \ldots, 2^\ell - 1\}$

$$\mathbb{E}[R_{k+1}] = (1 + \bar{\alpha}_k)\mathbb{E}[R_k] + \bar{\beta}_k.$$

Therefore, using the discrete Grönwall lemma we get that for any $k \in \{0, \ldots, 2^\ell - 1\}$

$$\mathbb{E}[R_{2^\ell}] \leq \bar{\beta}_{2^\ell - 1} + \exp[\sum_{n=0}^{2^\ell - 1}\bar{\alpha}_n]\sum_{j=0}^{2^\ell - 1}\bar{\beta}_j\bar{\alpha}_j.$$

In addition, there exists $C_8 \geq 0$ such that for any $k \in \{0, \ldots, 2^\ell\}$, $\bar{\alpha}_k \leq C_8 2^{-\ell}$ and $\bar{\beta}_k \leq C_8\gamma^3 2^{-2\ell}$. Hence, there exists $C_9 \geq 0$ such that

$$\mathbb{E}[R_{2^\ell}] \leq C_9\gamma^3 2^{-2\ell},$$

which concludes the proof upon using (S32).

$\square$

**Proposition S20.** *Assume* **A**1. *Then, there exists* $(\mathbf{X}_t)_{t \in [0,\gamma]}$ *such that* $\lim_{\ell \to +\infty} \sup\{d(\mathbf{X}_t^\ell, \mathbf{X}_t) : t \in [0,\gamma]\} = 0$ *and* $(\mathbf{X}_t)_{t \in [0,\gamma]}$ *is a weak solution to* $\mathrm{d}\mathbf{X}_t = b(t, \mathbf{X}_t)\mathrm{d}t + \mathrm{d}\mathbf{B}_t^{\mathcal{M}}$.

*Proof.* The proof is a straightforward application of Proposition S19 and (Cheng et al., 2022, A.1 (Step 2 and Step 3), A.2). □

**Proposition S21.** *Assume* **A**1. *Then, there exists* $C \geq 0$ *such that* $\mathbb{E}\left[d(X_1^0, \mathbf{X}_\gamma)^2 \leq C\gamma^{3/2}\right]$.

*Proof.* Using Proposition S19, there exists $C \geq 0$ such that for any $\ell \in \mathbb{N}$

$$\mathbb{E}[\sup_{t \in [0,\gamma]} d(\mathbf{X}_t^\ell, \mathbf{X}_t^{\ell+1})] \leq C\gamma^{3/2}2^{-\ell}.$$

Therefore, combining this result and Proposition S20 we get that for any $\ell \in \mathbb{N}$

$$\mathbb{E}[\sup_{t \in [0,\gamma]} d(\mathbf{X}_t^\ell, \mathbf{X}_t)] \leq 2C\gamma^{3/2},$$

which concludes the proof. □

Finally, we consider the two following processes $(X_k^1, X_k^2)_{k \in \mathbb{N}}$ such that for any $k \in \mathbb{N}$ and $i \in \{1, 2\}$

$$X_{k+1}^i = \exp_{X_k^i}[\gamma b(k\gamma, X_k^i) + \sqrt{\gamma}E_k^i Z_k],$$

where $\{Z_k\}_{k \in \mathbb{N}}$ is a family of independent Gaussian random variables with zero mean and identity covariance matrix, and for any $k \in \mathbb{N}$, $E_k^1$ is a frame for $\mathrm{T}_{X_k^1}\mathcal{M}$ and $E_k^2 = \Gamma_{X_k^1}^{X_k^2}E_k^1$.

**Proposition S22.** *Assume* **A**1. *Then, there exists* $C \geq 0$ *such that for any* $k \in \mathbb{N}$

$$\mathbb{E}\left[d(X_k^1, X_k^2)\right] \leq \exp[Ck\gamma]\mathbb{E}\left[d(X_0^1, X_0^2)\right].$$

*Proof.* Let $k \in \mathbb{N}$. Using Lemma S18, there exists $D \geq 0$ such that

$$\begin{aligned}
d(X_{k+1}^1, X_{k+1}^2)^2 &\leq (1 + D\kappa_k^2 \exp[4\kappa_k])d(X_k^1, X_k^2)^2 \\
&\quad + D\exp[4\kappa_k]\|\Gamma_{X_k^2}^{X_k^1}v_k - u_k\|^2 + 2\langle w'(0), \Gamma_{X_k^2}^{X_k^1}v_k - u_k \rangle,
\end{aligned}$$

with $w : [0,1] \to \mathcal{M}$ a minimizing geodesic between $X_k^1$ and $X_k^2$

$$\begin{aligned}
\kappa_k &= \|u_k\| + \|v_k\|, \\
u_k^1 &= \gamma b(k\gamma, X_k^1), \\
v_k^1 &= \gamma b(k\gamma, X_k^2), \\
u_k^2 &= \sqrt{\gamma}Z_k E_k^1, &\quad v_k^2 &= \sqrt{\gamma}Z_k E_k^2, \\
u_k &= u_k^1 + u_k^2, &\quad v_k &= v_k^1 + v_k^2.
\end{aligned}$$

We have that $\Gamma_{X_k^2}^{X_k^1}v_k^2 = v_k$ and

$$\|\Gamma_{X_k^2}^{X_k^1}v_k^1 - u_k^1\| \leq \mathrm{L}_1\gamma d(X_k^1, X_k^2).$$

In addition, $\|w'(0)\| \leq d(X_k^1, X_k^2)$. Therefore, we get that

$$d(X_{k+1}^1, X_{k+1}^2)^2 \leq (1 + D\kappa_k^2 \exp[4\kappa_k] + D\gamma^2 \exp[4\kappa_k] + 2\gamma)d(X_k^1, X_k^2)^2.$$

Hence, using that for any $t \geq 0$, $\sqrt{1+t} \leq 1 + t/2$, we have

$$d(X_{k+1}^1, X_{k+1}^2) \leq (1 + D\kappa_k^2 \exp[4\kappa_k] + D\gamma^2 \exp[4\kappa_k] + 2\gamma)d(X_k^1, X_k^2).$$

Therefore, we get that there exists $C \geq 0$ such that

$$\mathbb{E}[d(X_{k+1}^1, X_{k+1}^2)] \leq (1 + C\gamma)\mathbb{E}[d(X_k^1, X_k^2)],$$

which concludes the proof. □

# J  Proof of Proposition 3

*Proof.* Let $t \in (0, T]$ and $s_t \in C^\infty(\mathcal{M})$. Using the divergence theorem (see Lee, 2018, p.51), we have

$$
\begin{aligned}
\ell_{t|s}(s_t) &= \int_{\mathcal{M} \times \mathcal{M}} \|\nabla \log p_{t|s}(x_t|x_s)\|^2 \mathrm{d}\mathbb{P}_{s,t}(x_s, x_t) + \int_{\mathcal{M}} \|s_t(x_t)\|^2 \mathrm{d}\mathbb{P}_t(x_t) \\
&\quad - 2 \int_{\mathcal{M} \times \mathcal{M}} \langle \nabla \log p_{t|s}(x_t|x_s), s_t(x_t) \rangle_{\mathcal{M}} \mathrm{d}\mathbb{P}_{s,t}(x_s, x_t) \\
&= \int_{\mathcal{M} \times \mathcal{M}} \|\nabla \log p_{t|s}(x_t|x_s)\|^2 \mathrm{d}\mathbb{P}_{s,t}(x_s, x_t) + \int_{\mathcal{M}} \|s_t(x_t)\|^2 \mathrm{d}\mathbb{P}_t(x_t) \\
&\quad - 2 \int_{\mathcal{M} \times \mathcal{M}} \langle \nabla \log p_{t|s}(x_t|x_s), s_t(x_t) \rangle_{\mathcal{M}} p_{t|s}(x_t|x_s) p_s(x_s) \mathrm{d}(p_{\mathrm{ref}} \otimes p_{\mathrm{ref}})(x_s, x_t) \\
&= \int_{\mathcal{M} \times \mathcal{M}} \|\nabla \log p_{t|s}(x_t|x_s)\|^2 \mathrm{d}\mathbb{P}_{s,t}(x_s, x_t) + \int_{\mathcal{M}} \|s_t(x_t)\|^2 \mathrm{d}\mathbb{P}_t(x_t) \\
&\quad - 2 \int_{\mathcal{M}} \{ \int_{\mathcal{M}} \langle \nabla p_{t|s}(x_t|x_s), s_t(x_t) \rangle_{\mathcal{M}} \mathrm{d}p_{\mathrm{ref}}(x_t) \} p_s(x_s) \mathrm{d}p_{\mathrm{ref}}(x_s) \\
&= \int_{\mathcal{M} \times \mathcal{M}} \|\nabla \log p_{t|s}(x_t|x_s)\|^2 \mathrm{d}\mathbb{P}_{s,t}(x_s, x_t) + \int_{\mathcal{M}} \|s_t(x_t)\|^2 \mathrm{d}\mathbb{P}_t(x_t) \\
&\quad + 2 \int_{\mathcal{M}} \{ \int_{\mathcal{M}} \mathrm{div}(s_t)(x_t) p_{t|s}(x_t|x_s) \mathrm{d}p_{\mathrm{ref}}(x_t) \} p_s(x_s) \mathrm{d}p_{\mathrm{ref}}(x_s),
\end{aligned}
$$

which concludes the proof. $\qquad\square$

# K  Comparison with Moser flows

In this section, we compare ourselves with Rozen et al. (2021) in greater details. Rozen et al. (2021) also aims at interpolating between a reference distribution $p_{\mathrm{ref}}$ and a target distribution $p_0$. We assume that we have access to the density $p_{\mathrm{ref}}$ and that we know how to sample form $p_{\mathrm{ref}}$ (which is often the case if $p_{\mathrm{ref}}$ is the uniform distribution on $\mathcal{M}$). Contrary to RSGM, $p_{\mathrm{ref}}$ is not necessary the uniform distribution.

We then consider the following interpolation $\hat{p}_t = (1 - t)\hat{p}_0 + t\hat{p}_1$, with $\hat{p}_0 = p_{\mathrm{ref}}$ and $\hat{p}_1 = p_0$. Let $(\mathbf{X}_t)_{t \in [0,1]}$ be given by $\mathbf{X}_0 \sim \hat{p}_0$ and $\mathrm{d}\mathbf{X}_t = \mathbf{v}_t(\mathbf{X}_t)\mathrm{d}t$ where for any $t \in [0, 1]$, $\mathbf{v}_t = \mathbf{u}/((1 - t)\hat{p}_0 + \hat{p}_1)$, with $\mathrm{div}(\mathbf{u}) = \hat{p}_0 - \hat{p}_1$. Using the Fokker-Planck equation, we have that for any $t \in [0, 1]$, $\mathbf{X}_t \sim \hat{p}_t$. In Rozen et al. (2021), $\mathbf{u}$ is replaced by a parametric version $\mathbf{u}_\theta$ and the authors optimize the loss

$$
\ell(\theta) = \mathbb{E}[(\hat{p}_0 - \mathrm{div}(\mathbf{u}_\theta))^{+,\varepsilon}(\mathbf{X}_1)] + \lambda \int_{\mathcal{M}} (\hat{p}_0 - \mathrm{div}(\mathbf{u}_\theta))^{-,\varepsilon}(x)\mathrm{d}x,
$$

with $\lambda, \varepsilon > 0$ and for any $f : \mathcal{M} \to \mathbb{R}$, $f^{+,\varepsilon} = \max(f, \varepsilon)$ and $f^{-,\varepsilon} = \varepsilon - \min(f, \varepsilon)$. Given $\mathbf{u}_\theta$, we then consider $(\mathbf{X}_t^\theta)_{t \in [0,1]}$ such that $\mathrm{d}\mathbf{X}_t^\theta = \mathbf{v}_t^\theta(\mathbf{X}_t^\theta)\mathrm{d}t$, where for any $t \in [0, 1]$, $\mathbf{v}_t^\theta = \mathbf{u}_\theta/(\hat{p}_0 + t\mathrm{div}(\mathbf{u}_\theta))$. Note that $\mathbf{u}^\theta$ also enables density estimation using that $\hat{p}_1 = \hat{p}_0 - \mathrm{div}(\mathbf{u}^\theta)$. Density estimation is not directly accessible using RSGM, however in App. L we propose a way to perform such an estimation using Fisher score in a manner akin to Choi et al. (2021).

Let $\hat{p}_0 = p_{\mathrm{ref}}$ to be the uniform distribution on $\mathcal{M}$. As RSGM, Moser flow defines a continuous time interpolation between $p_0$ and $p_{\mathrm{ref}}$. One major difference between the two approaches is that Moser flows perform the interpolation in *density space*, i.e. $\hat{p}_t = (1 - t)\hat{p}_0 + t\hat{p}_1$ for any $t \in [0, 1]$, whereas RSGM performs the interpolation in *sample space*, i.e. $p_t = \int_{\mathcal{M}} p_0(y) p_{t|0}(y, x) \mathrm{d}p_{\mathrm{ref}}(y)$. Interpolation in the *density space* results in spontaneous creation of density, whereas interpolation in *sample space* corresponds to a displacement of the density, see Figs. S3a and S3b. In that respect, Moser flows can be seen as *vertical displacement* whereas RSGM corresponds to *horizontal displacement*, see Santambrogio (2017). The drawback with the 'spontaneous creation of density' of Moser flows, is that when solving trajectories in *sample space*—for sampling or likelihood evaluation purposes—the Stein score's amplitude can get extremely high in settings where the reference and target distributions have little overlap as shown on Fig. S3c.

# L  Density estimation with Fisher score

In this section, we show how we can adapt ideas from Choi et al. (2021) for density estimation on $\mathcal{M}$ using the Fisher score. The main idea of using Fisher score is to leverage the following decomposition for any $x \in \mathcal{M}$

$$
\log p_0(x) = \log p_T(x) - \int_0^T \partial_t \log p_t(x)\mathrm{d}t.
$$

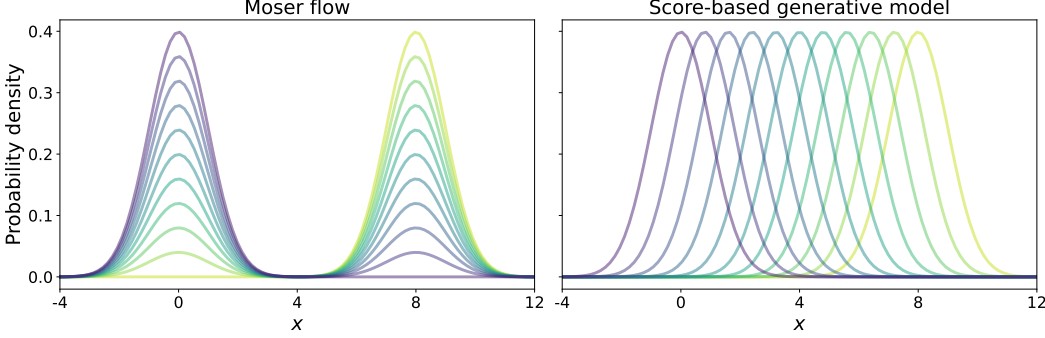

(a) Interpolated density between the reference $p_{\mathrm{ref}} = \mathrm{N}(0,1)$ and target $p_0 = \mathrm{N}(8,1)$ distributions.

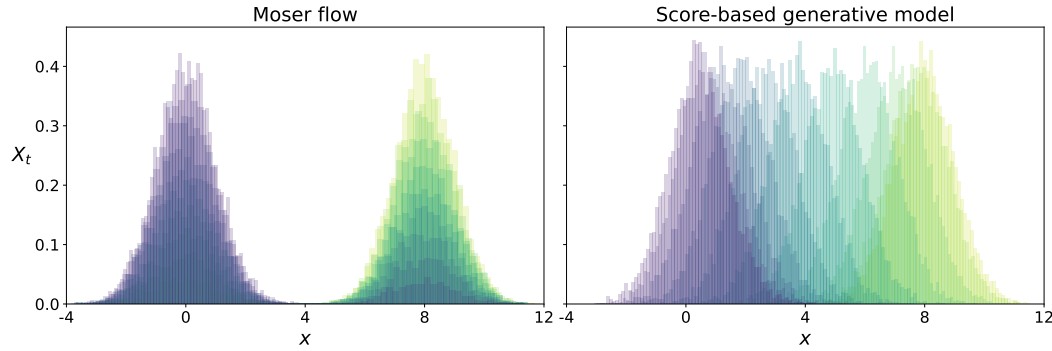

(b) Interpolated histograms between the reference $p_{\mathrm{ref}} = \mathrm{N}(0,1)$ and target $p_0 = \mathrm{N}(8,1)$ distributions.

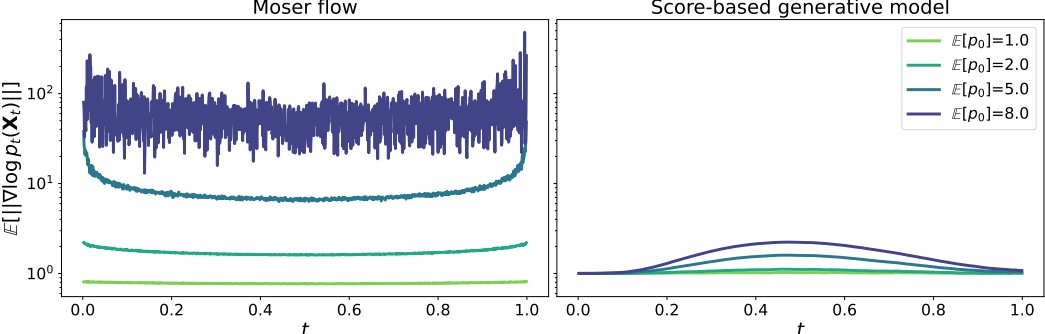

(c) Expected norm of the Stein score along trajectories interpolating between reference and target $p_0 = \mathrm{N}(a,1)$ distributions for different target mean.

Figure S3: The reference distribution is $p_{\mathrm{ref}} = \mathrm{N}(0,1)$.

Assume that an approximation $\hat{\mathbf{s}}_\theta$ of $\partial_t \log p_t$ (the Fisher score) is available then we have that for any $x \in \mathcal{M}$

$$\log p_0(x) \approx \log p_{\mathrm{ref}}(x) - \int_0^T \hat{\mathbf{s}}_\theta(x)\mathrm{d}t.$$

Before turning to our main result, we state the following lemma.

**Lemma S23.** *Assume* **A**1. *Then, there exists $C, T_0 \geq 0$ such that for any $x \in \mathcal{M}$ and $T \geq T_0$, $|p_T(x) - 1| \leq C \exp[-\lambda_1 T/2]$, where $\lambda_1$ is the first non-negative eigenvalue of $-\Delta_{\mathcal{M}}$ in $\mathrm{L}^2(p_{\mathrm{ref}})$.*

*Proof.* First, using Proposition S10, there exists $C_0 \geq 0$ such that for any $T \geq 1/2$ we have

$$\int_{\mathcal{M}} |p_T(x) - 1|\mathrm{d}p_{\mathrm{ref}}(x) \leq C_0 \mathrm{e}^{-\lambda_1 T}.$$

Using (Grigor'yan, 1999, Corollary 5.5), (Hsu, 1999, Theorem 1.2)and the fact that $\mathcal{M}$ is compact, there exists $C_1, \beta \geq 0$ such that for any $T \geq 1/2$ and $x_0, x_T \in \mathcal{M}$

$$\|\nabla p_{T|0}(x_T|x_0)\| \leq C_1(1 + T^\beta). \tag{S33}$$

In addition, using (Croke, 1980, Proposition 14) we have that there exists $C_2, r_0 > 0$ such that for any $x_0 \in \mathcal{M}$ and $r \in (0, r_0)$

$$\int_{\bar{\mathrm{B}}(x_0, r)} \mathrm{d}p_{\mathrm{ref}}(x) \geq C_2 r^d. \tag{S34}$$

Assume that that $\int_{\mathcal{M}} |p_T(x) - 1| \mathrm{d}p_{\mathrm{ref}}(x) \leq \varepsilon$ and that there exists $x_0 \in \mathcal{M}$ such that $|p_T(x) - 1| > \kappa\varepsilon$ with $\kappa > 0$ and let $T \geq T_0$ with $T_0 = (\kappa\varepsilon/(2C_1))^{1/\beta}$. Then, using (S33) and (S34), we have for any $r \in (0, r_0)$

$$\varepsilon \geq \int_{\bar{\mathrm{B}}(0, r)} |p_T(x) - 1| \geq C_2 r^d (\kappa\varepsilon - C_1(1 + T^\beta)r).$$

Since $\kappa\varepsilon/(2C_1(1 + T^\beta)) \in (0, r_0)$ we have

$$\varepsilon \geq C_2(\kappa\varepsilon)^{d+1}/(4C_1(1 + T^\beta)).$$

Therefore, we get that

$$\varepsilon \geq C_2(\kappa\varepsilon)^{d+1}/(4C_1(1 + T^\beta)).$$

Therefore, we get that $\kappa \leq (4C_1(1 + T^\beta)/C_2)^{1/(d+1)}\varepsilon^{-1/(d+1)}$. Therefore, we have that for any $x \in \mathcal{M}$

$$|p_T(x) - 1| \leq (8C_1(1 + T^\beta)/C_2)^{1/(d+1)}\varepsilon^{1-1/(d+1)}. \tag{S35}$$

Let $T_0 \geq 0$ such that for any $T \geq T_0$ we have

$$(8C_1(1 + T^\beta)/C_2)^{1/(d+1)}C_0^{1-1/(d+1)}\mathrm{e}^{-(1-1/(d+1))\lambda_1 T} \leq 2^{1-\beta}C_1.$$

Combining this result and (S36), we get that for any $x \in \mathcal{M}$ and $T \geq 0$

$$|p_T(x) - 1| \leq (8C_1(1 + T^\beta)/C_2)^{1/(d+1)}C_0^{1-1/(d+1)}\mathrm{e}^{-(1-1/(d+1))\lambda_1 T}, \tag{S36}$$

which concludes the proof. $\qquad\square$

The following proposition quantifies this approximation.

**Proposition S24.** *Assume* **A**1 *and that* $p_0 \in \mathrm{C}^\infty(\mathcal{M}, (0, +\infty))$. *Let* $x_0 \in \mathcal{M}$ *and assume that for any* $t \in [0, T]$, $|\hat{\mathbf{s}}_\theta(t, x_0) - \partial_t \log p_t(x_0)| \leq \mathtt{M}$ *with* $\mathtt{M} \geq 0$. *Then, there exists* $C, T_0 \geq 0$ *such that for any* $T \geq 0$

$$|\log p_0(x_0) - \int_0^T \hat{\mathbf{s}}_\theta(t, x_0)\mathrm{d}t| \leq C\exp[-\lambda_1 T/2] + \mathtt{M}T.,$$

*where* $\lambda_1$ *is the first non-negative eigenvalue of* $-\Delta_{\mathcal{M}}$ *in* $\mathrm{L}^2(p_{\mathrm{ref}})$.

*Proof.* First using, Lemma S23, there exists $C_0, T_0^{(a)} \geq 0$ such that for any $T \geq T_0^{(a)}$

$$|p_T(x_0) - 1| \leq C_0\exp[-\lambda_1 T/2].$$

Let $T_0^{(b)} = |\log(C_0)|/\lambda_1$. Using that for any $s \in [1/2, +\infty)$ we have that $|\log(1 + s)| \leq 2\log(2)|s|$ we get that for any $T \geq \max(T_0^{(a)}, T_0^{(b)})$

$$|\log p_T(x_0)| \leq 2\log(2)C_0\exp[-\lambda_1 T/2],$$

which concludes the proof. $\qquad\square$

In practice, we do not have access to $\partial_t \log p_t$. However, following (Choi et al., 2021, Proposition 2), we have the following property.

**Proposition S25.** *Let* $\hat{\mathbf{s}}$ *such that for any* $t \in [0, T]$ *and* $x \in \mathcal{M}$, $\hat{\mathbf{s}}(t, x) = \partial_t \log p_t(x)$. *Then, we have that* $\hat{\mathbf{s}} = \arg\min\{L(\mathbf{s}) : \mathbf{s} \in \mathrm{C}^\infty([0, T] \times \mathcal{M}, \mathbb{R})\}$, *where for any* $\mathbf{s} \in \mathrm{C}^\infty([0, T] \times \mathcal{M}, \mathbb{R})$ *we have*

$$L(\mathbf{s}) = (1/2)\mathbb{E}[\int_0^T \lambda(t)\mathbf{s}(t, \mathbf{X}_t)\mathrm{d}t] + \mathbb{E}[\int_0^T \lambda(t)\partial_t\mathbf{s}(t, \mathbf{X}_t)\mathrm{d}t]$$
$$+ \mathbb{E}[\int_0^T \partial_t\lambda(t)\partial_t\mathbf{s}(t, \mathbf{X}_t)\mathrm{d}t] + \mathbb{E}[\lambda(0)\mathbf{s}(0, \mathbf{X}_0)] - \mathbb{E}[\lambda(T)\mathbf{s}(T, \mathbf{X}_T)],$$

*where* $\lambda \in \mathrm{C}^\infty([0, T], \mathbb{R})$ *is a weighting function.*

*Proof.* For any $t \in [0, T]$ and $x_t \in \mathcal{M}$ we have

$$\hat{\mathbf{s}}(x_t) = \int_{\mathcal{M}} \partial_t \log p_{t|0}(x_t|x_0) p_{0|t}(x_0|x_t) \mathrm{d}x_0.$$

Hence, since $\mathcal{M}$ is compact and $\hat{\mathbf{s}} \in \mathrm{C}^\infty([0,T] \times \mathcal{M}, \mathbb{R})$, we have that $\hat{\mathbf{s}} = \arg\min\{L_0(\mathbf{s}) : \mathbf{s} \in \mathrm{C}^\infty([0,T] \times \mathcal{M}, \mathbb{R})\}$ where for any $\mathbf{s} \in \mathrm{C}^\infty([0,T] \times \mathcal{M}, \mathbb{R})$ we have

$$L_0(\mathbf{s}) = \int_0^T \lambda(t) \int_{\mathcal{M} \times \mathcal{M}} (\mathbf{s}(t, x_t) - \partial_t \log p_{t|0}(x_t|x_0))^2 \mathrm{d}p_{0,t}(x_0, x_t) \mathrm{d}t \qquad \text{(S37)}$$
$$= \int_0^T \lambda(t) \int_{\mathcal{M}} \mathbf{s}(t, x_t)^2 \mathrm{d}p_t(x_t) \mathrm{d}t - 2 \int_0^T \lambda(t) \int_{\mathcal{M} \times \mathcal{M}} \mathbf{s}(t, x_t) \partial_t \log p_{t|0}(x_0, x_t) \mathrm{d}p_{0,t}(x_0, x_t) \mathrm{d}t$$
$$+ \int_0^T \lambda(t) \int_{\mathcal{M}} \mathrm{d}p_t(x_t) \mathrm{d}t$$

In addition, we have that

$$\int_0^T \lambda(t) \int_{\mathcal{M} \times \mathcal{M}} \mathbf{s}(t, x_t) \partial_t \log p_{t|0}(x_t|x_0) \mathrm{d}p_{0,t}(x_0, x_t) \mathrm{d}t$$
$$= \int_0^T \int_{\mathcal{M} \times \mathcal{M}} \lambda(t) \mathbf{s}(t, x_t) \partial_t p_{t|0}(x_t) \mathrm{d}p_0(x_0) \mathrm{d}p_{\mathrm{ref}}(x_t) \mathrm{d}t.$$

By integration by parts we get

$$\int_0^T \int_{\mathcal{M} \times \mathcal{M}} \lambda(t) \mathbf{s}(t, x_t) \partial_t p_{t|0}(x_t) \mathrm{d}p_0(x_0) \mathrm{d}p_{\mathrm{ref}}(x_t) \mathrm{d}t$$
$$= -\int_0^T \int_{\mathcal{M} \times \mathcal{M}} \partial_t(\lambda(t) \mathbf{s}(\cdot, x_t))(t) \mathrm{d}p_{0,t}(x_0, x_t) \mathrm{d}t$$
$$+ \lambda(T) \int_{\mathcal{M}} \mathbf{s}(T, x_T) \mathrm{d}p_T(x_T) - \int_{\mathcal{M}} \mathbf{s}(0, x_0) \mathrm{d}p_0(x_0)$$
$$= -\int_0^T \int_{\mathcal{M} \times \mathcal{M}} \partial_t \lambda(t) \mathbf{s}(t, x_t) \mathrm{d}p_t(x_t) \mathrm{d}t - \int_0^T \int_{\mathcal{M} \times \mathcal{M}} \lambda(t) \partial_t \mathbf{s}(t, x_t) \mathrm{d}p_t(x_t) \mathrm{d}t$$
$$+ \lambda(T) \int_{\mathcal{M}} \mathbf{s}(T, x_T) \mathrm{d}p_T(x_T) - \lambda(0) \int_{\mathcal{M}} \mathbf{s}(0, x_0) \mathrm{d}p_0(x_0)$$

Combining this result and (S37) we get that

$$L_0(\mathbf{s}) = \int_0^T \lambda(t) \int_{\mathcal{M} \times \mathcal{M}} (\mathbf{s}(t, x_t) - \partial_t \log p_{t|0}(x_t|x_0))^2 \mathrm{d}p_{0,t}(x_0, x_t) \mathrm{d}t$$
$$= \int_0^T \lambda(t) \int_{\mathcal{M}} \mathbf{s}(t, x_t)^2 \mathrm{d}p_t(x_t) \mathrm{d}t + 2 \int_0^T \int_{\mathcal{M} \times \mathcal{M}} \partial_t \lambda(t) \mathbf{s}(t, x_t) \mathrm{d}p_t(x_t) \mathrm{d}t$$
$$+ 2 \int_0^T \int_{\mathcal{M} \times \mathcal{M}} \lambda(t) \partial_t \mathbf{s}(t, x_t) \mathrm{d}p_t(x_t) \mathrm{d}t - \lambda(T) \int_{\mathcal{M}} \mathbf{s}(T, x_T) \mathrm{d}p_T(x_T)$$
$$+ \lambda(0) \int_{\mathcal{M}} \mathbf{s}(0, x_0) \mathrm{d}p_0(x_0) + \int_0^T \lambda(t) \int_{\mathcal{M}}^2 \mathrm{d}p_t(x_t) \mathrm{d}t,$$

which concludes the proof. $\qquad\square$

Hence, using Proposition S25, we could estimate jointly the spatial (or Stein) score used in RSGM and the Fisher score considered in this section, see Choi et al. (2021).

## M   Extensions

### M.1   Schrödinger bridge.

For Euclidean SGM, the generative model is given by an approximation of the time-reversal of the noising dynamics $(\mathbf{X}_t)_{t \in [0,T]}$ while the backward dynamics $(\mathbf{Y}_t)_{t \in [0,T]}$ is initialized with the invariant distribution of the noising dynamics (the uniform distribution $p_{\mathrm{ref}}$ in case of RSGM). However, in order for the method to yield good results we need $\mathcal{L}(\mathbf{Y}_0) \approx \mathcal{L}(\mathbf{X}_T)$ (see De Bortoli et al., 2021, Theorem 1). Usually, this requires the number of steps in the backward process to be large in order to keep $T$ large and $\gamma$ small (where $\gamma > 0$ is the stepsize in the GRW). Another limitation of SGM is that existing methods target an easy-to-sample reference distribution. Hence, classical SGM cannot interpolate between two distributions defined by datasets. To circumvent this problem, one can consider a process whose initial and terminal distribution are pinned down using Schrödinger bridges (Schrödinger, 1932; Léonard, 2012a; Chen et al., 2016; De Bortoli et al., 2021; Vargas et al., 2021).

## M.2 Conditional RSGM.

Another extension of interest is conditional sampling. By amortizing SGM with respect to an observation $y$ it is possible to approximately sample from a given posterior distribution. In the Euclidean setting this idea has been successfully applied for several image processing problems such as deblurring, denoising or inpainting (see for instance Kawar et al., 2021a,b; Lee et al., 2021; Sinha et al., 2021; Batzolis et al., 2021; Chung et al., 2021). Similarly, RSGM can be amortized to handle such situations in the case where the underlying posterior distribution is supported on a manifold. Practically, this requires for the score network takes an additional input, i.e $s_\theta(t, x; y)$.

## M.3 Invariant distributions

In what follows, we propose an extension for modelling probability distributions which known invariance. That is, we assume that $p_0(\rho(g)x) = p_0(x)$ for all $g \in G$, with $G$ a group and $\rho : G \to \mathrm{GL}_n(\mathbb{R})$ a representation. Following Köhler et al. (2020), we have that if $p_{\mathrm{ref}}$ is invariant w.r.t. $G$ and $\phi : \mathcal{M} \to \mathcal{M}$ is equivariant w.r.t. to $G$, then the pushforward probability density $p = p_{\mathrm{ref}} \circ \phi^{-1}$ is invariant w.r.t. $G$.

Let's consider the probability flow $\phi$ associated with the reverse diffusion (4)—given by $\mathrm{d}\mathbf{Y}_t = \{-b(\mathbf{Y}_t) + \nabla \log p_{T-t}(\mathbf{Y}_t)\}\mathrm{d}t + \mathrm{d}\mathbf{B}_t^{\mathcal{M}}$— i.e. the solution of the following ODE (see App. D)
$$\mathrm{d}\mathbf{Y}_t = \{-b(\mathbf{Y}_t) + \tfrac{1}{2} \nabla \log p_{T-t}(\mathbf{Y}_t)\}\mathrm{d}t.$$

In practice, the Stein score $\nabla \log p_t$ is approximated with the score network $s_\theta(t, \cdot)$. It is sufficient to parametrize the score network so that it is equivariant w.r.t. its second argument —assuming that $\rho(g)$ and the drift $b$ commute (e.g. which is true for a linear drift)—since we then have
$$\left[-b + \tfrac{1}{2} s_\theta(T-t, \cdot)\right](\rho(g)\mathbf{Y}_t) = \rho(g)\left[-b + \tfrac{1}{2} s_\theta(T-t, \cdot)\right](\mathbf{Y}_t).$$

# N   Stereographic baseline details

In the experiments on the sphere we compare to a Stereographic Score-Based baseline model. This model is an alternative to the RSGM the we propose in order to construct score-based models on manifolds without having to construct the intrinsic approach presented in the paper as Riemannian Score-Based models. They can be applied to more cases than just the sphere.

In general these models work as follows:

1. Project the datapoints from the manifold to Euclidean space through a invertible[11] function $f : \mathcal{M} \to \mathbb{R}^d$.
2. Train a Euclidean score-based generative model on the datapoints projected to Euclidean space, giving a density $p_\theta$ on $\mathbb{R}^d$ (where $\theta$ are the parameters of the density).
3. Define the density on the manifold as the pushforward of the density in Euclidean space under the inverse of the bijection, $P_{\theta,\mathcal{M}} = f_*^{-1} p_\theta$.

One could also apply these models to the torus. By using the bijection $f : \theta \mapsto \tan(\theta)$ we can project each coordinate onto the real line.

In general we found that these models perform less well than their intrinsic counterparts. In order to map density near the seams of the bijection, it requires the model to send data points off to infinity in the Euclidean space. This is numerically challenging and leaves artefacts in the pushforward density on the manifold. In addition, these methods depend on the bijection used to project the data into a Euclidean space and therefore are not intrinsic.

# O   Experimental details

In what follows we describe the experimental settings used to generate results introduced in Sec. 6. The models and experiments have been implemented in Jax (Bradbury et al., 2018), using a modified version of the Riemannian geometry library Geomstats (Miolane et al., 2020b).

---

[11]Note that this may not be a bijection. For example for the sphere we use the stereographic projection of the earth onto the plane, which misses out a single point, opposite the projection point.

Anonymized code can be found at here[12]. Due to difficulties referencing anonymized repositories, the modified version of geomstats is included as a zip file in the supplementary material. Additionally modified versions of the `submitit` and `hydra-submitit-launcher` packages are not supplied for the same reasons, but the default versions of these will suffice for most users. Full code and all repos will be publicly available after publication.

**Models**   Following Song et al. (2021), the score-based generative models (SGMs) diffusion coefficient is parametrized as $g(t) = \sqrt{\beta(t)}$ with $\beta : t \mapsto \beta_{\min} + (\beta_{\max} - \beta_{\min}) \cdot t$.

**Architecture**   The architecture of the score network $s_\theta$ is given by a multilayer perceptron with $5$ hidden layers for the Earth and $SO(3)$ experiments, and $3$ for the high-dimension experiments with $512$ units each. We use sinusoidal activation functions. We decompose the output of the score network on the set of divergence free vector fields as per Sec. 3.4.

**Loss**   Where not specified, SGMs are trained with the sliced score matching (SSM) loss $\ell_t^{\mathrm{im}}$, relying on the Hutchinson estimator for computing the divergence with Rademacher noise described in Sec. 3.4. We found that training with the denoising score matching (DSM) loss $\ell_{t|0}$ gave similar results. Regarding the weighting function, for DSM loss $\ell_{t|0}$ we use $\lambda_t = \mathrm{Var}[X_t|X_0]$ (where we rely on the closed-form standard deviation available in the Euclidean setting as a proxy for the compact manifold setting), while for the ISM/SSM losses $\ell_t^{\mathrm{im}}$ we use $\lambda_t = g(t)^2 = \beta(t)$.

**Optimization**   All models are trained by the stochastic optimizer Adam (Kingma and Ba, 2015) with parameters $\beta_1 = 0.9$, $\beta_2 = 0.999$, batch-size of $512$ data-points. The learning rate is annealed with a linear ramp from $0$ to $1000$ and from then with a cosine schedule.

**Likelihood evaluation and sample drawing**   We rely on the Dormand-Prince solver (Dormand and Prince, 1980), an adaptive Runge-Kutta 4(5) solver, with absolute and relative tolerance of $1e-5$ to compute approximate numerical solutions of any ODEs. For the rollouts of the SGM SDEs we use a Euler Maruyama predictor and no corrector. Unless stated we use 100 step rollouts.

**Hardware**   Models are trained on a cluster with a mixture of GeForce RTX 1080, 1080 Ti and 2080 Ti GPU cards.

## O.1   Sphere

**Data**   We randomly split the datasets intro training, validation and test datasets with $(0.8, 0.1, 0.1)$ proportions. In each case the earth is approximated as a perfect sphere.

**Models**   The mixture of Kent distributions (Peel et al., 2001) were optimised using the EM algorithm and the number of components were selected from a grid search over the range $5, 10, 15, 20, 25, 30, 40, 50, 75, 100$, based on validation set likelihood and 250 EM iterations. The number of components selected were: Volcano 25, Earthquake 50, Flood 100 and Fire 100.

For the stereographic SGM–which is a standard SGM with an Ornstein–Uhlenbeck process followed with the inverse stereographic projection–we found $\beta_{\min} = 0.001$ and $\beta_{\max} = 2$ to work best.

**Optimization**   The score-based models are trained for $600k$ iterations for all datasets but 'Flood' where $300k$ performed best.

**Additional experimental results**

**Approximate forward sampling**   Standard Euclidean SGMs rely on a Ornstein–Ulhenbeck (OU) forward process (1) which can easily be simulated since $\mathbf{X}_t|\mathbf{X}_0$ is Gaussian. In contrast, for most manifolds one has to rely on an approximate sampling scheme—see Sec. 3.3. First, we directly assess the quality of the approximate samples $\hat{\mathbf{X}}_t|\mathbf{X}_0$ obtained via geodesic random walk (GRW), against 'exact' samples $\mathbf{X}_t|\mathbf{X}_0$ which are obtained by using a high number of discretization steps

---

[12]https://anonymous.4open.science/r/rimannian-score-sde

($N = 1000$). We report on Fig. S4a the discrepancy between these distributions for different values of discretization steps $N$, as measured by maximum mean discrepancy (MMD) (Gretton et al., 2012). We see that from $N = 5$ the approximate samples are very closely distributed to the true samples. Then, in order to assess the impact of this approximation on the RSGMs' performance, we report on Fig. S4b the log-likelihood when varying the number of discretization steps $N$. We similarly observe that apart from very small values of $N$, the models' performance is very robust to the approximation quality of the forward sampling samples.

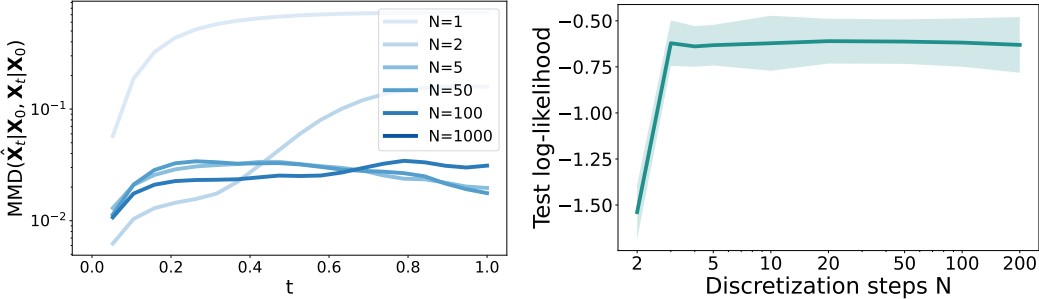

(a) Maximum mean discrepancy (MMD) distance between *'exact'* (i.e. approximated with $N = 1000$ steps) $\mathbf{X}_t|\mathbf{X}_0$ and *approximate* $\hat{\mathbf{X}}_t|\mathbf{X}_0$ at for every $t \in [0, 1]$.

(b) Test log-likelihood of trained RSGMs on the Flood dataset while varying the number of discretization steps $N$ when simulating forward sampling $\mathbf{X}_t|\mathbf{X}_0$.

Figure S4: Ablation study on the impact of the forward sampling approximation quality on $\mathbb{S}^2$.

**DSM loss** $\ell_{t|0}$     On Fig. S5, we show how the test log-likelihood varies with respect to the two hyparameters of the DSM loss, by training RSGMs over a grid of values for $\tau$ and $J$ on the Flood dataset. We can see that the Varadhan approximation by itself ($\tau = 1$) yields descent performance, although a wise combination of Varadhan approximation with a truncation of the heat kernel can give even better results. The performance is relatively robust to the choice of such hyperparameters as long as $\tau$ and $J$ are high enough.

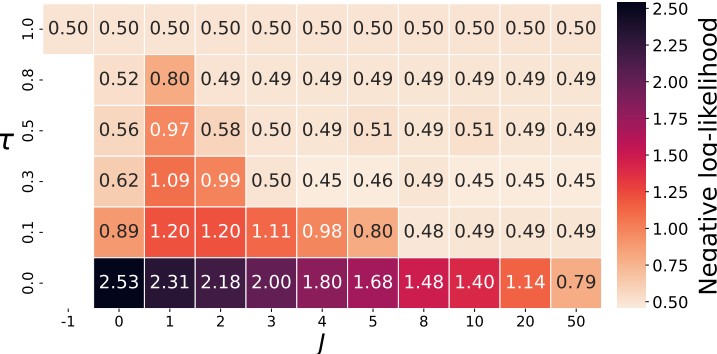

Figure S5: Ablation study on the denoising score matching (DSM) loss $\ell_{t|0}$ when combining the heat kernel truncation and the Varadhan approximation: $\nabla_{x_t} \log p_{t|0}(x_t|x_0) \approx \mathbb{1}(t \leq \tau) \exp^{-1}_{x_t}(x_0) + \mathbb{1}(t > \tau) S_{J,t}(x_0, x_t)$.

## O.2    Torus

**Data**     The synthetic data trained on consists of a wrapped Gaussian distribution on $\mathbb{T}^n$ with uniformly chosen random mean and standard deviation of $0.2$. Such a distribution is defined by taking the density of a Normal distribution in the tangent space of the manifold at the mean and passing it through the exponential map at the mean.

**Architecture**     To parametrize the vector field on $\mathbb{T}^n$ we use a single filed per dimension pointing in a consistent direction around the $i^{th}$ component in the product, with unit norm.

**Models**  All models were trained with the same 3 layer, 512 units per layer MLP across different dimension sizes.

**Optimization**  The models are optimized for $50k$ iterations. The RSGM models are trained with both the implicit score-matching loss and the sliced score-matching loss.

### O.3  Special Orthogonal group

Applications of orthogonal constraints span various fields, such as protein docking with ligands binding pose prediction (Ganea et al., 2022), robotics and Computer vision with rigid body transformation estimation (Barfoot et al., 2011; Prokudin et al., 2018), and medical imaging for data alignment (Hou et al., 2018).

**Data**  We consider the synthetic dataset consisting of samples in $\mathrm{SO}_3(\mathbb{R}^d)^{13}$ from the mixture distribution with density $p(Q) = \frac{1}{K} \sum_{k=1}^{K} \mathrm{N}^W(Q|Q_k, \sigma_k^2)$ with $K \in \mathbb{N}$, where for any $k \in \{1, \ldots, K\}$, we have that $Q = Q_k \exp_{\mathrm{Id}}[\sigma_k \hat{z}]$ with $z \sim \mathrm{N}(0, \mathrm{Id}_{\mathbb{R}^3})$ satisfies $Q \sim \mathrm{N}^W(Q_k, \sigma_k)$ and $(\cdot)^\wedge : \mathbb{R}^3 \to \mathfrak{so}(3)$. For any $k \in \{1, \ldots, K\}$, we set $Q_k \sim \mu$ where $\mu$ is the uniform distribution on $\mathrm{SO}_3(\mathbb{R})$ and $\sigma_k^2 \sim \mathrm{IG}(\alpha = 100, \beta = 1)$, where IG is the inverse Gaussian distribution. We choose $K = 32$ mixture components. We showcase a conditional sampling extension of our model—see App. M for more details— by targeting individual mixture components $p(Q|k)$. Our model is trained using the $\ell_{t|0}$ (DSM) loss along with the Varadhan asymptotic approximation, see (8).

**Architecture**  To parametrize the vector field, we rely on the basis of the Lie group, $\mathfrak{so}(n) = \{A \in \mathrm{M}_d(\mathbb{R}) : A^\top = -A\}$ given by $\mathrm{E}_{ij} = \mathrm{U}_{ij} - \mathrm{U}_{ji}$ for $i, j \in \{1, \ldots, d\}$ with $i < j$ and $\mathrm{U}_{ij} = (\delta_{ij}(k, \ell))_{1 \le k, \ell \le d}$, which induces a basis on the tangent spaces $\mathrm{T}_Q \mathrm{SO}_d$ for any $Q \in \mathrm{SO}_d(\mathbb{R})$ given by $\{Q\mathrm{E}_{ij}\}_{1 \le i < j \le d}$. This is the divergence-free vector field approach described in Sec. 3.4.

**Models**  We compare our proposed approach against Moser flows (Rozen et al., 2021) and a wrapped-exponential baseline (Falorsi et al., 2019) defined as the pushforward along the transformation $\mathbb{R}^3 \xrightarrow{F_\theta^{-1}} \mathbb{R}^3 \xrightarrow{g} \mathbb{R}^3 \xrightarrow{\wedge} \mathfrak{so}(3) \xrightarrow{\exp} \mathrm{SO}_3(\mathbb{R})$ with $F_\theta^{-1}$ denoting the approximate time-reversed diffusion, $g$ denoting the radial operator defined by $g : x \mapsto 2\pi \tanh(\|x\|)x/\|x\|$, $(\cdot)^\wedge : \mathbb{R}^3 \to \mathfrak{so}(n)$ the isomorphism given by the basis on $\mathfrak{so}(3)$ and $\exp$ the matrix exponential. The radial $g$ operator's constant $2\pi$ is chosen as the injectivity radius of the group so that the transformation $\tanh \circ \wedge \circ \exp$ is injective (the set of elements with no preimage is then only the cut locus which is known to have measure zero). Henceforth, this wrapped-exponential transformation cannot be bijective, it is either injective *or* surjective depending on the choice of radius in the radial operator $g$.

**Optimization**  Models are trained for $100k$ iterations. The Riemannian SGM is trained with the Varhadan approximation of the denoising score-matching loss (DSM) Sec. 3.4, and the wrapped-exponential model relies on the exact DSM loss. After a first hyperparameter exploration, a grid search is performed over `learning_rate` $\in [2e-5, 4e-5]$, for SGMs over $\beta_f \in [0.5, 1, 2, 4, 6, 8, 10]$ and for Moser flows over $K \in [1000, 10000]$ and $\lambda_{\min} \in [1, 10, 100]$.

---

[13]This manifold is 3-dimensional.