# OpenReview forum: "Riemannian Score-Based Generative Modelling"
_NeurIPS.cc/2022/Conference — NeurIPS 2022 Accept_

### Official Review · Reviewer_z9yP · 2022-07-01

**Rating:** 7
**Confidence:** 2
**Soundness:** 4 excellent
**Presentation:** 3 good
**Contribution:** 3 good

**Summary:**

In this paper, the authors propose an extension of score based generative modeling to data supported on a compact Riemannian manifold. To that end, concepts the need to be extended from the classical Euclidean setting include an easy to sample reference distribution (uniform on the manifold), that can be further seen as the limit of a noising process (brownian motion on the manifold), as well a an SDE reverting the noising process, in order to be able to generate samples by integrating this SDE thanks to geodesic random walks. The authors also present how to obtain score estimation on the manifold by using the heat kernel, as well as the parametization of vector fields as neural networks, and finally a way to obtain exact likelihood computations, as in the Euclidean case. They obtain a convergence result showing that when the score estimation cost function gets smaller, their model obtains a density that gets arbitrarily close to the true one in terms of Wasserstein-1 distance. They show that their model compares favorably in terms of computional complexity to riemannian continuous normalizing flows or Moser flows. Several experiments real goescience data (on the sphere) and synthetic data on the torus and the special orthogonal group support those claims.

**Questions:**

- Where does the denoising score matching (equation below equation 2) identity come from ? I couldn’t find the same equation in [Song et al. 2021b], where the score matching objective is obtained from an integration by parts?

- Here it makes sense to consider a uniform distribution as the simple distribution to start from, thanks to the convergence result of Brownian motion on compact manifolds to that distribution. Could other possibilities be envisioned? Has this case already been treated in the Euclidean case, where typically one chooses a normal distribution for the base distribution (and if so, how does it compare to the classical score based with a gaussian base distribution)?

**Limitations:**

The main limitation of the method is the restriction to compact manifolds, which is clearly stated. Other manifolds of practical interest are not covered. Nevertheless, the construction for compact manifolds is already sound and covers a certain number of practical cases of interest. Especially, this construction translates into what seems to be efficient algorithms, which is often a drawback of geometric methods operating on manifold valued data.


**Strengths And Weaknesses:**

Strengths :

- The authors are able to extend score based generative models that have now become very popular to compact Riemannian manifolds by generalizing or adapting all the necessary ingredients to this setting.

- The theoretical construction and guarantees are extensive and show that the propose method converges and has a better computational complexity than existing alternatives for manifold valued data. In particular one appealing feature is the scalability to higher dimensional manifolds that is verified in the experiments (up to 200 on the torus data and 64 in the special orthogonal data in the synthetic experiments), while competing approaches cannot deal efficiently (or in the extreme case cannot even run) with such high dimensional non euclidean data


Weaknesses :

- The approach is only limited to compact manifolds, which encompass a number of interesting cases already but leave out many more (SPD matrices, hyperbolic spaces…) . The main difficulties are the convergence of the Browian motion to the uniform distribution that only holds in the compact case, and the manipulation of the transition kernel of the Brownian motion in that case.

- The only real case study with geoscience data does not show clear improvement over other methods, most likely because the problem is too simple and the dimension of the manifold (2 here) is not high enough for the computational benefits of the proposed approach to be visible. Maybe it would be possible to find a real application in higher dimensions where the method can be shown to clearly outperform others?

---

> ### Author Response · Authors · 2022-08-02
> **Answer to Reviewer z9yP**
>
> > The approach is only limited to compact manifolds, which encompass a number of interesting cases already but leave out many more.
>
> We completely agree that there are numerous non-compact manifolds of interest. Our manuscript did not emphasize the more general nature of some of the theoretical results (we have revised our paper to show that Theorem 1 and Proposition 3 hold even in non-compact settings). We have added a discussion on the non-compact setting in the revised version (see Appendix M). Our methodology can be extended in a straightforward manner to the non compact setting by making the following changes:
> * Define a forward noising process on the (non compact) manifold targeting an easy to sample distribution A discussion of how to do this is in Appendix M. The forward process and the target distribution can still be sampled using geodesic random walk.
> * Our time reversal results still hold in the non-compact setting. In fact our only assumption for Theorem 1 is that the manifold admits a Riemannian metric.
> * Since the time-reversal result holds we can define the backward process and sample from it using geodesic random walks once the score is learned.
> * The last step is to learn this score function which can be done using the same tools as the compact setting since the DSM and ISM loss do not require the manifold to be compact.
>
> To summarize, the only change one has to make to extend our methodology to non compact manifolds resides in the choice of a reference distribution and a process targeting this distribution.
>
> > Here it makes sense to consider a uniform distribution as the simple distribution to start from, thanks to the convergence result of Brownian motion on compact manifolds to that distribution. Could other possibilities be envisioned? Has this case already been treated in the Euclidean case, where typically one chooses a normal distribution for the base distribution?
>
> We agree with the reviewer that other possibilities could be envisioned. To our knowledge a few works exist to modify the forward trajectory. This has been done in the Euclidean setting. For example [2] also parameterize and learn the forward dynamics. In [3], the diffusion model is defined in a latent space of VAE and therefore the noising process in the latent  space process of the VAE corresponds to a non-Gaussian process once pushed onto the image space by the decoder. In another line of work, [4,5,6] study Schrodinger Bridges and learn the optimal forward process given a reference diffusion. We also emphasize that the choice of a Brownian motion as a forward dynamics has also been studied in the Euclidean space, see [7] for instance. In this case the forward process does not admit an invariant probability measure since its invariant measure is the Lebesgue measure although it can be approximated by a Gaussian with variance $T$. Using specific forward dynamics allow for faster sampling and specific choices of reference measures for dataset interpolation for instance
>
> > The only real case study with geoscience data does not show clear improvement over other methods [...]. Maybe it would be possible to find a real application in higher dimensions where the method can be shown to clearly outperform others?”
>
> We highlight three possible high dimensional applications of our work: robotics, lattice QCD and protein modeling.
> * In robotics, one can refer to the CMU motion datasets (http://mocap.cs.cmu.edu/) in which each element is represented by a unit vector in $\mathbb{R}^3$ (total dimension $(\mathbb{S}^2)^{30}$).
> * Lattice quantum field theory, and specifically lattice quantum chromodynamics are methods at the forefront of understanding standard model physics. In these problems, the state space is a product of gauge groups, in QCD $SU(3)$ on a 4D grid. Current SoTA methods work on grids of $96^4$, and so scalable methods are required.
> * Finally, one possible application is molecule/protein modeling. Indeed, proteins are often represented either as a set of torsional/dihedral angles ($(\mathbb{S}^1)^n$) or as a set of positions/basis ($\mathbb{R}^3$ and $SO_3(\mathbb{R}$). Our RSGM can then be used to estimate diffusion models on $SO_3(\mathbb{R})$. Some papers build upon work to tackle this challenge by considering diffusion models on the $n$-dimensional torus.
>
> In this paper, we do not develop further these applications as their specific study would deserve a publication  in its own right (see for instance the many diffusion papers on protein modeling [8,9,10] that build upon our work). In that respect, we view our paper as laying the foundations for further applications of diffusion models in these fields.
> The synthetic experiments we perform while not “real life” tasks, are in fact quite non-trivial, and we believe show our methods will be useful in the domains discussed. In addition they show the inherent shortcomings of comparative methods well, and demonstrate the benefits of a score-based formulation.

---

> > ### Author Response · Authors · 2022-08-02
> > **Answer to Reviewer z9yP (continued)**
> >
> > > Where does the denoising score matching (equation below equation 2) identity come from ?
> >
> > This equation is sometimes called the Tweedie identity and can be derived as follows:  $\nabla \log p_t(x) = \nabla p_t / p_t = \int_M \nabla p_{t|0}(x|x_0) p(x_0) \mathrm{d} p_{ref}(x_0) / p_t(x_t) = \int_M \nabla \log p_{t|0}(x|x_0) \mathrm{d} p_{0|t}(x_0|x)$. This representation directly yields the DSM loss. However, we agree with the reviewer that the DSM loss can also be obtained by integration by part from the ISM loss (which itself can be derived without using the Tweedie identity).
> >
> > [1] A Wrapped Normal Distribution on Hyperbolic Space for Gradient-Based Learning – Nagano (2019)
> >
> > [2] Diffusion normalizing flow – Zhang et al. (2021)
> >
> > [3] Score based generative modeling in latent space – Vahdat et al. (2021)
> >
> > [4] Solving Schrodinger Bridge via Maximum Likelihood – Vargas et al. (2021)
> >
> > [5] Diffusion Schrodinger Bridge, with Application to generative modeling – De Bortoli et al. (2021)
> >
> > [6] Likelihood training of Schrodinger Bridges using FBSDE – Chen et al. (2021)
> >
> > [7] Score based generative modeling through SDE – Song et al. (2021)
> >
> > [8] Diffusion probabilistic modeling of protein backbones in 3D for the motif-scaffolding problem – Brian L. Trippe et al. (2022)
> >
> > [9] GeoDiff: a Geometric Diffusion Model for Molecular Conformation Generation – Minkai Xu et al. (2022)
> >
> > [10] A Score-based Geometric Model for Molecular Dynamics Simulations – Fang Wu et al. (2022)

---

### Official Review · Reviewer_X25L · 2022-07-10

**Rating:** 7
**Confidence:** 3
**Soundness:** 4 excellent
**Presentation:** 3 good
**Contribution:** 3 good

**Summary:**

This work extends score-based generative modeling with time-reversed diffusion processes from Euclidean geometry to compact Riemannian manifolds. Several modifications have been made to adapt the method to compact Riemannian manifolds, including using Brownian motion in forward pass, uniform base distribution, and geodetic random walks in forward/backward sampling.  Experiments on real and synthetic datasets show the effectiveness of proposed methods in modeling the symmetry in data.

**Questions:**


#### Strengths
- This work proposes novel methods for extending score-based generative modeling techniques beyond Euclidean geometry.
- This paper is clearly written, covering both the big picture of the considered research topic and the technical details of the proposed method.
- Both theoretical convergence analysis and empirical experimental results are included in the paper to show the validity of the proposed approach.

#### Weaknesses
- It is not quite clear to me whether the assumption of compactness of the Riemannian manifold can be too limited for the consideration of generative modeling for real-world applications. It would be more helpful to provide mode detailed analysis from this perspective.
- For the consideration of scalability, stochastic approximations are needed to obtain a tractable training objective.
- Most of the experimental results are on synthetic datasets, and the choices of real-world datasets satisfying the assumptions of this work seem to be limited.



**Limitations:**

- Have the authors tried different parametrizations for the score function, i.e. using the clean data or noise as the prediction target similar to DDPM?
- How general are the proposed approach in terms of the applicability of various samplers or ODE solvers? like DDIM and other advanced solvers.
- How large the gap is to extend this work to Riemannian manifolds which are non-compact or with boundaries or implicitly defined?

**Strengths And Weaknesses:**

Strength: the originality and quality of the theoretical part seem good, and the paper is clearly written.  The extension to the Riemannian manifold is a reasonable idea to consider. The experiments show that this algorithm can work in some cases.

Weakness: The main weakness of the paper is its significance. First, very few datasets really lie on some compact Riemannian manifold without boundary (very very restrictive in terms of math, in fact. ). This setting is more like a mathematical toy to me. This is why the experiments are mainly about synthetic datasets. Second, there are some questions about its feasibility. For example, why uniform distribution on a compact manifold is easy to sample? It highly depends on how the manifold is represented in the memory. So this paper is more about math toys not about solving real machine learning problems.

---

> ### Author Response · Authors · 2022-08-02
> **Answer to Reviewer X25L**
>
> Thank you for your comments and thoughtful questions
>
> >It is not quite clear to me whether the assumption of compactness of the Riemannian manifold can be too limited for the consideration of generative modeling for real-world applications. [...] Most of the experimental results are on synthetic datasets, and the choices of real-world datasets satisfying the assumptions of this work seem to be limited.”
>
> We completely agree that there are numerous non-compact manifolds of interest. Our manuscript did not emphasize the more general nature of some of the theoretical results (we have revised our paper to show that Theorem 1 and Proposition 3 hold even in non-compact settings). We have added a discussion on the non-compact setting in the revised version (see Appendix M). Our methodology can be extended in a straightforward manner to the non compact setting by making the following changes:
> * Define a forward noising process on the (non compact) manifold targeting an easy to sample distribution A discussion of how to do this is in Appendix M. The forward process and the target distribution can still be sampled using geodesic random walk.
> * Our time reversal results still hold in the non-compact setting. In fact our only assumption for Theorem 1 is that the manifold admits a Riemannian metric.
> * Since the time-reversal result holds we can define the backward process and sample from it using geodesic random walks once the score is learned.
> * The last step is to learn this score function which can be done using the same tools as the compact setting since the DSM and ISM loss do not require the manifold to be compact.
>
> To summarize, the only change one has to make to extend our methodology to non compact manifolds resides in the choice of a reference distribution and a process targeting this distribution.
> We believe that there are in fact a number of very interesting problems that lie on compact manifolds. We introduce a number of these in the introduction of the paper, but expand on them here.
> * First, in the field of Robotics, states are often represented as a collection of joints which can be described as torsion angles or 3D rotations depending on the degrees of freedom. See for instance the CMU motion datasets (http://mocap.cs.cmu.edu/).
> * Secondly, molecules and proteins are often represented either as a set of torsional/dihedral angles ($(\mathbb{S}^1)^n$) or as a set of positions/basis ($\mathbb{R}^3$ and $SO_3(\mathbb{R}$). Our RSGM can then be used to estimate diffusion models on $SO3(\mathbb{R})$. This methodology has already been successfully applied by recent papers building on top of our work (we are hesitant to provide references at this stage given the nature of the double blind review, but can at the reviewers request).
>
> Two of the three experiments in the paper are synthetic, whilst Section 5.1 deals with real datasets rooted in earth and climate science. While these tasks are synthetic, they are more challenging than they might appear, and still represent a challenge for existing Riemannian generative models as empirically shown in our experiments.
>
> > why uniform distribution on a compact manifold is easy to sample?
>
> In this work, we focused on the uniform distribution since one can sample exactly and efficiently from it on the manifolds we have considered . Also it can be seen as a ‘natural’ choice for a prior since it is the maximum entropy probability distribution. Note that an approximate fast sampler of the uniform distribution can be defined on every compact manifold since the Brownian motion converges geometrically fast to the uniform distribution. To do so, start from any given point or distribution on the manifold and then simulate Brownian trajectories on the manifold for large time $T$ (note that the time $T$ needed to obtain approximate samples from the uniform distribution depends linearly on $1/\lambda_1$, where $\lambda_1$ is the smallest non-negative eigenvalue of the Laplace-Beltrami operator).
>
> However, if an application setting suggests choosing a specific reference distribution then our RSGM can be applied as long as the target distribution admits a density w.r.t. the volume form. To do so, we add a drift term in the forward process  that is given by the gradient of the logarithmic target density.  These are the only modifications needed to obtain a diffusion model with a non-uniform reference density. We have added this discussion in the revised paper (Appendix M).

---

> > ### Author Response · Authors · 2022-08-02
> > **Answer to Reviewer X25L (continued)**
> >
> > > “For the consideration of scalability, stochastic approximations are needed to obtain a tractable training objective.”
> >
> > If the reviewer refers to the stochasticity of the Hutchinson estimator in the estimation of the ISM loss then we fully agree that this stochasticity (and the number of samples required to obtain an estimator with low variance) may depend on the dimension.  Figure 2 demonstrates that relying on an approximate divergence is not necessarily an issue, even in high dimension, as one can see that performance is not degraded while maintaining speed.
> > We would like to note though that If one has access to the transition kernel one can use the DSM loss where no stochastic approximation is needed as reminded in Table in Appendix E.
> >
> > > Have the authors tried different parametrizations for the score function, i.e. using the clean data or noise as the prediction target similar to DDPM?
> >
> > Predicting "clean data" (without noise) is not possible or extremely difficult in the manifold setting. Similarly, such approaches for diffusion generative models, such as DPPM, rely on being able to sample $x_{t-1}|x_t, x_0$ for backward sampling, which is typically not available beyond the Euclidean setting with Gaussian forward diffusion.
> >
> > > How general are the proposed approach in terms of the applicability of various samplers or ODE solvers? like DDIM and other advanced solvers.
> >
> > Similar to above, DDIM also relies on closed form sampling from the posterior $x_{t-1}|x_t, x_0$, so is not applicable. However, for solving the likelihood ODE and SDE sampling, any standard acceleration methods that do not require closed form sampling or large step sizes should be applicable.
> >
> > > How large the gap is to extend this work to Riemannian manifolds which are non-compact or with boundaries or implicitly defined?
> >
> > Please see the above answers for a discussion on the extensions to non-compact manifolds. The extension to manifolds with boundary is one we are actively working on. Methodologically the extension is reasonably trivial. One simply has to replace the Brownian motion with a reflected Brownian motion. Theoretically, we are confident this will work out, but it is trickier to deal with than the without boundary case, and so we left it out of this work while we work on the formalism.
> > For implicitly defined manifolds, we believe this is going to be manifold dependent. If one knows the manifold is compact, and one has access to the exponential map or a retraction, then our method as described using Brownian motion as the noising process readily applies. If the implicit manifold is not compact, one would need to know of a suitable noising process on the manifold in addition.

---

### Official Review · Reviewer_uQqV · 2022-07-11

**Rating:** 7
**Confidence:** 3
**Soundness:** 3 good
**Presentation:** 3 good
**Contribution:** 3 good

**Summary:**

This paper explore continuous-time score-based generative models in non-Euclidean space specifically on the Riemannian manifold. The paper defines the forward process of the SGM model directly on the Riemannian manifold and integrates Geodesic Random Walks (GRW) to project the Brownian motion onto a manifold for sampling on the reverse diffusion process. The authors validate their proposal by testing on multiple datasets (e.g. Earth and climate science dataset, Synthetic data on tori, and orthogonal group).

**Questions:**

- How are stereographic score-based models projected and could a similar method (albeit a different projection) be possible for comparison on the tori experiment?
- On the torus experiment, why were the Moser Flows the only benchmark comparison?

**Limitations:**

The authors discusses and addresses the current limitations on their model (e.g. incompatible manifolds).

**Strengths And Weaknesses:**

Strengths:
- The overall paper was clear and organized well. The paper did a good job of translating and introducing the SGM forward noising process for the Euclidean space to the Riemannian forward process.
- The author provides a nice comprehensive overview of related literature works with useful insight on each past work.
- The paper is highly motivated and novel as there have not been any past papers that have applied score-based/diffusion models in a non-Euclidean space.
- Derivations are concise and eloquent, but it would be nice to see the elaborated derivations that are available in the Appendix.

Weaknesses:
- Many of the experiments do not have many benchmark models to compare with the proposed model.
- There is not an explanation on the stereographic Score-Based method. A quick explanation on this model would be helpful.

Additional Remarks:
- Section 5.2: I believe that "torii" should be spelled "tori" as it is the plural form of "torus".

---

> ### Author Response · Authors · 2022-08-02
> **Answer to Reviewer uQqV**
>
> Thank you for taking the time to review our submission and for your constructive feedback.
>
> > There is not an explanation on the stereographic Score-Based method.[...] How are stereographic score-based models projected and could a similar method (albeit a different projection) be possible for comparison on the tori experiment?
>
> In the revised version, we detail our stereographic-wrapped SGM method in the appendix Section N. The goal of the stereographic method was to provide a score-based alternative to intrinsic RSGM. Although as highlighted in Section 4, most manifolds (e.g. the sphere) are not topologically equivalent to $\mathbb{R}^n$, so there does not exist a bijective map $\psi: \mathbb{R}^n \rightarrow \mathcal{M}$ along which SGMs samples could be pushed onto the manifold.The stereographic method works as follows. Given the dataset on the sphere we transform this dataset in a new one on the vector space using the stereographic projection. Then, we apply classical SGM in this Euclidean space (the forward process targeting a unit Gaussian). Once we have learned an Euclidean generative model we project it back on the sphere using the inverse of the stereographic projection. In practice, we found this method to give worse results and be less stable than our intrinsic RSGM.
> We agree with the reviewer that such an approach could also be used in the case of the torus as proposed in [1], since the function $\theta \mapsto \tan(\theta/2  - \pi/2)$ maps $\mathbb{S}^1$ minus $\theta=0$ bijectively onto $\mathbb{R}$.
>
> > Many of the experiments do not have many benchmark models to compare with the proposed mode.
> > On the torus experiment, why were the Moser Flows the only benchmark comparison?
>
> At the time of the writing of the paper, the main SOTA method for density estimation on manifolds was the Moser flow method [2], which is the reason why it is our main baseline. We also compare our method to a mixture of Kent distributions as it is a standard method in directional statistics and to Riemannian Continuous Normalizing Flows [3], the methodology Moser flows innovates on.
> There are few other methods that have been proposed in the literature, and we discuss these in Section 4. In particular, the exponential-map sum-of-radial ﬂow (EMSRE) [3], Mobius Flows [3], IRCPM [4] and  RCPM [5]. Of these methods, [3,5] struggle to scale to high dimensions due to exponential parameter costs (see [4], section 2, for a discussion of this). [4] potentially could scale well with dimension, however the paper itself only provides a proof-of-concept for very low dimensional manifolds, and without any open source code available for this method it is very non-trivial to compare to this method on new problems.
> For the high dimension torus experiments, Moser flows were the only potentially competitive baseline as they were the only other method giving sensible results in a reasonable amount of time.
> We hope this provides a sufficient explanation of our choice of baselines, and our reasons for not including the other available methods in the literature. We would however be happy to run any particular baselines the reviewers would like to see comparisons to during the discussion period.
>
> > “Derivations are concise and eloquent, but it would be nice to see the elaborated derivations that are available in the Appendix.”
>
> In the fortunate event where the paper were to be accepted we would use the extra space to provide short proof sketches to help the reader understand the key arguments of Theorem 1 and Theorem 3.
>
> [1] Normalizing Flows on Tori and Spheres – Rezende et al. (2020)
>
> [2] Moser Flow: Divergence-based Generative Modeling on Manifolds – Rozen et al. (2021)
>
> [3] Riemannian Continuous Normalizing Flows – Mathieu et al. (2020)
>
> [4] Implicit Riemannian Concave Potential Maps – Rezende et al. (2021)
>
> [5] Riemannian Convex Potential Maps -  Cohen et al. (2021)

---

> > ### Author Response · Authors · 2022-08-08
> > **Closing of the discussion period**
> >
> > Dear Reviewer,
> > As the discussion period is closing we would like to check whether you had a chance to look at our rebuttal.
> > We do hope that our justification of the choice of baselines (and the new results and clarifications included in our paper) have resolved the concerns raised in your review.
> > Could you please confirm if we have addressed your concerns?

---

> > > ### Comment · Reviewer_uQqV · 2022-08-09
> > > **Thank you for the reply**
> > >
> > > Dear author, I appreciate the clarification and I understand your justification for the baseline model comparison. I think your paper is very strong and promising, thus I will raise my score.

---

### Official Review · Reviewer_Rw15 · 2022-07-14

**Rating:** 9
**Confidence:** 4
**Soundness:** 4 excellent
**Presentation:** 3 good
**Contribution:** 4 excellent

**Summary:**

The paper proposes a score-based generative model (SGM) on compact Riemannian manifolds. The paper first points out three major components that contribute to the success of SGMs in Euclidean spaces.
Linear SDEs in Euclidean spaces (more precisely, OU processes) have stationary distributions and are Normal distributions.
For the SDEs, the reverse diffusions exist, whose initial distributions are the stationary distributions above, and the form of the reverse SDEs is known, yielding efficient parameterizations.
The solutions of OU processes are Normal distributions; thus, it is easy to sample and compute loss in denoising score matching (DSM).
The parametric reverse diffusions can be trained via minimizing various score matching losses, including DSM.
Thus, the paper proposes those counterparts for SGMs on compact Riemannian manifolds, specifically by identifying and solving the non-trivialities of the counterparts.

Then, the paper shows that linear diffusions on compact Riemannian manifolds have reverse diffusions, and the resulting SDEs are the same form as diffusions in Euclidean spaces.
However, the paper identifies that for the linear diffusions on compact Riemannian manifolds, its solution for a given data at time 0 doesn't have analytical forms, unlike OU processes. This makes two problems: the difficulty of simulation x_t given x_0 and the difficulty of computing denoising score matching loss.

To address this, the paper proposes using Geodesic random walks (GRW) for simulating both forward noising and reverse denoising processes; note that it is approximate sampling, unlike efficient sampling of the forward noising processes in SGMs on Euclidean spaces.

Fourth, the paper proposes two approximate methods to approximate the conditional log density gradients (or score) $\nabla_{x_t} p(x_t|x_0)$ of the forward noising process. When we know the eigenvalues and eigenvectors of the manifold of interest, one can approximate the conditional scores using the first N eigenvectors, i.e., the Truncation method. One can employ Varadhan's asymptotic, trading-off approximation errors when we don't have such information.

Next, the papers introduce DSM and implicit score matching (ISM) losses, whose computation will involve the simulation and approximation techniques above.

Finally, the paper introduces the neural network-based score function parameterizations suitable for compact Riemannian manifolds. Briefly speaking, when d-dimensional Euclidean spaces can describe the manifold, the suitable parameterization requires n > d sets of vector components.

For the experiments, the paper demonstrates the effectiveness of the proposed method, especially by comparing it with other manifold-valued generative models.

**Questions:**

For Varadhan's asymptotic, What can we do when $t$ is large? Does Eq. 7 mean that using the approximation results in biased learning signals when we use the asymptotic to estimate score matching losses?


**Limitations:**

Please find the comments on the Weakness section above. In general, the paper's contributions wrt the novelty are clear, and the proposed methods are well-defined. In addition, I found that the paper has a well-organized structure that makes it clear to understand the proposed methods. Thus, I'm inclined to accept the paper.

**Strengths And Weaknesses:**

**Strength**

In my understanding, the paper's contributions are clear, and I also consider that the results are essential for several reasons:
  - The paper provides novel manifold-valued generative models based on SGMs.
 - The paper motivates each proposed method well so that readers can understand how each method addresses the challenges of designing manifold-valued SGMs.
 - I found that the paper has a well-organized structure that makes it clear to understand the proposed methods.
 - The paper provides sufficient information about their theoretical claims, helping readers understand the proposed methods' characteristics and limitations.

**Weakness**
The followings can be improved.
 - Clarifying the terminologies will help readers to understand the paper more clearly. For instance, one can mention that "heat kernel" means "(fundamental) solutions of (linear) SDEs". I understand such terms are widely accepted in SDE literature but possibly confuse the ML communities.
 - The papers can introduce notations before introducing the main theoretical claims. As the paper proposes technically demanding methods, the clarifications of the notations seem necessary beforehand—for instance, exponential mapping notation is introduced in line 203 but appears in Eq 7.

---

> ### Author Response · Authors · 2022-08-02
> **Answer to Reviewer Rw15**
>
> Thank you for your positive comments and feedback.
>
> > Clarifying the terminologies will help readers to understand the paper more clearly.[...] The papers can introduce notations before introducing the main theoretical claims.
>
> Following your advice, we clarified the notations in the updated submission so as to make it more accessible to the ML communities. We have also revised the paper so that the exponential mapping is introduced before it is used. We also point out that in case the paper is accepted we will use the extra space to provide short proof sketches to help the reader understand the key arguments of Theorem 1 and Theorem 3.
>
> > For Varadhan's asymptotic, What can we do when t  is large?”
>
> Indeed, the use of Varadhan’s asymptotic should be restricted to ‘small’ times while for ‘larger’ times one should turn to either a) the ISM loss (which does not use the heat kernel at the price of a divergence computation) or b) to the DSM loss with a truncated heat kernel. However, we empirically found out and reported in Figure S5 (in the appendix O.1) that on $\mathbb{S}^2$ the Varadhan approximation was surprisingly robust to ‘large’ times. We believe this is due to two reasons: a) matching well the score at smaller times is more important than at larger ones since the score’s norm is relatively small when t is large [1], and b) we conjecture that the Varadhan approximation $\exp_{x_t}^{-1}(x_0)$’s direction is the same as the score’s.
>
> > Does Eq. 7 mean that using the approximation results in biased learning signals when we use the asymptotic to estimate score matching losses?”
>
> You are correct. Since the approximation is only valid in the asymptotic limit $t \to 0$, there is a bias when sampling after training with the Varadhan approximation. This bias is added to the one stemming from the use of neural networks to approximate the score. One of our theoretical results (Theorem 4) takes into account this bias by making the assumption that $\| s - \nabla \log p_t \| \leq M$. More precisely if the score network is within some error margin of the true score everywhere, this will account for the error in this approximation. We have clarified this point in the revised version of our manuscript after Theorem 1.
>
> [1] Soft Truncation: A Universal Training Technique of Score-based Diffusion Model for High Precision Score Estimation - Kim et al. (2022)

---

### Author Response · Authors · 2022-08-02
**General comment**

We thank the reviewers for their insightful comments and are encouraged by their positive feedback regarding the proposition, soundness, and clarity of our work. We have uploaded a revised version of the paper with changes highlighted in red. Please note in addition to the red changes, section 3 has been reordered.
We provide detailed responses to each reviewer but summarize here their main feedback:

> Commentary on the limitations of our method to compact manifolds and on the requirement that the target measure is uniform.

We would like to thank the reviewers particularly for questions regarding the scope of our proposed method (i.e. compactness and boundary assumptions). Upon reflection, we may not have been sufficiently clear on this point, but we have been able to improve the paper by adding a new section (Appendix M) which shows how to construct noising processes on (connected and complete yet) potentially non-compact manifolds. We have also clarified the manuscript to show that large parts of our theoretical results still hold in the non-compact setting. We hope that this demonstrates the general nature of our proposed framework, and opens it up to many new applications in the eyes of the reviewers.

> Suggestion on clarifying some of our notations and results to make them more amenable to the machine learning community.

Indeed we agree with the reviewers that, although we made an effort to make the submission accessible for a broad audience, there was still quite some margin for improvement. We have updated the paper with the suggested point on clarity, and will later add an exhaustive notation section in the appendix for ease of reading.

> Request to add additional baselines and to discuss real world applications.

We later discuss in detail our reasons for our choice in baselines, but the short summary of this discussion is that there are not a large number of suitable baselines to compare to in this setting. We do compare ourselves to the SoTA, and to other baselines  that can be reproduced in a reasonable amount of time. There are a few methods that we would have ideally liked to additionally evaluate, yet the non-triviality and the lack of publicly available code was detrimental.
With regard to real world applications, we also discuss in more detail later, but in short, we suggest a few exciting ones, yet these are non-trivial, and tackling them would be publication worthy in its own right. Indeed there are already papers building on our work tackling protein modeling problems. The synthetic experiments we perform in this paper are by no means trivial, and were designed to validate the use of this approach in these new settings.

---

### Meta-Review · Area_Chair_cqCk · 2022-08-25

**Recommendation:** Accept
**Confidence:** Certain

**Metareview:**

The manuscript generalizes score-based generative models to compact Riemannian manifolds. All referees agree that the method is novel, technically sound, and gives nice results. I recommend the paper to be accepted to the conference.

**Award:**

Yes

---

### Decision · Program_Chairs · 2022-09-14

Accept